# Threshold-Based Exclusive Batching for LLM Inference

**Weifang Zhang** [* 1]   **Yuzhou Nie** [* 2]   **Bowen Pang** [3]   **Guangrui Ma** [4]   **Shining Wu** [1]

## Abstract

Mixed batching (MB)—interleaving prefill and decode in a single batch—has become the standard scheduling strategy for large language model (LLM) inference due to its efficiency in maximizing compute and memory utilization. However, through controlled experiments, we find that prefill–decode interference inflates MB's per-step marginal cost above that of pure decode. On the high-bandwidth H200 (4.8 TB/s), this occurs only when decode tokens exceed 80% of the batch; however, on the bandwidth-constrained RTX PRO 6000 (1.792 TB/s), this threshold plummets to just 20%. Consequently, the optimal choice between MB and exclusive batching (EB) fundamentally depends on GPU memory bandwidth, model size, and workload composition. We derive a closed-form condition for this EB–MB performance crossover, along with asymptotically optimal phase-switching thresholds and memory-safe batch sizing for EB. Optimized EB achieves up to 41.9% higher throughput on bandwidth-constrained GPUs, while MB retains its advantage on high-bandwidth hardware with larger models. Our hybrid scheduler EB+ applies this condition online to dynamically switch between EB and MB without manual intervention. Under non-stationary traffic with distribution or concurrency shifts, EB+ attains the highest or near-highest throughput in every setting, outperforming MB by up to 36.4%.[1]

## 1. Introduction

Large language model (LLM) inference consists of two distinct phases with fundamentally different computational characteristics. The *prefill* phase processes input tokens in parallel to populate the key-value (KV) cache, making it compute-bound. The *decode* phase generates tokens autoregressively, requiring repeated memory accesses to the KV cache, making it memory-bandwidth-bound (Kwon et al., 2023; Pope et al., 2023; Wang et al., 2025). This dichotomy creates an inherent inefficiency: during decoding, GPU compute units remain underutilized, while during prefill, memory bandwidth is not fully exploited.

Two dominant scheduling paradigms have emerged. *Mixed batching* (MB) (Agrawal et al., 2023) interleaves prefill and decode operations within the same batch, simultaneously utilizing GPU compute for prefill and memory bandwidth for decode. *Exclusive batching* (EB) processes prefill and decode in separate batches, alternating phases by a scheduling rule. In this work we study a *capacity-triggered* policy that switches to the prefill phase whenever $k$ decode slots become idle, denoting this family EB($k$). MB has gained widespread adoption, with major inference engines including vLLM (Kwon et al., 2023) and SGLang (Zheng et al., 2024) transitioning to MB as their default scheduling mode. Throughout the paper we use v1 to denote vLLM v1 (as the MB baseline) and v0 to denote the vLLM v0 exclusive-batching scheduler, equivalent to EB($k{=}1$) when saturated.

In the context of single-GPU deployments, an intriguing dichotomy nonetheless persists in practice: while Western inference engines have largely standardized on MB, many large-scale production systems in China continue to favor EB. A plausible contributing factor is hardware—GPUs accessible in the Chinese market operate under tighter memory-bandwidth budgets, owing in part to export restrictions. This raises a concrete question: does memory bandwidth fundamentally alter the EB-vs-MB trade-off, and if so, when should each strategy be preferred?

To probe this question, we examine the marginal cost of processing tokens under different batching disciplines. We model the iteration time for processing $n_{\text{tok}}$ tokens in a single batch as $T_{\text{iter}} = \alpha + \beta \cdot n_{\text{tok}}$, where $\alpha$ is the fixed overhead and $\beta$ the marginal cost per token. Crucially, both $\alpha$ and $\beta$ depend on (1) the hardware profile (e.g., available memory bandwidth) and (2) the batch composition, which we characterize by the *decode ratio* $r := n_{\text{decode}}/n_{\text{tok}}$, the fraction of decode tokens in a batch ($r{=}1$ for pure decode;

*Equal contribution [1]The Hong Kong Polytechnic University [2]UC Santa Barbara [3]Technology and Engineering Center for Space Utilization, Chinese Academy of Sciences [4]Meituan. Correspondence to: Shining Wu <sn.wu@polyu.edu.hk>.

*Proceedings of the 43rd International Conference on Machine Learning*, Seoul, South Korea. PMLR 306, 2026. Copyright 2026 by the author(s).

[1]Code: https://github.com/weifang231/eb-vllm.

$0<r<1$ for a mixed batch).

Controlled experiments on two GPUs with contrasting bandwidth—NVIDIA RTX PRO 6000 (1.792 TB/s) and NVIDIA H200 (4.8 TB/s)—reveal a sharp hardware dependence. The crossover point at which the mixed-batch marginal cost exceeds the pure-decode cost falls at $r \approx 20\%$ on the RTX PRO 6000 but only at $r \approx 80\%$ on the H200 (Figure 1; averaged over 20 runs, variance negligible). These observations lead to our central hypothesis: EB with an optimized threshold should outperform MB in bandwidth-constrained environments, while MB should retain its advantage on high-bandwidth hardware, motivating both an analysis of the EB–MB crossover and the design of adaptive EB scheduling.

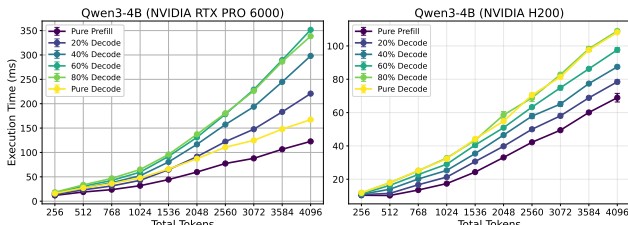

*Figure 1.* Execution time for mixed batches (Qwen3-4B, ctx=16) on RTX PRO 6000 (left) and H200 (right).

Kernel profiling (Figure 2) localizes the source of this gap: for a fixed total token count, GEMM time is largely invariant to $r$, whereas Attention time grows with $r$. Mixed-batch Attention becomes slower than pure-decode Attention at $r \approx 20\%$ on the RTX PRO 6000, but only near $80\%$ on the H200. Consistent patterns hold across model scales (Gemma-3-1B-IT, Qwen3-8B, Qwen3-30B-A3B) and decode context lengths (Appendix A.2): both curves shift uniformly without altering their shape or crossover points.

This cross-GPU gap is consistent with bandwidth limitations (Williams et al., 2009): decode attention streams the full KV-cache context per token and is therefore memory-bandwidth-bound. On bandwidth-constrained GPUs, co-locating prefills and decodes intensifies bandwidth contention and disproportionately inflates Attention latency; on high-bandwidth GPUs, this interference is weaker. This also suggests that current FlashAttention kernels (Dao et al., 2022; Dao, 2024; Shah et al., 2024) are not fully optimized for mixed-batch inference on bandwidth-limited GPUs, motivating a re-examination of the EB-vs-MB trade-off.

Based on these findings, we derive a closed-form condition characterizing when EB outperforms MB, governed by the marginal-cost gap between mixed and exclusive batches (Section 3). To realize optimal EB in practice, we derive an analytical expression for the normalized threshold $\theta^* = k^*/N$ and develop an adaptive scheduler, denoted EB($\hat{k}^*$), that computes $\hat{k}^*$ online from workload characteristics at

runtime. We further use the same crossover condition to drive online mode selection between EB and MB.

This paper makes the following contributions:

- We provide empirical evidence that MB does not universally dominate EB, and derive a closed-form condition for the EB–MB crossover governed by the marginal-cost gap and amortized fixed-cost advantage.
- We develop an adaptive EB scheduler with closed-form, asymptotically optimal phase-switching thresholds under stochastic output-length distributions, paired with memory-aware batch sizing that has probabilistic OOM guarantees.
- We propose EB⁺, a hybrid scheduler that applies the crossover condition online to switch between EB and MB at runtime, remaining robust under non-stationary traffic and matching or exceeding PD-disaggregation throughput without extra hardware.
- We validate these results on four NVIDIA GPUs spanning 0.9–8.0 TB/s bandwidth: EB($\hat{k}^*$) achieves up to 41.9% throughput gains on bandwidth-constrained GPUs; on high-bandwidth hardware with larger models, v1 retains its advantage; and EB⁺ delivers the highest joint TTFT/TPOT goodput across traffic regimes.

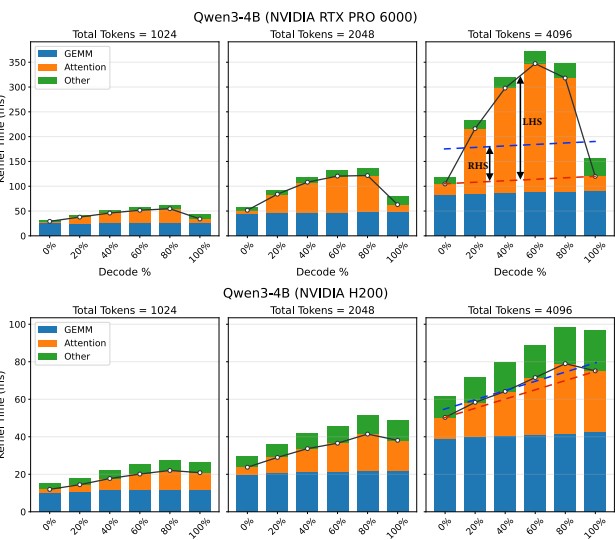

*Figure 2.* Kernel time breakdown for Qwen3-4B on RTX PRO 6000 (top) and H200 (bottom) vs. decode ratio $r$ at three total token counts. Black line: total mixed-batch cost. Right panels add the EB cost (red dashed) and EB-vs-MB threshold (blue dashed, Eq. (7)); EB is favored when the black line exceeds the blue.

## 2. Related Work

The decode phase is memory-bandwidth-bound even at large batch sizes, leaving over 50% of GPU compute idle (Recasens et al., 2025). Its Byte-per-FLOP demand exceeds prefill's by ~100×, and this bottleneck deepens across GPU

generations as compute outpaces bandwidth (Li et al., 2025). Three scheduling strategies emerged in response.

**Exclusive Batching** processes one phase per iteration (Yu et al., 2022; Kwon et al., 2023; NVIDIA, 2023; Aminabadi et al., 2022), avoiding intra-iteration prefill–decode interference; Pang et al. (2025) extend this regime with a Lagrangian analysis of prefill-vs-decode insertion.

**Mixed Batching** co-locates prefill and decode tokens via *chunked prefill*, splitting long prefills under a fixed token budget. Sarathi (Agrawal et al., 2023; 2024) introduced this with decode-maximal batching, with a similar formulation in DeepSpeed-FastGen's Dynamic Split-Fuse (Holmes et al., 2024); it is now the default in vLLM (v1), SGLang (Zheng et al., 2024), TGI (HuggingFace, 2023), TensorRT-LLM (NVIDIA, 2023), and LightLLM (ModelTC, 2023). Chunking itself is orthogonal to EB/MB: EB can also chunk prefills to bound iteration time. MB pays two costs: co-location induces prefill–decode interference, and the aggressive chunking it requires (forced by sharing the token budget with decode) inflates MoE memory traffic by up to 39% via redundant expert reloads (Lee et al., 2026) and scales long-context KV-cache loads as $O(N^2)$ for $N$-chunk prompts (Zhong et al., 2024).

**Disaggregated Serving** eliminates inter-phase interference via dedicated P/D GPU pools (DistServe (Zhong et al., 2024), Splitwise (Patel et al., 2024), Mooncake (Qin et al., 2025)). The cost is twofold: architecturally, KV-cache transfer overhead (severe without NVLink/IB) and a doubled minimum GPU count; operationally, the static P:D split underutilizes decode under imbalance (Shi et al., 2025) and cannot adapt to demand shifts (Hong et al., 2025).

Orthogonal techniques target other layers of the serving stack: MoE serving (Rajbhandari et al., 2022; Gale et al., 2023) and quantization (Xiao et al., 2023; Lin et al., 2024) optimize the model, KV cache management (Zhang et al., 2023; Xiao et al., 2024) compresses memory, and speculative decoding (Leviathan et al., 2023; Miao et al., 2024) reduces decode iterations.

Within the scheduling-policy axis, we optimize EB via a closed-form online controller and combine it with MB into an adaptive hybrid (EB⁺) that navigates the EB/MB trade-off on a single GPU pool.

## 3. Scheduling Model for Exclusive Batching

We develop a scheduling framework for LLM serving under **exclusive batching**, where prefill and decode operations cannot be executed concurrently. We formulate the phase-switching decision as an optimization problem, derive optimal thresholds, and present an online adaptive algorithm.

### 3.1. Problem Formulation

**System Model.** Consider a system with $N$ *slots* (the *maximum batch size*); each slot can hold the state of one in-progress *request* and is either *busy* (currently occupied by a request) or *idle* (vacated upon completion). The system alternates between prefill and decode phases and operates in a saturated regime: the queue is consistently backlogged, so each batch operates at capacity $N$.

During a decode phase, which begins with $N$ busy slots, a slot becomes idle whenever its corresponding request completes (i.e., generates an end-of-sequence token). A *phase switch* occurs when the number of idle slots reaches a predetermined threshold $k$: the system then enters a prefill phase, loading $k$ new requests into the vacated slots.

During this prefill phase, as the $k$ new requests are processed, the $(N - k)$ incomplete requests remain inactive—their KV cache is preserved but no tokens are generated. Once the prefill completes, the system resumes decoding with a full batch of $N$ requests, consisting of the $(N - k)$ ongoing requests and the $k$ newly admitted ones. This dynamic creates a throughput-latency trade-off: **large** $k$ (late switching) leaves completed slots idle longer and inflates queued requests' wait time, while **small** $k$ (early switching) amortizes the fixed prefill overhead $\alpha_p$ over fewer requests and reduces efficiency.

**Timing Model.** Each request has a random input length $L$ (i.i.d. with mean $\mu_L$) and output length $O$ drawn from a general distribution $F$ with hazard rate $h(t)$, mean $\mu_O$, and standard deviation $\sigma_O$. Assuming linear iteration time and modeling each prefill phase as a single batch, the prefill time for a batch of $k$ requests (i.e., the length of a prefill phase) is $T_p(k) = \alpha_p + \beta_p \sum_{i=1}^{k} L_i$, where $\alpha_p$ is the fixed prefill overhead and $\beta_p$ the per-token prefill cost. The decode iteration time at a batch size $n$ is $t_d(n) = \alpha_d + \beta_d n$, and the total decode time $T_d(k; N)$ is the sum of iteration times in a decode phase, depending on both the starting batch size $N$ and the number $k$ of requests that complete before the phase ends.

**Throughput Optimization Problem.** We define throughput as the average number of request completions per unit time and seek the threshold $k$ and batch size $N$ that maximize it subject to a KV-cache memory constraint. Let $C$ denote the KV-cache capacity and $X_{\max}(k, N)$ the peak instantaneous memory footprint of the unconstrained execution. The problem is formulated as follows:

$$\max_{k,N} \quad \text{TP}(k, N) := \frac{k}{\mathbb{E}[T_d(k; N)] + \alpha_p + \beta_p k \mu_L}, \quad (1)$$

$$\text{s.t.} \quad k \in \{1, \dots, N\},$$

$$\Pr(X_{\max}(k, N) > C) \leq \epsilon, \quad (2)$$

**Approximate Solution Approach.** We adopt a *decoupled approximation* that decomposes the joint optimization into two sequential steps. First, we derive the optimal normalized threshold $\theta^* = k^*/N$ in an asymptotic regime as $N \to \infty$, where $k^*$ is the optimal solution to (1) without the memory constraint (2) for a fixed $N$. We show that $\theta^*$ converges to a limit $\theta_0$ that depends exclusively on the output distribution and is independent of $N$ (Section 3.2). Second, we fix $k = \lfloor \theta_0 N \rfloor$ and determine the maximum batch size $N^*$ that satisfies (2) (Section 3.3).

While this decoupled method solves an approximation to the original problem (1)—meaning the threshold $\theta_0$ is optimal asymptotically and may not perfectly coincide with the exact finite-$N$ solution—it provides significant analytical advantages. Specifically, the approximation yields tractable closed-form expressions, revealing important structural insights into LLM inference scheduling and enabling efficient online adaptation. We empirically validate its near-optimal performance in Section 4.2.

Furthermore, to derive a closed-form expression for the expected duration of a decode phase, $\mathbb{E}[T_d(k; N)]$, we apply *fluid approximation*. Here, the stochastic decode completion process is approximated by its deterministic limit as the system scale approaches infinity. This approximation is highly accurate in the large-batch regime typical of modern LLM serving.

### 3.2. Optimal Switching Threshold

Intuitively, balancing the amortization of the fixed prefill cost against the waste of idle decode slots depends fundamentally on how likely additional completions are in the near future. Thus, the *hazard rate* $h(t) = f(t)/\bar{F}(t)$ of the output-length distribution plays a key role: a higher hazard rate means completions arrive faster (favoring delayed switching), while a lower rate makes waiting costly (favoring earlier switching).

We first analyze the constant failure rate (CFR) case, which admits a closed-form solution, and then extend the analysis to the increasing failure rate (IFR) case, which better captures real LLM workloads where longer-running requests become progressively more likely to complete.

#### 3.2.1. BASE THRESHOLD UNDER CFR

Assume that the decode length distribution exhibits a constant hazard rate $h(t) = p_0$ (i.e., geometric output lengths), implying $\mu_O = 1/p_0$. Under the fluid approximation, the scaled number of decode requests decreases according to the differential equation $\dot{n}(t) = -p_0\, n(t)$ during a decode phase, yielding the expected duration:

$$\mathbb{E}[T_d(k; N)] = [\beta_d N\theta - \alpha_d \ln(1 - \theta)]\, p_0^{-1},$$

where $\theta = \frac{k}{N}$. Substituting $\mathbb{E}[T_d(k; N)]$ into (1), we obtain the throughput under EB:

$$\mathrm{TP}_{\mathrm{EB}}(k, N) = \left[ \frac{\alpha_p - \alpha_d\mu_O \ln(1-\theta)}{k} + \beta_{\mathrm{EB}}^w(\mu_L + \mu_O) \right]^{-1},$$

where $\beta_{\mathrm{EB}}^w = \frac{\beta_p\mu_L + \beta_d\mu_O}{\mu_L + \mu_O}$ represents the workload-weighted average of the prefill and decode marginal costs. Solving $k^* = \arg\max_k \mathrm{TP}_{\mathrm{EB}}(k, N)$ yields the optimal switching threshold for the unconstrained problem (1). The following proposition establishes the limiting behavior of $k^*$ as the batch size $N \to \infty$.

**Proposition 3.1** (Base Threshold under CFR)**.** *Under a constant hazard rate $h(t) = p_0$ and the fluid approximation of the decode phase duration, the limiting optimal normalized threshold $\theta_0 := \lim_{N\to\infty} \theta^* = \lim_{N\to\infty} k^*/N$ is the unique solution to*

$$\theta_0(1 - \theta_0)^{-1} + \ln(1 - \theta_0) = p_0\alpha_p\alpha_d^{-1}, \qquad (3)$$

*which depends only on the single ratio $p_0\alpha_p/\alpha_d$. In particular, it is independent of $N$, $\mu_L$, and the per-token costs $\beta_p, \beta_d$. We define $\zeta \triangleq -\ln(1 - \theta_0) > 0$ for brevity in what follows. (Proof in Appendix B.)*

This proposition implies that $\theta^*$ can be accurately approximated by $\theta_0$ in the large-batch regime. To facilitate solving the capacity-constrained problem in the second step, we use the approximated value $k_0^* := \lfloor \theta_0 N \rfloor$ as our practical solution to the unconstrained first-step problem. The simplicity of this parameter dependence makes the threshold highly practical to deploy: it is computed once from the easily measured tuple $(\alpha_p, \alpha_d, p_0)$.

The optimal throughput for the first-step problem is then given by:

$$\mathrm{TP}_{\mathrm{EB}}(k_0^*, N) = \left[ \frac{\alpha_p + \alpha_d\zeta\mu_O}{k_0^*} + \beta_{\mathrm{EB}}^w(\mu_L + \mu_O) \right]^{-1}, \tag{4}$$

#### 3.2.2. EXTENSION TO IFR

Real LLM workloads exhibit IFR: as a request generates more tokens, it becomes progressively more likely to complete in the near future. Within a decode phase, this accelerating completion rate reduces the additional wall-clock cost of waiting for one more idle slot. Consequently, the opportunity cost of delaying the phase switch is lower than under CFR, allowing the system to profitably accumulate more completions before switching. In short, IFR workloads support *higher* optimal switching thresholds than the CFR baseline.

Because the strength of IFR varies across workloads, the exact optimal limit $\theta^*$ can differ substantially from the base CFR limit $\theta_0$, motivating an analytical correction. We model IFR using a linear hazard rate $h(t) = p_0 + \eta t$ with $\eta > 0$.

**Theorem 3.2** (IFR Threshold Correction). *Under a linear hazard rate $h(t) = p_0 + \eta t$ with $\eta > 0$, the optimal threshold admits the expansion $\theta^* = \theta_0 + \Delta\theta + O(\eta^2)$, where $\theta_0$ is the CFR base threshold from Proposition 3.1 and*

$$\Delta\theta = \frac{\eta(1-\theta_0)^2}{p_0^2\,\theta_0}\left[\underbrace{\zeta\left(\frac{\theta_0}{1-\theta_0} - \frac{\zeta}{2}\right)}_{\text{duration effect}} + \underbrace{\frac{\beta_d N}{\alpha_d}(\zeta - \theta_0)}_{\text{per-token cost effect}}\right],$$
(5)

*with $\zeta = -\ln(1-\theta_0)$. The correction satisfies $\Delta\theta > 0$ for all $\eta > 0$. (Proof in Appendix C; empirical validation in Appendix E.3.)*

The IFR correction reveals two structural features absent in the base case. First, unlike the $N$-independent $\theta_0$, $\Delta\theta$ depends explicitly on $\rho = \beta_d N/\alpha_d$, so IFR effects amplify for large batches with high per-token overhead ($\rho \gg 1$). Second, the prefactor $\eta/p_0^2$ automatically rescales the correction across workloads with varying degrees of IFR.

### 3.3. Memory-Constrained Batch Sizing

Given $\theta_0$ and setting $k = k_0^* = \lfloor\theta_0 N\rfloor$, we now determine the maximum batch size $N^*$ such that the memory constraint (2) is satisfied. Memory evolves dynamically during decode—increasing as tokens are generated and dropping abruptly upon request completion—creating a sawtooth pattern whose peak determines feasibility.

**Proposition 3.3** (Memory-Safe Batch Size). *Under the CFR model with threshold $\theta_0$ (i.e., $k = \lfloor\theta_0 N\rfloor$), the maximum batch size satisfying $\Pr(X_{\max}(k, N) > C) \le \epsilon$ is*

$$N^* = \left\lfloor \frac{C - \ln(1/\epsilon)/(p_0^2\mu_L)}{\mu_L + \frac{1-\theta_0}{\theta_0\,p_0}\ln\frac{1}{1-\theta_0}} \right\rfloor.$$
(6)

*(Proof in Appendix D.)*

Here, we utilize the geometric model for analytical tractability. The solution (6) also serves as a conservative bound for IFR scenarios, as IFR workloads yield lower peak memory due to more predictable completion patterns.

### 3.4. Online Adaptive Algorithm

We design an online controller that jointly adapts $(\hat{k}^*, \hat{N}^*)$ by estimating workload parameters $(\hat{p}_0, \hat{\eta}, \hat{\mu}_L)$ from recent requests and evaluating the closed-form expressions of Sections 3.2–3.3 at the current estimates. Here, the hat notation $(\hat{\cdot})$ denotes the empirically estimated values of unknown parameters.

**Online estimation.** The system maintains sliding windows of the most recent output lengths ($\mathcal{W}_O$) and input lengths ($\mathcal{W}_L$). From $\mathcal{W}_O$, we estimate the empirical hazard rate at step $t$ as the fraction of completions at exactly $t$ among those still active at $t$: $\hat{h}(t) = \#\{O \in \mathcal{W}_O : O=t\}/\#\{O \in$

$\mathcal{W}_O : O \ge t\}$, and fit $\hat{h}(t) = \hat{p}_0 + \hat{\eta}t$ via weighted least squares over $t \in [1, t_{95}]$, where $t_{95}$ is the 95th percentile of recent output lengths. From $\mathcal{W}_L$, we estimate $\hat{\mu}_L$ via the sample mean.

**Threshold and batch-size update.** Given $(\hat{p}_0, \hat{\eta})$, we compute $\hat{\theta}_0$ by solving (3) and apply the IFR correction (5) to obtain $\hat{\theta}^* = \hat{\theta}_0 + \widehat{\Delta\theta}$ (clipped to a practical range $[\theta_{\min}, \theta_{\max}]$). Because $\widehat{\Delta\theta}$ depends on $N$ through $\rho = \beta_d N/\alpha_d$, we evaluate (5) using the previous cycle's $\hat{N}^*$, yielding a one-step fixed-point update that we empirically observe to converge within a few cycles. We then compute a memory-safe batch size by applying Proposition 3.3 with $\theta = \hat{\theta}^*$ and $(p_0, \mu_L) = (\hat{p}_0, \hat{\mu}_L)$, yielding $\hat{N}^* = N^*(\hat{\theta}^*, \hat{p}_0, \hat{\mu}_L; C, \epsilon)$. Finally, we set $\hat{k}^* = \lfloor\hat{\theta}^*\hat{N}^*\rfloor$ and apply the updated tuple $(\hat{k}^*, \hat{N}^*)$. Independent of this periodic update cycle, a runtime KV-aware gate (Appendix D.6) monitors memory and defers the decode→prefill transition whenever instantaneous KV occupancy leaves insufficient headroom. This absorbs any random memory excursions that $\hat{N}^*$ may underestimate under high-$\sigma_O$ workloads. Pseudo-code is given in Algorithm 1 (Appendix D.7).

### 3.5. Adaptive Mode Selection: EB⁺

Having derived (4) to approximate the throughput of EB($\hat{k}^*$), we now formulate the throughput of MB. By comparing the two, we deduce the condition (referred to as the *crossover condition*) under which one strategy outperforms the other. Building on this analytical foundation, we then design **EB⁺**, a hybrid scheduler that adaptively switches between EB and MB at runtime to globally maximize efficiency.

**MB Versus EB.** We first approximate the throughput of MB in steady-state. Let $N_p$ be the average number of requests finishing prefill per iteration. By Little's Law, the average number of tokens decoded in each iteration is $N_p\mu_O$. Because the MB discipline generates exactly one decoded token per request per iteration, the average number of decoding requests in a batch is directly $N_d = N_p\mu_O$. Consequently, the average effective batch size per iteration is $N = N_p + N_d = N_p(1 + \mu_O)$.

The system throughput under MB is thus formulated as:

$$\begin{aligned}\text{TP}_{\text{MB}}(N) &= N_p\left[\alpha_{\text{MB}} + \beta_{\text{MB}}^e(N_p\mu_L + N_d)\right]^{-1} \\ &= \left[\alpha_{\text{MB}}(1 + \mu_O)N^{-1} + \beta_{\text{MB}}^e(\mu_L + \mu_O)\right]^{-1},\end{aligned}$$

where $\alpha_{\text{MB}}$ is the fixed overhead for launching a mixed batch, and $\beta_{\text{MB}}^e$ is the empirically measured effective per-token marginal cost.

When standardizing $N$ to be both the maximum batch size for EB and the effective average batch size for MB, we can directly compare their throughputs.

**Proposition 3.4.** *Under the steady-state approximations*

*above,* $\text{TP}_{\text{MB}}(N) > \text{TP}_{\text{EB}}(k_0^*, N)$ *if and only if*

$$\beta_{\text{MB}}^e - \beta_{\text{EB}}^w < \frac{1}{\mu_L + \mu_O} \left[ \frac{\alpha_p + \alpha_d \zeta \mu_O}{k_0^*} - \frac{\alpha_{\text{MB}}(1+\mu_O)}{N} \right]. \tag{7}$$

In Proposition 3.4, the LHS represents the *marginal-cost gap* incurred by co-locating prefill and decode tokens in the same forward pass. The RHS captures the *amortized fixed-cost advantage* of MB, which requires fewer distinct kernel launches. Substituting $k_0^* = \theta_0 N$, both terms inside the bracket of the RHS scale as $1/N$. Therefore, the RHS is $O(1/N)$ and naturally vanishes at high utilization. In saturated regimes, the comparison is entirely governed by the marginal-cost gap $\beta_{\text{MB}}^e - \beta_{\text{EB}}^w$. On bandwidth-constrained GPUs, prefill–decode interference drastically inflates this gap. Figure 2 confirms this empirically: on the H200, the mixed-batch marginal cost remains low across most decode ratios, whereas on the RTX PRO 6000, bandwidth constraints inflate the cost above the threshold over intermediate ratios, heavily favoring EB.

**Online switching criterion.** At runtime, EB⁺ instantiates (7) using online system estimates and the EMA-smoothed active batch occupancy $N_{\text{obs}}$. To avoid oversensitivity to the noisy $\hat{\eta}$ estimate, we substitute $\hat{k}_0^* = \theta_0 N_{\text{obs}}$. (Because the RHS scales as $1/N_{\text{obs}}$ and the gap $\hat{\theta}^* - \hat{\theta}_0 = O(\eta)$ is bounded, the resulting estimation bias is $O(1/N_{\text{obs}})$ and vanishes under saturation.) By reversing the inequality from Proposition 3.4, we formally define the EB⁺ control rule—the system executes an exclusive batch (EB) when:

$$\hat{\beta}_{\text{MB}}^e(\hat{r}) - \beta_{\text{EB}}^w > \frac{(\alpha_p + \alpha_d \hat{\zeta}\hat{\mu}_O)/\hat{\theta}_0 - \alpha_{\text{MB}}(1+\hat{\mu}_O)}{N_{\text{obs}}(\hat{\mu}_L + \hat{\mu}_O)} + \delta, \tag{8}$$

and defaults to MB otherwise. Here, the parameter $\hat{\beta}_{\text{MB}}^e(\hat{r})$ at the estimated decode ratio $\hat{r} = \hat{\mu}_O/(\hat{\mu}_L + \hat{\mu}_O)$ is retrieved from a one-shot, per-hardware kernel-time profile (see the calibration procedure in Appendix A.4). The scalar $\delta$ serves as a tunable priority margin: setting $\delta > 0$ lowers the threshold for MB (favoring lower TTFT), while $\delta < 0$ biases the system toward EB (favoring higher token throughput).

**Traffic- and workload-awareness.** Equation (8) ensures adaptation along two distinct axes. First, the RHS scales inversely with the observed occupancy $N_{\text{obs}}$: under light traffic loads, the RHS is large, causing EB⁺ to safely default to MB. As traffic load increases and $N_{\text{obs}}$ grows, the RHS shrinks; once resource contention dominates, the system dynamically shifts to EB. Independently, $\hat{\beta}_{\text{MB}}^e(\hat{r})$ continuously tracks workload drift via the shifting decode ratio. Through these mechanisms, EB⁺ cleanly subsumes both EB($\hat{k}^*$) and MB as theoretical endpoints, seamlessly recovering the superior scheduling discipline regime-by-regime without the need for manual intervention.

## 4. Evaluation

### 4.1. Experimental Setup

**Hardware.** We evaluate on four GPU platforms (Table 1); H200 and RTX PRO 6000 serve as primary high-bandwidth and bandwidth-constrained baselines, with B300 and L40S used for scalability.

*Table 1.* Hardware specifications of evaluated GPUs.

| GPU | BW (TB/s) | FP16 (TFLOPS) | FLOP/B |
|-----|-----------|---------------|--------|
| B300 | 8.0 | 4,500 | 562 |
| H200 | 4.8 | 1,979 | 412 |
| RTX PRO 6000 | 1.792 | 252 | 141 |
| L40S | 0.864 | 362.5 | 419 |

**Implementation.** We implement EB($\hat{k}^*$) by extending vLLM (Kwon et al., 2023) with exclusive batching and online ($\hat{k}^*, \hat{N}^*$) updates, leaving PagedAttention unchanged. We compare three strategies—v0, v1, and EB($\hat{k}^*$)—on the same vLLM build in BF16.

**Models.** Our primary models are Qwen3-8B and Qwen3-30B-A3B (MoE) (Yang et al., 2025), with Gemma-3-1B-IT (Gemma Team, 2025) used for supplementary results.

**Scheduling strategies.** We compare three strategies. **v0**: the vLLM v0 scheduler, an exclusive-batching variant that switches to prefill whenever a slot opens; under our saturated regime (batches always full at $N$) it reduces to EB($k$=1). **v1**: the vLLM v1 scheduler, implementing mixed batching with decode-maximal allocation. **EB($\hat{k}^*$)**: exclusive batching with our online controller from Section 3.4, which adapts the phase-switching threshold $\hat{k}^*$ and the memory-safe batch size $\hat{N}^*$ from estimated workload statistics.

**Workloads.** We evaluate both synthetic and real-world workloads to span the prefill–decode mix. *Synthetic workloads* fix the mean input and output token counts, with per-request lengths drawn uniformly between $0.5\times$ and $1.5\times$ the corresponding mean: Decode-heavy (128 input / 1024 output), Balanced (512/512), and Prefill-heavy (1024/128). *Real-world workloads*: ShareGPT (ShareGPT, 2023) (conversational; 105 input / ≤500 output), LongBench (Bai et al., 2024) (long-context; 1,000–4,000 / 20), NuminaMath (Li et al., 2024) (chain-of-thought reasoning; 122 / 800–4,000), and WildChat (Zhao et al., 2024) (multi-turn dialogue; ≥6 turns). Output lengths are truncated during serving; detailed statistics in Appendix E.1.

**Metrics and traffic.** We report throughput in tokens/s (prefill and decode) and requests/s (RPS), and latency as TTFT (s), TPOT (ms), and ITL (s). Unless otherwise noted, experiments operate in a saturated regime (concurrency $c$=2,048) with 4,000 requests per configuration (WildChat: 3,000 multi-turn, ∼27,900 requests).

## 4.2. Model Validation

We validate the analytical predictions from Section 3 on H200 with Qwen3-8B and synthetic geometric workloads.

**Baseline threshold properties.** We test the structural properties of the CFR baseline threshold $\theta_0$ from Proposition 3.1: (P1) scale invariance in $N$, (P2) insensitivity to mean input length $\mu_L$, and (P3) monotone decrease in mean output length $\mu_O$. All three are empirically confirmed (Appendix E.2).

**Validation of decoupled optimization.** We validate the accuracy of our closed-form, online adaptive controller on H200 using synthetic geometric output workloads. We compare (i) exhaustive sweeps over fixed switching thresholds $k$ at a fixed batch size $N = 1024$ (vLLM default), and (ii) our online controller from Section 3.4, which estimates $(p_0, \eta)$ online and updates $(\hat{k}^*, \hat{N}^*)$ accordingly.

Figure 3 plots throughput and time per output token (TPOT) as functions of $k$ for three workloads. The closed-form choice achieves throughput comparable to, and in our experiments higher than, the best fixed $k$ in the sweep, improving throughput by up to **8.0%** (decode-heavy), **3.6%** (balanced), and **0.6%** (prefill-heavy) over the best fixed-$k$ point at $N = 1024$. The gain is consistent with EB($\hat{k}^*$) co-adapting both $\hat{\theta}^*$ and $\hat{N}^*$, while the sweep is constrained to a single batch size. TPOT remains competitive: it is close to the best-throughput setting in the decode-heavy case and is substantially lower in the balanced and prefill-heavy cases. Overall, these results support that the decoupled, closed-form selection eliminates the need for threshold search while retaining near-optimal performance.

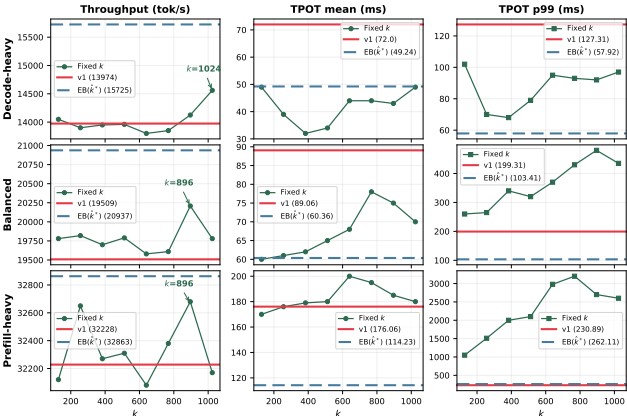

*Figure 3.* Validation on H200 ($N$=1024, geometric outputs): EB($\hat{k}^*$) matches the best fixed-$k$ sweep without manual tuning.

## 4.3. End-to-End Performance

We evaluate on RTX PRO 6000 and H200, sweeping the token budget $B \in \{4096, \ldots, 18432\}$ and max batch size $N \in \{256, \ldots, 2048\}$ via grid search and reporting the best-

throughput configuration per scheduler (per-workload optima in Appendix F.1.2). For EB($\hat{k}^*$), the adaptive controller computes a memory-safe batch size $\hat{N}^*$ online (Proposition 3.3); the effective batch size is $\min(\hat{N}^*, N)$.

### 4.3.1. SYNTHETIC WORKLOADS

We first evaluate on synthetic workloads with controlled input/output length ratios to isolate how workload composition affects scheduling performance.

Figure 4 compares v0, v1, and EB($\hat{k}^*$) across three input/output regimes on RTX PRO 6000 and H200. v1 consistently reduces ITL relative to v0 via phase overlapping, but often sacrifices throughput under extreme imbalance (Sun et al., 2024). EB($\hat{k}^*$) improves throughput across all regimes on RTX PRO 6000 with competitive latency; relative to the best of v0/v1, its gains peak in balanced and decode-heavy workloads (v0 already excels on prefill-heavy, narrowing the margin there). On H200 the throughput gap to v0/v1 narrows substantially. This confirms EB is particularly effective on bandwidth-constrained GPUs, where avoiding prefill–decode contention yields clearer throughput benefits.

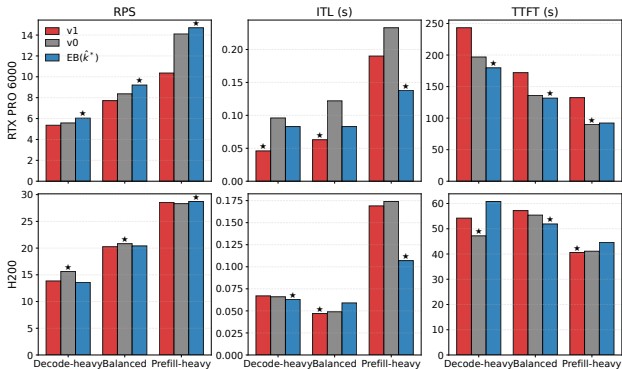

*Figure 4.* Synthetic workloads (Qwen3-8B): end-to-end performance on RTX PRO 6000 and H200. Stars mark the best strategy per workload/metric.

### 4.3.2. REAL-WORLD WORKLOADS

We further evaluate on four real-world workloads that represent diverse serving scenarios. Table 2 summarizes results for both Qwen models on RTX PRO 6000 and H200; Gemma-3-1B-IT results are deferred to Appendix F.2.

On the bandwidth-constrained RTX PRO 6000, EB($\hat{k}^*$) outperforms v1 on every workload for Qwen3-8B (average **+7.9%**); on Qwen3-30B-A3B the gain shrinks to **+1.4%** on average, with two workloads slightly negative (see analysis below). On the high-bandwidth H200, the performance gap narrows considerably and v1 matches or exceeds EB($\hat{k}^*$) on several configurations: consistent with our analysis, abundant memory bandwidth mitigates prefill–decode contention and erodes EB's advantage. These results corroborate our

central hypothesis that the EB–MB crossover is governed by memory bandwidth; we analyze the per-workload spread under the **Decode-ratio sweet spot** below.

*Table 2.* Throughput (RPS) on real-world workloads. % Diff. denotes the relative improvement of EB($\hat{k}^*$) over v1, computed as $(\text{EB}(\hat{k}^*) - \text{v1})/\text{v1} \times 100\%$.

| | | | Qwen3-8B | | | | Qwen3-30B-A3B | | |
|---|---|---|---|---|---|---|---|---|---|
| | Workload | v0 | v1 | EB($\hat{k}^*$) | % Diff. | v0 | v1 | EB($\hat{k}^*$) | % Diff. |
| RTX 6000 | ShareGPT | 15.45 | 17.07 | **19.68** | +15.3% | 12.38 | **15.98** | 15.48 | −3.1% |
| | LongBench | 8.17 | 8.35 | **8.66** | +3.7% | 11.48 | 11.13 | **11.67** | +4.8% |
| | WildChat | 10.39 | 12.75 | **14.19** | +11.3% | 6.91 | 9.39 | **10.48** | +11.6% |
| | Numina | 0.68 | 0.73 | **0.74** | +1.4% | 0.42 | **0.52** | 0.48 | −7.7% |
| | *Average* | | | | +7.9% | | | | +1.4% |
| H200 | ShareGPT | 19.36 | 41.93 | **42.88** | +2.3% | 17.68 | **48.96** | 47.72 | −2.5% |
| | LongBench | 14.22 | 15.77 | **15.81** | +0.3% | 21.31 | **24.43** | 24.03 | −1.6% |
| | WildChat | 17.58 | 20.87 | **21.50** | +3.0% | 17.59 | 26.46 | **26.66** | +0.8% |
| | Numina | 1.49 | 1.95 | **1.96** | +0.5% | 1.42 | **1.83** | 1.68 | −8.2% |
| | *Average* | | | | +1.5% | | | | −2.9% |

**Effect of model size.** Eq. (7) predicts that larger models, which incur higher per-iteration fixed costs $\alpha_p, \alpha_d, \alpha_{\text{MB}}$, inflating the RHS, make v1 (MB) more competitive. Table 2 aligns with this prediction: as we scale from Qwen3-8B to Qwen3-30B-A3B, EB's average gain over v1 drops from **7.9%** to **1.4%** on RTX PRO 6000, and from **1.5%** to **−2.9%** on H200. We attribute this diminishing gap to the $\alpha$ scaling implied by Eq. (7); the underlying interference-convexity analysis is in Appendix A.3.

**Decode-ratio sweet spot.** A convexity analysis (Appendix A.3) predicts EB's largest gains at intermediate decode ratios $r$, diminishing at the extremes ($r \to 0$ or $r \to 1$). Our workloads span this range: LongBench ($r \approx 0.004$), ShareGPT/WildChat ($r \approx 0.5$–$0.7$), and NuminaMath ($r > 0.85$). On RTX PRO 6000 with Qwen3-8B, EB's gains over v1 indeed peak at moderate $r$ (ShareGPT +15.3%, WildChat +11.3%) and shrink at the extremes (LongBench +3.7%, NuminaMath +1.4%); on Qwen3-30B-A3B, the high-$r$ extreme (NuminaMath, −7.7%) is consistent with the two effects compounding, while mid-$r$ workloads split between convexity's prediction (WildChat +11.6%) and $\alpha$-scaling (ShareGPT −3.1%); LongBench remains positive (+4.8%).

### 4.3.3. LATENCY ANALYSIS

**TTFT** (Fig. 5a). v1 generally achieves lower TTFT than EB($\hat{k}^*$) on most workloads by interleaving prefill with decode to reduce queuing delays. However, on prefill-heavy workloads (LongBench), EB($\hat{k}^*$) achieves lower TTFT across both GPUs (e.g., 169.64 s vs. 175.56 s on RTX PRO 6000 with Qwen3-8B), as EB avoids fine-grained chunk scheduling overhead.

**TPOT** (Fig. 5b). On RTX PRO 6000, EB($\hat{k}^*$) achieves substantially lower TPOT than v1 on most workloads: 65% reduction on ShareGPT (93.68 ms vs. 268.97 ms, Qwen3-

8B), 35% on WildChat (113.22 ms vs. 173.33 ms), and 20% on NuminaMath (65.22 ms vs. 81.75 ms). The exception is LongBench, where v1 achieves lower TPOT due to its prefill-heavy nature. On H200, the differences are smaller, though EB($\hat{k}^*$) remains competitive on decode-heavy workloads such as WildChat (79.04 ms vs. 122.97 ms) and NuminaMath (26.87 ms vs. 31.03 ms).

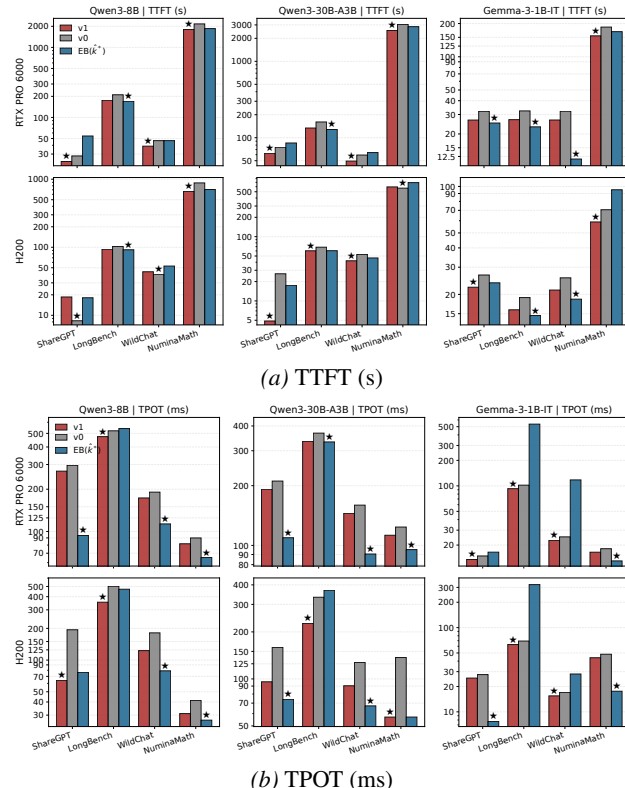

*(a)* TTFT (s)

*(b)* TPOT (ms)

*Figure 5.* TTFT (top) and TPOT (bottom) on real-world workloads.

These results reflect a fundamental trade-off: v1 favors TTFT through phase overlapping, while EB favors TPOT by avoiding bandwidth contention.

### 4.4. Adaptive Mode Selection: EB⁺

The results above commit to a single mode (v1 or EB($\hat{k}^*$)). We now evaluate the hybrid scheduler EB⁺ (Section 3.5), which switches online via the crossover criterion (8).

**Traffic-level sensitivity.** Table 3 (top) sweeps concurrency $c \in \{32, 512, 2048\}$. The two hardware columns realize different segments of the crossover. On the bandwidth-constrained RTX PRO 6000, EB($\hat{k}^*$) is TTFT-handicapped at low load ($\sim 1.5\times$ v1's TTFT), but EB⁺ selects MB at $c=32$ and recovers v1's TTFT to within 1 ms; at moderate load it improves throughput by +62% over v1 with $1.8\times$ lower TPOT; and at high load it commits to EB, attaining the best throughput (+50% over v1) with TTFT competitive to EB($\hat{k}^*$). On H200, where v1 is already strong, EB⁺

reproduces v1's throughput and TTFT at $c$=32, 512 via its MB choice and matches EB($\hat{k}^*$)'s throughput at $c$=2048.

**Non-stationary workloads.** We test EB⁺ under (i) a *distribution shift* ($\mu_L$: 1024→512→128, $\mu_O$: 128→512→1024; 2k requests/phase, $c = 2048$) and (ii) a *concurrency shift* ($c$: 32→512→1024→256→2048). EB⁺ attains throughput within ∼1% of the better of v1 and EB($\hat{k}^*$) in all four (hardware×scenario) cells (Table 3, bottom): distribution shift gives +36.4% over v1 on RTX PRO 6000 and +3.9% on H200, and concurrency shift gives +22.6% and +0.6% respectively. The EMA-smoothed $N_{obs}$ enables rapid adaptation without manual re-tuning.

*Table 3.* EB⁺ under stationary traffic (top three blocks; concurrency $c$, $\mu_L$=512, $\mu_O$=256) and non-stationary workloads (bottom two blocks). Qwen3-8B. TP in tok/s, TTFT in s, TPOT in ms.

| | | | RTX PRO 6000 | | | H200 | |
|---|---|---|---|---|---|---|---|
| Scenario | Metric | v1 | EB($\hat{k}^*$) | EB⁺ | v1 | EB($\hat{k}^*$) | EB⁺ |
| $c$=32 | TP | 4,928 | 4,728 | **4,930** | **11,584** | 10,779 | 11,586 |
| | TTFT | **0.061** | 0.090 | 0.061 | **0.041** | 0.062 | 0.042 |
| | TPOT | 19.4 | **16.5** | 19.5 | 8.2 | **7.1** | 8.2 |
| $c$=512 | TP | 8,330 | **13,549** | 13,510 | **27,464** | 27,207 | 27,460 |
| | TTFT | **1.2** | 8.3 | 5.1 | **0.7** | 4.6 | 0.7 |
| | TPOT | 164.2 | **77.6** | 90.4 | 52.7 | **36.8** | 52.7 |
| $c$=2048 | TP | 8,830 | 13,179 | **13,214** | 26,368 | **27,198** | 27,043 |
| | TTFT | 83.8 | 70.8 | **68.2** | **24.4** | 34.6 | 33.7 |
| | TPOT | 207.3 | **82.7** | 101.6 | 128.1 | **72.8** | 79.2 |
| Distrib. shift | TP | 7,025 | **9,647** | 9,582 | 20,396 | 20,762 | **21,182** |
| | TTFT | 190.7 | 157.5 | **152.1** | **41.2** | 66.5 | 69.7 |
| | TPOT | 178.5 | 105.0 | **102.2** | 101.0 | **42.3** | 63.1 |
| Concur. shift | TP | 8,364 | **10,350** | 10,250 | 24,251 | 23,374 | **24,397** |
| | TTFT | **24.0** | 29.8 | 26.6 | **7.7** | 13.7 | 9.4 |
| | TPOT | 142.4 | **51.6** | 77.5 | 77.9 | **45.6** | 66.8 |

**Goodput and PD-disaggregation.** At $c$=512 with SLO (TTFT< 10 s, TPOT< 100 ms), EB⁺ attains 80.3% joint goodput versus 77.3% (EB($\hat{k}^*$)) and 5.8% (v1). On 4× RTX PRO 6000 it is best or near-best in 7 of 9 workload×$c$ cells against vLLM's PD-disaggregation scheduler, and matches or exceeds v1 on the 2-GPU setup—without P:D tuning or OOMs. See Appendices F.3, F.4.

### 4.5. Scalability

Due to the high cost of exhaustive evaluation, we assess scalability using the **WildChat** workload with the same grid search procedure as in Section 4.3.

#### 4.5.1. Across GPU Platforms

Table 4 extends the GPU comparison to L40S (bandwidth-constrained) and B300 (highest bandwidth). On L40S, EB($\hat{k}^*$) achieves **41.9%** higher RPS than v1 (14.70 vs. 10.36) with lower TTFT and TPOT. On B300, EB($\hat{k}^*$) is on par with v1 (52.34 vs. 50.47 RPS) with comparable TTFT and TPOT—consistent with the bandwidth-driven crossover

*Table 4.* Scalability across GPU platforms (Qwen3-8B, WildChat). RPS in requests/s; TTFT/TPOT in s and ms respectively.

| | v0 | | | v1 | | | EB($\hat{k}^*$) | | |
|---|---|---|---|---|---|---|---|---|---|
| GPU | RPS | TTFT | TPOT | RPS | TTFT | TPOT | RPS | TTFT | TPOT |
| B300* | – | – | – | 50.47 | 2.08 | 24.00 | 52.34 | 2.87 | 22.12 |
| L40S | 4.14 | 157.21 | 171.02 | 10.36 | 132.47 | 189.70 | 14.70 | 92.16 | 138.16 |

*v0 does not support PyTorch 2.9 and CUDA 13.0, which are required for B300.

established in Section 4.3.

#### 4.5.2. Across Model Architectures

On RTX PRO 6000 with WildChat, Figure 6 evaluates generalization across four model families: Llama-3.1-8B-Instruct (Grattafiori et al., 2024), Mathstral-7B-v0.1 (Mistral AI, 2024), Qwen2.5-Coder-7B (Hui et al., 2024), and DeepSeek-R1-Distill-Qwen-7B (Guo et al., 2025). EB($\hat{k}^*$) consistently achieves the highest throughput and lowest TPOT across all models. Compared to v1, EB($\hat{k}^*$) improves RPS by 8–17% and reduces TPOT by 17–47%, with the largest gains on Llama-3.1-8B (47% TPOT reduction). On this bandwidth-constrained GPU, EB dominates v1 across all tested architectures.

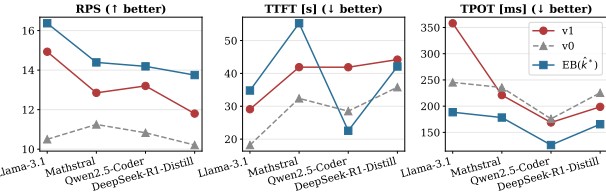

*Figure 6.* Scalability across models on RTX PRO 6000.

#### 4.5.3. Robustness to Configuration Parameters

A configuration sweep over $(B, N)$ further shows that all three schedulers exhibit comparable robustness, so EB($\hat{k}^*$) imposes no additional tuning burden (Appendix F.1).

## 5. Conclusion

This paper shows that neither mixed batching (MB) nor exclusive batching (EB) universally dominates LLM serving—the optimal choice is governed by a closed-form crossover condition driven by GPU bandwidth, model size, and workload composition. Our adaptive EB scheduler, with asymptotically optimal phase-switching thresholds and memory-safe batch sizing, achieves up to 41.9% throughput improvement on bandwidth-constrained GPUs. The hybrid EB⁺ scheduler instantiates this crossover condition online, matching or exceeding both baselines across stationary and non-stationary regimes without manual tuning, and rivaling disaggregated serving without extra hardware. Future directions include tighter SLO-aware switching and joint scheduling with sequence/pipeline parallelism.

## Acknowledgments

The authors sincerely thank the anonymous reviewers for their insightful comments and constructive suggestions, which significantly improved the quality of this work. This research was supported in part by the Hong Kong Research Grants Council under Grants 15508021 and 15511424.

## Impact Statement

This paper advances the field of machine learning by improving the efficiency of LLM inference scheduling. Our work is a systems-level optimization: it makes existing models cheaper to serve rather than enabling new capabilities, and the scheduling strategies are agnostic to generated content. A positive consequence is that EB's gains on bandwidth-constrained accelerators broaden the range of hardware on which LLMs can be deployed efficiently, lowering the cost of access. We do not foresee negative societal consequences beyond those well-established for LLM deployment.

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

# A. Cost Model: Empirical Validation and Extensions

## A.1. Empirical Verification of the Linear Iteration-Time Model

Our theoretical derivations (Propositions 3.1, 3.3 and Theorem 3.2) build on the linear iteration-time model $T_p(k) = \alpha_p + \beta_p \sum_i L_i$ in Section 3.1. Although self-attention has theoretically quadratic complexity in context length, we show empirically that the quadratic term is negligible in the regime where our scheduler actually operates.

**Scope: the budget that matters is the per-iteration token budget.** Both MB (vLLM v1) and our EB scheduler operate under *chunked prefilling*: a long input is split across multiple iterations, with each iteration processing at most $B = \texttt{max\_num\_batched\_tokens}$ tokens (typically 2K–8K, up to 16K on high-end GPUs). The linear model is therefore applied *per iteration* on at most $B$ tokens; it does not need to hold over an entire long input. All experiments in Section 4.3 respect this budget.

**Per-iteration validation within the practical regime.** We profile prefill iteration time on RTX PRO 6000 Blackwell (96 GB) across seven models spanning four architectures and hidden dimensions $d \in [2048, 5120]$: Qwen3-{4B, 8B, 14B, 30B-A3B (MoE)}, Llama-{3.2-1B, 3.1-8B}, and Mistral-Nemo-12B. For each model we fit $T = \alpha + \beta\, n_{\text{tok}}$ to measured prefill times and report $R^2$ within three budget ranges (Table 5). Within the practical range $B \leq 8\text{K}$, every model achieves $R^2 > 0.986$; extending to $B \leq 16\text{K}$, $R^2$ stays $> 0.979$. As a cross-GPU check, repeating the protocol with Qwen3-4B on H200 also yields $R^2 = 0.998$, with the quadratic correction improving $R^2$ by only $0.001$.

*Table 5.* Linear-model $R^2$ within practical token-budget ranges on RTX PRO 6000 Blackwell.

| $B$ range | Qwen3-4B ($d$=2560) | Qwen3-8B ($d$=4096) | Qwen3-14B ($d$=5120) | Qwen3-30B-A3B ($d$=2048) | Llama-3.2-1B ($d$=2048) | Llama-3.1-8B ($d$=4096) | Mistral-Nemo-12B ($d$=5120) |
|---|---|---|---|---|---|---|---|
| $\leq$ 4K | 0.9993 | **0.9999** | 0.9998 | 0.9939 | 0.9993 | 0.9996 | 0.9988 |
| $\leq$ 8K | 0.9957 | **0.9994** | 0.9977 | 0.9863 | 0.9971 | **0.9996** | 0.9990 |
| $\leq$ 16K | 0.9870 | 0.9907 | **0.9960** | 0.9790 | 0.9913 | 0.9918 | 0.9930 |

The robustness of the linear fit is consistent with the FLOP decomposition of prefill: GEMM dominates, contributing $24d^2$ FLOPs per token versus $4Ld$ for attention. Even at $L = 4096$ with $d = 2560$, the attention fraction is $L/(6d) \approx 27\%$, and GQA further reduces this share, so the quadratic-in-$L$ component is a small fraction of overall prefill cost in the operating regime of interest.

**End-to-end serving at 128K context.** To verify that per-iteration linearity translates to serving-level gains at long context, we run end-to-end benchmarks at 128K input / 64 output tokens on the same GPU under vLLM's default $B = 8192$. Each request's prefill is chunked into $\sim$16 iterations, all within the validated linear regime ($R^2 > 0.99$ at $B \leq 8\text{K}$). Concurrency is capped at the maximum value that avoids OOM on a single 96 GB GPU, since the KV cache for a 128K-token request is large. Even at this low-concurrency regime—where MB's scheduling overhead is amortized and prefill–decode contention is minimal—EB($\hat{k}^*$) achieves $+1.4\%$ to $+4.0\%$ higher throughput than v1 (MB) across all seven models (Table 6), confirming that exclusive batching provides consistent gains at long context.

*Table 6.* End-to-end throughput (tok/s) at 128K input / 64 output tokens on RTX PRO 6000 (96 GB). Concurrency is the maximum that fits in GPU memory.

| Model | Concurrency | v1 (MB) | EB($\hat{k}^*$) | $\Delta$ |
|---|---|---|---|---|
| Llama-3.2-1B | 4 | 26,664 | **27,265** | $+2.3\%$ |
| Qwen3-30B-A3B | 4 | 4,826 | **5,021** | $+4.0\%$ |
| Qwen3-4B | 4 | 6,134 | **6,314** | $+2.9\%$ |
| Qwen3-8B | 3 | 5,621 | **5,764** | $+2.5\%$ |
| Llama-3.1-8B | 3 | 6,331 | **6,482** | $+2.4\%$ |
| Qwen3-14B | 3 | 3,830 | **3,883** | $+1.4\%$ |
| Mistral-Nemo-12B | 3 | 4,774 | **4,877** | $+2.2\%$ |

**Stress test beyond practical settings.** For completeness, we additionally probe how the linear model behaves outside its intended operating regime by forcing $B = 131{,}072$ so that a single iteration processes the entire 128K input without chunking. This far exceeds typical deployment configurations, yet even here the linear fit retains $R^2 \geq 0.943$ across all seven models (Table 7), indicating that the approximation degrades gracefully rather than breaking down. Figure 7 provides

a visual summary spanning both regimes: the left panel shows the measured prefill time curves, and the right panel traces how $R^2$ degrades smoothly as the fit range is extended from the practical budget to the full 128K range.

Table 7. Linear-model $R^2$ under the stress setting $B = 131{,}072$ (one iteration on the full 128K input).

| $B$ range | Qwen3-4B | Qwen3-8B | Qwen3-14B | Qwen3-30B-A3B | Llama-3.2-1B | Llama-3.1-8B | Mistral-Nemo-12B |
|---|---|---|---|---|---|---|---|
| $\leq$ 32K | 0.9731 | 0.9852 | **0.9906** | 0.9675 | 0.9762 | 0.9872 | **0.9899** |
| $\leq$ 64K | 0.9597 | 0.9753 | **0.9831** | 0.9549 | 0.9625 | 0.9777 | **0.9829** |
| $\leq$ 128K | 0.9471 | 0.9607 | **0.9664** | 0.9434 | 0.9481 | 0.9631 | **0.9669** |

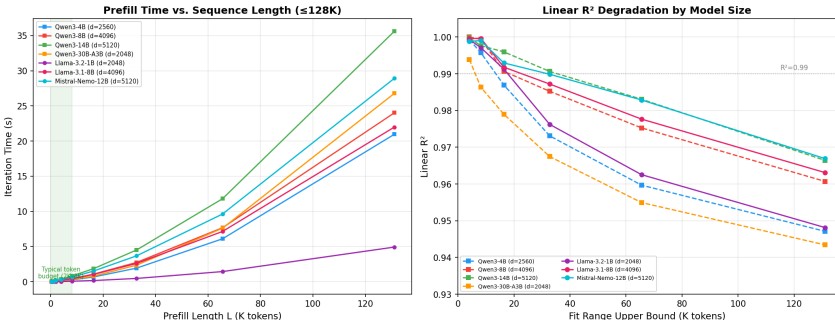

Figure 7. Linear-model validation across seven models on RTX PRO 6000 Blackwell. **(Left)** Measured prefill iteration time vs. context length $L$ (up to 128K tokens); the shaded band marks the typical per-iteration token budget ($\leq$ 8K). **(Right)** Linear-fit $R^2$ as a function of the fit-range upper bound. Within the practical budget ($\leq$ 16K) all models retain $R^2 > 0.97$; even when stretched to 128K the approximation degrades gracefully, staying $R^2 \geq 0.94$.

## A.2. Additional Marginal Cost and Kernel Breakdown Results

Section 1 presents marginal cost and kernel breakdown analyses using Qwen3-4B. Here we extend these to Gemma-3-1B-IT, Qwen3-8B, and Qwen3-30B-A3B (a Mixture-of-Experts variant) on the same two-GPU setup (RTX PRO 6000 and H200), examining how model scale and architecture affect the prefill–decode interference phenomenon.

**Marginal cost.** Figure 8 reports execution time vs. total token count at various decode ratios for each model on both GPUs. For the two Qwen models, RTX PRO 6000 results are consistent with the Qwen3-4B findings in the main text: mixed-batch execution exceeds pure decode at moderate decode ratios. On H200, the mixed-batch curves remain between the pure-prefill and pure-decode baselines across most operating points, with crossover only at high decode ratios. The interference effect intensifies with model size—Qwen3-30B-A3B exhibits a wider gap between mixed-batch and pure-decode costs than Qwen3-8B on RTX PRO 6000, reflecting increased memory-bandwidth pressure from larger (or more numerous) weight matrices.

For Gemma-3-1B-IT, however, the mixed-batch curves remain strictly between the pure-prefill and pure-decode baselines on both GPUs, and no crossover is observed. We attribute this to its lightweight architecture: a 1B-parameter dense model has fewer weight matrices and attention heads, so even the RTX PRO 6000's 1.792 TB/s bandwidth is sufficient to serve mixed batches without severe contention. This indicates that the interference effect is *model-architecture-dependent*—it emerges when the model's memory footprint is large enough to saturate the available bandwidth.

**Kernel breakdown.** Figure 9 ((a)–(c)) decomposes the iteration time into kernel-level components. We highlight three observations:

*(i) GEMM time is insensitive to batch composition.* Across all three models and both GPUs, GEMM time remains largely invariant to decode ratio, confirming that the linear projection cost depends primarily on total token count rather than the prefill–decode composition.

*(ii) Attention is the primary source of composition sensitivity for Qwen models.* For both Qwen3-8B and Qwen3-30B-A3B, as well as Qwen3-4B in the main text, the Attention component accounts for the majority of the latency increase as decode ratio grows, and exhibits the same GPU-dependent crossover behavior: mixed-batch Attention becomes slower than pure-decode Attention at a much lower decode ratio on the RTX PRO 6000 than on the H200.

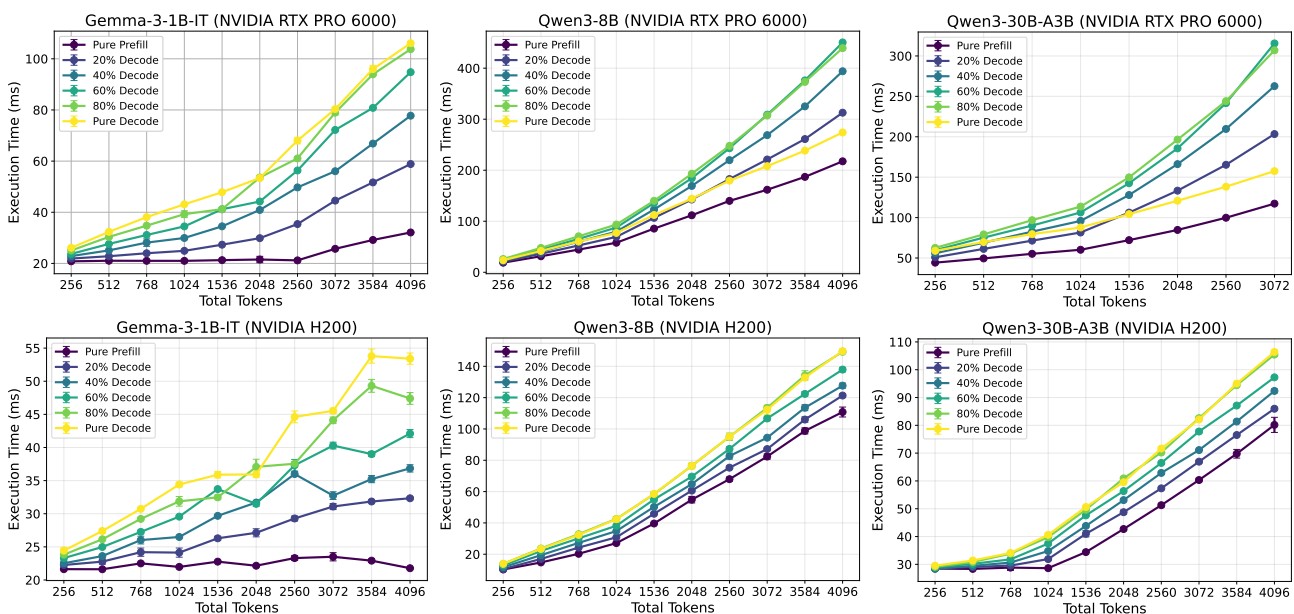

*Figure 8.* Execution time vs. total token count at various decode ratios for Gemma-3-1B-IT (left), Qwen3-8B (middle), and Qwen3-30B-A3B (right) on RTX PRO 6000 (top) and H200 (bottom).

*(iii) Model-specific kernel characteristics.* For Qwen3-30B-A3B, MoE routing and expert computation dominate kernel time but remain insensitive to batch composition, behaving similarly to GEMM. For Gemma-3-1B-IT, the "Other" category (elementwise operations, normalization, memory management) grows noticeably with decode ratio on RTX PRO 6000: GEMM and Attention complete quickly for a 1B-parameter model, so per-token overheads (KV-cache management, kernel launch latency, synchronization)—which scale with decode tokens—become proportionally significant.

Taken together, these results show that the prefill–decode interference identified in Section 1 is robust across the Qwen model family at different scales, including MoE architectures. The absence of interference for Gemma-3-1B-IT further supports our bandwidth-based explanation: the effect manifests when the model's memory footprint is sufficient to create bandwidth contention, making it most relevant for the medium-to-large models that dominate production deployments.

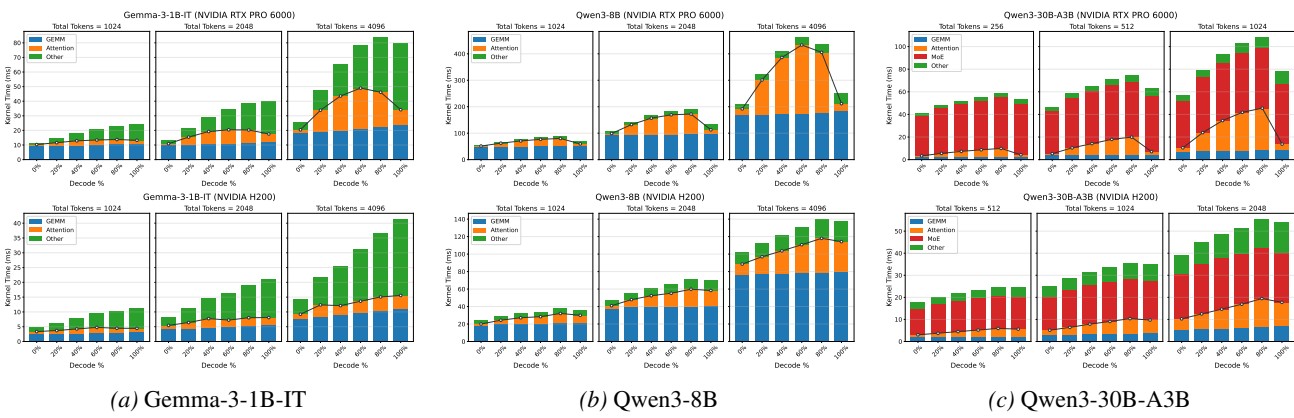

*(a)* Gemma-3-1B-IT        *(b)* Qwen3-8B        *(c)* Qwen3-30B-A3B

*Figure 9.* Kernel time breakdown by decode ratio on RTX PRO 6000 (top) and H200 (bottom) for (a) Gemma-3-1B-IT, (b) Qwen3-8B, and (c) Qwen3-30B-A3B.

**Effect of decode context length.** Figures 1 and 2 in the main text use ctx= 16. Figure 10 extends those results by varying ctx to examine how KV cache length affects iteration time for Qwen3-4B on H200 with a fixed total batch size of 1024 tokens. Here, *decode context length* (ctx) denotes the number of tokens already accumulated in each decode request's KV cache: at ctx= $c$, every decode token must stream $c$ key–value pairs from GPU memory during attention.

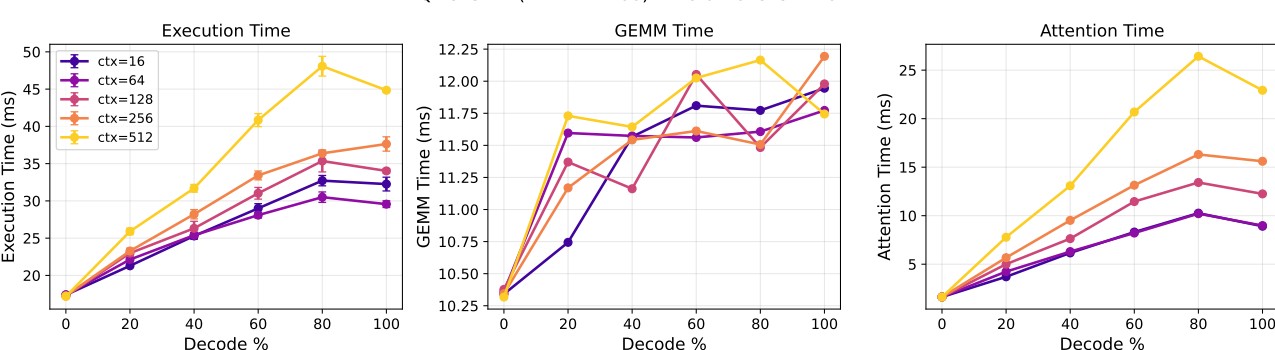

*Figure 10.* Effect of decode context length on (left) iteration time, (middle) GEMM time, and (right) attention time for Qwen3-4B on NVIDIA H200 (total batch = 1024 tokens).

Three patterns emerge. First, Execution time (left) increases with ctx at all decode ratios, with the five curves ordered by ctx and diverging as decode percentage grows; at 0% decode (pure prefill) all curves converge because no KV cache is accessed. Second, GEMM time (middle) is nearly invariant to ctx—the curves remain tightly clustered across the full decode range—confirming that linear-projection cost depends only on the number of tokens, not on KV cache size. Third, attention time (right) is the sole driver of context-length sensitivity: it grows monotonically with ctx and accounts for almost all of the vertical spread visible in the execution-time panel, with ctx$= 512$ reaching $\sim 26$ ms at 80% decode versus $\sim 10$ ms for ctx$= 16$.

Taken together, increasing ctx uniformly lifts the execution-time and attention-time curves without changing their shape or crossover structure. This validates the claim in Section 1 that the qualitative conclusions of Figures 1–2 generalize across context lengths.

### A.3. Convexity Analysis of Mixed-Batch Cost

This appendix derives the *interference convexity index* $\kappa$ referenced in Section 3.2 and shows that two qualitative predictions used in the evaluation—(i) more negative $\kappa$ favors EB, and (ii) EB's advantage peaks at intermediate decode ratios—follow as direct algebraic consequences of the crossover condition (7).

**Setup.** Let $\mathcal{K}(r)$ denote the per-iteration kernel time of a mixed batch of fixed total token count, expressed as a function of the decode ratio $r \in [0, 1]$. Let

$$\bar{\mathcal{K}}(r) \;=\; (1 - r)\,\mathcal{K}(0) + r\,\mathcal{K}(1)$$

be the linear interpolation between pure-prefill ($r = 0$) and pure-decode ($r = 1$) costs. The marginal-cost gap on the LHS of (7) satisfies

$$\beta_{\mathrm{MB}}^{e}(r) - \beta_{\mathrm{EB}}^{w}(r) \;\propto\; \mathcal{K}(r) - \bar{\mathcal{K}}(r), \tag{9}$$

because $\beta_{\mathrm{MB}}^{e}(r)$ is the per-token cost of the mixed batch at ratio $r$ (proportional to $\mathcal{K}(r)$), while $\beta_{\mathrm{EB}}^{w}(r)$ is the workload-weighted average of the exclusive prefill and decode per-token costs (proportional to $\bar{\mathcal{K}}(r)$).

**Quadratic fit and $\kappa$.** Empirically, $\mathcal{K}(r)$ is well approximated by a quadratic $\mathcal{K}(r) = c_0 + c_1 r + c_2 r^2$. Direct calculation gives

$$\mathcal{K}(r) - \bar{\mathcal{K}}(r) \;=\; c_2 r^2 - c_2 r \;=\; -c_2\, r(1 - r). \tag{10}$$

The deviation $\mathcal{K}(r) - \bar{\mathcal{K}}(r)$ is therefore controlled by the single coefficient $c_2$ (the second derivative $\mathcal{K}''/2$). To obtain a dimensionless, hardware-comparable scalar, we normalize $c_2$ by the pure-decode cost $\mathcal{K}(1) = c_0 + c_1 + c_2$ and define the *interference convexity index*

$$\kappa \;=\; \frac{2c_2}{c_0 + c_1 + c_2}. \tag{11}$$

**Consequences.** Combining (9), (10), and (11) (rearranged as $c_2 = \kappa \mathcal{K}(1)/2$) yields

$$\beta_{\mathrm{MB}}^{e} - \beta_{\mathrm{EB}}^{w} \;\propto\; -\frac{\kappa\, \mathcal{K}(1)}{2}\, r(1 - r),$$

from which two predictions follow:

*(i) More negative $\kappa$ favors EB.* Since $\mathcal{K}(1) > 0$ and $r(1-r) \geq 0$ on $[0,1]$, a more negative $\kappa$ enlarges the LHS of (7), so the inequality fails at smaller mismatches and EB outperforms MB more easily. Empirically, $\kappa$ is negative on most GPUs (the kernel-time curve is concave); its magnitude varies substantially across hardware. On Qwen3-4B with 4096 tokens we measure $\kappa = -11.6$ on the RTX PRO 6000 versus $\kappa = -0.69$ on the H200, confirming that bandwidth-constrained hardware exhibits substantially stronger prefill–decode interference.

*(ii) EB's advantage peaks at intermediate decode ratios.* The factor $r(1-r)$ in (10) is maximized at $r = 1/2$ and vanishes at the extremes $r \in \{0,1\}$. Consequently, even on hardware with strongly negative $\kappa$, EB yields its largest gains over MB on workloads operating at moderate decode ratios; prefill-heavy ($r \to 0$) and decode-heavy ($r \to 1$) workloads see diminishing returns. This prediction is verified empirically in Section 4.3, where on RTX PRO 6000 with Qwen3-8B, EB($\hat{k}^*$) improves over v1 by +15.3% on ShareGPT and +11.3% on WildChat (both $r \approx 0.5$–$0.7$) but only +3.7% on LongBench ($r \approx 0.004$) and +1.4% on NuminaMath ($r > 0.85$).

## A.4. Empirical Calibration of $\beta_{\mathrm{MB}}^e(r)$ for EB$^+$

EB$^+$'s online switching rule in Eq. (8) requires runtime evaluation of $\beta_{\mathrm{MB}}^e(\hat{r})$, the mixed-batch per-token cost as a function of decode ratio. Unlike $\beta_{\mathrm{EB}}^w$, which derives analytically from the prefill/decode kernel coefficients $(\alpha_p, \beta_p, \alpha_d, \beta_d)$ already calibrated for EB($\hat{k}^*$), $\beta_{\mathrm{MB}}^e(r)$ captures prefill$\leftrightarrow$decode interference inside a mixed batch and must be measured empirically.

When the interference convexity $\kappa$ has been profiled (Appendix A.3), the coefficients $(c_0, c_1, c_2)$ of $\beta_{\mathrm{MB}}^e(r) = c_0 + c_1 r + c_2 r^2$ follow directly from $c_2 = \kappa\,\mathcal{K}(1)/2$. When $\kappa$ is not available, $(c_0, c_1, c_2)$ can be obtained from a small set of synthetic v1 runs at varying decode ratios: fit $1/\mathrm{throughput}_{v1}(r)$ to a quadratic in $r$. On H200 with Qwen3-8B, fitting v1 grid data at $r \in \{0.11, 0.50, 0.89\}$ yields coefficients in the $10^{-5}$ s/tok range. The priority margin $\delta$ in Eq. (8) should match the observed |LHS − RHS| magnitude. For the calibration above, typical |LHS − RHS| at $\hat{N}_{\mathrm{obs}} \in [200, 2000]$ is in the $10^{-6}$–$10^{-5}$ range; we set $\delta = 10^{-5}$, yielding stable MB$\leftrightarrow$EB switching across workload drift (verified on non-stationary workloads in Table 3, bottom). With this calibration, EB$^+$ on Qwen3-8B distribution-shift correctly switches MB$\to$EB during decode-rich phases and back to MB as concurrency drops near experiment end.

## B. Proof of Proposition 3.1: Baseline Threshold under CFR

We derive the optimal switching threshold under constant failure rate (CFR), where the hazard rate is $h(t) = p_0$. This corresponds to geometric output length distributions, with mean output length $\mu_O = 1/p_0$.

### B.1. Stochastic Model and Fluid Approximation

**Exact stochastic dynamics.** Let $N_t$ denote the number of requests still decoding after $t$ iterations. The process $\{N_t\}_{t \geq 0}$ is a pure death process: at each iteration, each of the $N_t$ active requests independently completes with probability $p_0$. Thus,

$$N_{t+1} = N_t - D_t, \quad D_t \mid N_t \sim \mathrm{Binomial}(N_t, p_0),$$

where $D_t$ is the number of completions in iteration $t$.

**Fluid approximation.** For large batch sizes, we approximate the discrete stochastic process by a continuous deterministic flow. Define the scaled process $\bar{N}_t^{(N)} = N_t/N$. By the law of large numbers for density-dependent population processes (Ethier & Kurtz, 1986; Kurtz, 1970), as $N \to \infty$, the scaled process converges uniformly in probability to the solution of the ordinary differential equation:

$$\frac{dn}{dt} = -p_0 \cdot n(t), \quad n(0) = 1, \tag{12}$$

where $t$ denotes the *iteration index* (not wall-clock time), consistent with the hazard-rate argument used in the main text. Specifically, for any fixed $T > 0$,

$$\sup_{0 \leq t \leq T} \left| \bar{N}_t^{(N)} - n(t) \right| \xrightarrow{P} 0 \quad \text{as } N \to \infty.$$

The approximation error is $O(1/\sqrt{N})$, arising from the central limit theorem for the cumulative martingale differences.

The ODE (12) has solution:

$$n(t) = e^{-p_0 t}.$$

The exact discrete trajectory is $(1 - p_0)^t$; the continuous-time fluid limit $e^{-p_0 t}$ used here introduces an additional $O(p_0)$ relative error (the gap between $-p_0$ and $\ln(1 - p_0)$), enumerated as Source 4 in the error analysis below.

## B.2. Decode Phase Duration

A phase switch occurs when $k = \theta N$ requests have completed, i.e., when the batch size drops to $N(1 - \theta)$. Under the fluid approximation, this occurs at iteration index $t^*$ satisfying $n(t^*) = 1 - \theta$:

$$e^{-p_0 t^*} = 1 - \theta \quad \Longrightarrow \quad t^* = \frac{-\ln(1 - \theta)}{p_0} = \frac{\zeta(\theta)}{p_0},$$

where we define $\zeta(\theta) \triangleq -\ln(1 - \theta)$ for notational convenience.

**Wall-clock decode time.** The wall-clock time for the decode phase is the sum of iteration times. Each iteration with batch size $n$ takes time $t_d(n) = \alpha_d + \beta_d n$, where $\alpha_d$ is the fixed overhead and $\beta_d$ is the per-request cost. Under the fluid approximation, the expected decode time is:

$$\mathbb{E}[T_d] = \int_0^{t^*} t_d(N \cdot n(t)) \, dt = \int_0^{t^*} \left( \alpha_d + \beta_d N e^{-p_0 t} \right) dt. \tag{13}$$

Evaluating the integral:

$$\mathbb{E}[T_d] = \alpha_d t^* + \beta_d N \int_0^{t^*} e^{-p_0 t} \, dt$$

$$= \alpha_d \cdot \frac{\zeta}{p_0} + \beta_d N \cdot \frac{1 - e^{-p_0 t^*}}{p_0}$$

$$= \frac{\alpha_d \zeta}{p_0} + \frac{\beta_d N \theta}{p_0},$$

where the last equality uses $1 - e^{-p_0 t^*} = \theta$.

*Remark* B.1 (Interpretation of the integral). The integral (13) sums the wall-clock time over iterations. The variable $t$ is the iteration index, and $dt = 1$ represents one iteration. This is the fluid-limit analog of the discrete sum $\sum_{i=1}^{M} t_d(N_i)$, where $M$ is the (random) number of iterations until $k$ completions.

## B.3. Approximation Error Analysis

The fluid approximation introduces several sources of error relative to the exact stochastic model. We characterize each below and argue that they do not materially affect the optimal threshold.

**Source 1: Stochastic fluctuation.** The fluid model replaces the random completions $D_t \sim \text{Binomial}(N_t, p_0)$ with their conditional expectation $p_0 N_t$. By Kurtz's theorem (Kurtz, 1970), the scaled batch size $N_t/N$ converges uniformly to the ODE solution $n(t) = e^{-p_0 t}$, with fluctuations of order $O(1/\sqrt{N})$. Since practical LLM serving systems operate at batch sizes $N \geq 64$, this contributes at most a few percent relative error to the decode time estimate.

**Source 2: Discretization and stopping-time mismatch.** The fluid model assumes the batch size decreases continuously and stops exactly when $k$ requests have completed. In the actual discrete system, this cannot happen for two reasons: (i) iterations are discrete, so the system can only stop at integer iteration indices $M = \lceil t^* \rceil$, not at the continuous stopping point $t^*$; and (ii) multiple requests may complete within a single iteration, causing the cumulative completion count to *overshoot* the threshold $k$ rather than hitting it exactly. Both effects share the same root cause—the discrete system cannot stop at an arbitrary point—and produce a bounded relative error that does not vanish with $N$. The bias is more pronounced at small $\theta$, because the batch size at the switching point is still large ($\approx N(1-\theta)$), leading to larger per-iteration completion variance and thus larger overshoot relative to $k$.

**Source 3: Integer rounding.** The continuous threshold $\theta N$ is rounded to $k = \lfloor \theta N \rfloor$ in practice. This contributes an $O(1/N)$ relative error and is negligible for $N \geq 32$.

**Source 4: Continuous-time fluid limit.** The ODE $\dot{n} = -p_0 n$ has solution $n(t) = e^{-p_0 t}$, whereas the exact discrete trajectory is $(1 - p_0)^t = e^{t \ln(1-p_0)}$. The gap $-p_0 - \ln(1 - p_0) = -p_0^2/2 + O(p_0^3)$ contributes an additional $O(p_0)$ relative error to $\mathbb{E}[T_d]$ and to $\theta_0$. This source is independent of $N$, deterministic, and would be absorbed exactly by reparametrizing $p_0 \to -\ln(1 - p_0)$ in (15). For our operating regime $p_0 \in [0.003, 0.01]$, the numerical impact on $\theta_0$ is below 1% and is dominated by the $O(1/\sqrt{N})$ stochastic fluctuation (Source 1).

*Table 8.* Summary of approximation error sources.

| Source | Origin | Relative error | Vanishes as $N \to \infty$? |
|---|---|---|---|
| Stochastic fluctuation | Binomial $\to$ expectation | $O(1/\sqrt{N})$ | Yes |
| Discretization & overshoot | Discrete stopping | Bounded | No |
| Integer rounding | $\lfloor \theta N \rfloor$ | $O(1/N)$ | Yes |
| Continuous-time fluid limit | $e^{-p_0 t}$ vs $(1 - p_0)^t$ | $O(p_0)$ | Yes (as $p_0 \to 0$) |

### B.4. Expected Prefill Time

For the prefill phase, each of the $k = \theta N$ completed requests generates a new request. The prefill time is:

$$\mathbb{E}[T_p] = \alpha_p + \beta_p \cdot (\theta N) \cdot \mu_L = \alpha_p + \beta_p N \mu_L \theta,$$

where $\mu_L$ is the mean input length.

### B.5. Complete Cycle Time

The total cycle time combines decode and prefill phases:

$$\mathcal{T}(\theta) = \mathbb{E}[T_d] + \mathbb{E}[T_p] = \frac{\alpha_d \zeta}{p_0} + \frac{\beta_d N \theta}{p_0} + \alpha_p + \beta_p N \mu_L \theta.$$

Grouping terms:

$$\mathcal{T}(\theta) = \frac{\alpha_d \zeta(\theta)}{p_0} + \alpha_p + \left( \frac{\beta_d N}{p_0} + \beta_p N \mu_L \right) \theta.$$

### B.6. Throughput Optimization

The normalized throughput is:

$$R(\theta) = \frac{\theta}{\mathcal{T}(\theta)}.$$

To maximize $R$, we take the derivative and set it to zero:

$$\frac{dR}{d\theta} = \frac{\mathcal{T}(\theta) - \theta \mathcal{T}'(\theta)}{\mathcal{T}(\theta)^2} = 0.$$

This yields the first-order optimality condition:

$$\mathcal{T}(\theta) = \theta \mathcal{T}'(\theta). \tag{14}$$

**Computing $\mathcal{T}'(\theta)$.** Since $\zeta(\theta) = -\ln(1 - \theta)$, we have $\zeta'(\theta) = \frac{1}{1-\theta}$. Thus:

$$\mathcal{T}'(\theta) = \frac{\alpha_d}{p_0(1 - \theta)} + \frac{\beta_d N}{p_0} + \beta_p N \mu_L.$$

**Applying the optimality condition.** Substituting into (14):

$$\frac{\alpha_d \zeta}{p_0} + \frac{\beta_d N \theta}{p_0} + \alpha_p + \beta_p N \mu_L \theta = \theta \left[ \frac{\alpha_d}{p_0(1 - \theta)} + \frac{\beta_d N}{p_0} + \beta_p N \mu_L \right].$$

Expanding the right-hand side:

$$\frac{\alpha_d \zeta}{p_0} + \frac{\beta_d N \theta}{p_0} + \alpha_p + \beta_p N \mu_L \theta = \frac{\alpha_d \theta}{p_0(1-\theta)} + \frac{\beta_d N \theta}{p_0} + \beta_p N \mu_L \theta.$$

The terms $\frac{\beta_d N \theta}{p_0}$ and $\beta_p N \mu_L \theta$ appear on both sides and cancel:

$$\frac{\alpha_d \zeta}{p_0} + \alpha_p = \frac{\alpha_d \theta}{p_0(1-\theta)}.$$

Multiplying both sides by $\frac{p_0}{\alpha_d}$:

$$\zeta + \frac{p_0 \alpha_p}{\alpha_d} = \frac{\theta}{1-\theta}.$$

Rearranging and substituting $\zeta = -\ln(1-\theta)$:

$$\boxed{\frac{\theta_0}{1-\theta_0} + \ln(1-\theta_0) = \frac{p_0 \alpha_p}{\alpha_d}} \tag{15}$$

## B.7. Existence and Uniqueness

**Lemma B.2.** *Equation* (15) *has a unique solution* $\theta_0 \in (0, 1)$ *for any* $\gamma = p_0 \alpha_p / \alpha_d > 0$.

*Proof.* Define $\Psi(\theta) = \frac{\theta}{1-\theta} + \ln(1-\theta)$.

**Step 1: Monotonicity.** Computing the derivative:

$$\Psi'(\theta) = \frac{(1-\theta) - \theta \cdot (-1)}{(1-\theta)^2} - \frac{1}{1-\theta} = \frac{1}{(1-\theta)^2} - \frac{1}{1-\theta} = \frac{1 - (1-\theta)}{(1-\theta)^2} = \frac{\theta}{(1-\theta)^2}.$$

Since $\theta > 0$ and $(1-\theta)^2 > 0$ for $\theta \in (0, 1)$, we have $\Psi'(\theta) > 0$, so $\Psi$ is strictly increasing.

**Step 2: Boundary behavior.**

- As $\theta \to 0^+$: $\frac{\theta}{1-\theta} \to 0$ and $\ln(1-\theta) \to 0$, so $\lim_{\theta \to 0^+} \Psi(\theta) = 0$.
- As $\theta \to 1^-$: $\frac{\theta}{1-\theta} \to +\infty$ and $\ln(1-\theta) \to -\infty$, but the first term dominates (it grows like $1/(1-\theta)$ while the second decreases like $\ln(1-\theta)$), so $\lim_{\theta \to 1^-} \Psi(\theta) = +\infty$.

**Step 3: Existence and uniqueness.** Since $\Psi$ is continuous and strictly increasing on $(0, 1)$ with $\Psi(0^+) = 0$ and $\Psi(1^-) = +\infty$, by the intermediate value theorem, for any $\gamma > 0$, there exists a unique $\theta_0 \in (0, 1)$ such that $\Psi(\theta_0) = \gamma$. $\square$

## B.8. Parameter Independence

**Corollary B.3.** *The optimal threshold* $\theta_0$ *depends only on the ratio* $p_0 \alpha_p / \alpha_d$ *and is independent of* $\beta_d$, $\beta_p$, $N$, *and* $\mu_L$.

*Proof.* This follows directly from the cancellation observed in the derivation: the terms involving $\beta_d$, $\beta_p$, $N$, and $\mu_L$ appear linearly in $\theta$ on both sides of the optimality condition and cancel exactly.

Intuitively, this occurs because:

1. The per-token decode cost $\beta_d N \theta / p_0$ depends on the *number* of requests that complete, which equals $\theta N$ regardless of *when* they complete.
2. The prefill cost $\beta_p N \mu_L \theta$ similarly depends only on the total tokens prefilled, not the timing.

Thus, the marginal cost-benefit trade-off at the optimal threshold depends only on the timing-related parameters $(\alpha_d, \alpha_p, p_0)$. $\square$

### B.9. Second-Order Condition

To confirm $\theta_0$ is a maximum (not a minimum), we verify the second-order condition.

**Lemma B.4.** *The critical point $\theta_0$ satisfying* (15) *is a global maximum of $R(\theta)$.*

*Proof.* Define $\Phi(\theta) = \mathcal{T}(\theta) - \theta\mathcal{T}'(\theta)$. Then:

$$\Phi'(\theta) = \mathcal{T}'(\theta) - \mathcal{T}'(\theta) - \theta\mathcal{T}''(\theta) = -\theta\mathcal{T}''(\theta).$$

Since $\mathcal{T}''(\theta) = \frac{\alpha_d}{p_0(1-\theta)^2} > 0$, we have $\Phi'(\theta) < 0$ for $\theta > 0$, so $\Phi$ is strictly decreasing.

Checking boundary values:

- $\Phi(0^+) = \mathcal{T}(0) - 0 = \alpha_p > 0$.
- $\Phi(1^-) \to -\infty$ (since $\mathcal{T}'(\theta) \to +\infty$ as $\theta \to 1^-$).

Thus $\Phi$ has a unique zero at $\theta_0$, with $\Phi(\theta) > 0$ for $\theta < \theta_0$ and $\Phi(\theta) < 0$ for $\theta > \theta_0$. Since $\frac{dR}{d\theta} = \Phi(\theta)/\mathcal{T}(\theta)^2$, this confirms $R$ is increasing for $\theta < \theta_0$ and decreasing for $\theta > \theta_0$, so $\theta_0$ is the unique global maximum. $\square$

This completes the proof of Proposition 3.1. $\square$

## C. Proof of Theorem 3.2: IFR Threshold Correction

We now extend the analysis to the increasing failure rate (IFR) case with linear hazard $h(t) = p_0 + \eta t$, where $\eta \geq 0$ captures the degree of IFR behavior. The proof proceeds by: (1) deriving the modified batch dynamics, (2) computing cycle time as a perturbation expansion in $\eta$, and (3) applying the implicit function theorem to obtain the threshold correction.

**Scope of the analysis.** The argument below operates on the fluid (deterministic) model, treating the decode batch as if every request evolved under $h(t) = p_0 + \eta t$ measured from the start of the current phase. Under CFR (memoryless), this is exact because hazards are state-independent; under IFR, continuing requests carry heterogeneous ages across cycles, so successive cycles are no longer i.i.d. and the renewal-reward foundation does not apply directly. The fluid argument below does not require i.i.d. cycles; the finite-$N$ stochastic analog would require a mixing-time argument for the age-dependent Markov chain, which we leave to future work. Empirically, $\Delta\theta > 0$ is observed on both synthetic IFR workloads (Appendix E.3) and real LLM workloads (Appendix E.4).

### C.1. Batch Size Evolution under IFR

Under the fluid approximation with linear hazard rate, the batch size evolves according to (for algebraic convenience we use $n(t)$ here as the unnormalized batch size with $n(0) = N$, in contrast to the normalized convention $n(0) = 1$ adopted in Appendix B; the two are related by an overall factor of $N$):

$$\frac{dn}{dt} = -h(t) \cdot n(t) = -(p_0 + \eta t)\, n(t), \quad n(0) = N.$$

This is a separable ODE. Separating variables:

$$\frac{dn}{n} = -(p_0 + \eta t)\, dt.$$

Integrating both sides:

$$\ln n(t) - \ln N = -p_0 t - \frac{\eta t^2}{2}.$$

Exponentiating:

$$n(t) = N \exp\left(-p_0 t - \frac{\eta t^2}{2}\right). \tag{16}$$

**Verification.** At $\eta = 0$, this reduces to $n(t) = Ne^{-p_0 t}$, recovering the CFR solution.

## C.2. Decode Phase Duration

A phase switch occurs when $k = \theta N$ requests have completed, i.e., when $n(T) = N(1 - \theta)$. Substituting into (16):

$$N(1 - \theta) = N \exp\left(-p_0 T - \frac{\eta T^2}{2}\right).$$

Taking logarithms:

$$p_0 T + \frac{\eta T^2}{2} = -\ln(1 - \theta) = \zeta(\theta).$$

This is a quadratic equation in $T$. Solving for the positive root:

$$T(\theta) = \frac{-p_0 + \sqrt{p_0^2 + 2\eta\zeta(\theta)}}{\eta}.$$

**Perturbation expansion.** For small $\eta$, we expand $T$ to first order. Define $\xi = \eta\zeta/p_0^2$ (dimensionless). Then:

$$T = \frac{p_0}{\eta}\left(-1 + \sqrt{1 + 2\xi}\right) = \frac{p_0}{\eta}\left(-1 + 1 + \xi - \frac{\xi^2}{2} + O(\xi^3)\right) = \frac{p_0 \xi}{\eta}\left(1 - \frac{\xi}{2} + O(\xi^2)\right).$$

Substituting $\xi = \eta\zeta/p_0^2$:

$$T(\theta) = \frac{\zeta}{p_0} - \frac{\eta\zeta^2}{2p_0^3} + O(\eta^2) \triangleq T_0 + \eta T_1 + O(\eta^2), \tag{17}$$

where $T_0 = \zeta/p_0$ is the CFR decode time and $T_1 = -\zeta^2/(2p_0^3) < 0$ is the first-order correction (negative because IFR reduces decode time).

## C.3. Cumulative Batch Size

Define the cumulative batch size integral:

$$I(\theta) = \int_0^{T(\theta)} n(s)\, ds = \int_0^T N \exp\left(-p_0 s - \frac{\eta s^2}{2}\right) ds.$$

**Zeroth-order term.** At $\eta = 0$:

$$I_0 = N \int_0^{T_0} e^{-p_0 s} ds = \frac{N}{p_0}\left(1 - e^{-p_0 T_0}\right) = \frac{N}{p_0}\left(1 - e^{-\zeta}\right) = \frac{N}{p_0}(1 - (1 - \theta)) = \frac{N\theta}{p_0}.$$

**First-order correction.** To compute the $O(\eta)$ correction, we use:

$$I(\theta) = \int_0^{T_0 + \eta T_1} Ne^{-p_0 s} \cdot e^{-\eta s^2/2} ds + O(\eta^2).$$

Expanding $e^{-\eta s^2/2} \approx 1 - \eta s^2/2$ and accounting for the change in upper limit:

$$I = I_0 + \eta\left[Ne^{-p_0 T_0} T_1 - \frac{N}{2}\int_0^{T_0} s^2 e^{-p_0 s} ds\right] + O(\eta^2).$$

Computing the integral $\int_0^{T_0} s^2 e^{-p_0 s} ds$ by parts (twice) and simplifying:

$$I(\theta) = \frac{N\theta}{p_0} + \frac{\eta N}{p_0^3}\left[(1 - \theta)\zeta - \theta\right] + O(\eta^2). \tag{18}$$

**Verification.** The correction term $(1 - \theta)\zeta - \theta = (1 - \theta)(-\ln(1 - \theta)) - \theta$. For small $\theta$, $\zeta \approx \theta + \theta^2/2$, so the correction $\approx (1 - \theta)(\theta + \theta^2/2) - \theta = -\theta^2/2 + O(\theta^3) < 0$, confirming that IFR reduces cumulative batch size.

## C.4. Complete Cycle Time Expansion

The total cycle time is:

$$\mathcal{T}(\theta) = \mathbb{E}[T_d] + \mathbb{E}[T_p] = \alpha_d T + \beta_d I + \alpha_p + \beta_p N \mu_L \theta.$$

Substituting the expansions (17) and (18):

$$\mathcal{T}(\theta) = \alpha_d \left( \frac{\zeta}{p_0} - \frac{\eta \zeta^2}{2p_0^3} \right) + \beta_d \left( \frac{N\theta}{p_0} + \frac{\eta N}{p_0^3} \left[ (1-\theta)\zeta - \theta \right] \right) + \alpha_p + \beta_p N \mu_L \theta + O(\eta^2)$$

$$= \underbrace{\frac{\alpha_d \zeta}{p_0} + \frac{\beta_d N \theta}{p_0} + \alpha_p + \beta_p N \mu_L \theta}_{\mathcal{T}_0(\theta)} + \eta \underbrace{\left[ -\frac{\alpha_d \zeta^2}{2p_0^3} + \frac{\beta_d N}{p_0^3} \left( (1-\theta)\zeta - \theta \right) \right]}_{\mathcal{T}_1(\theta)} + O(\eta^2).$$

We write this as:

$$\mathcal{T}(\theta) = \mathcal{T}_0(\theta) + \eta \, \mathcal{T}_1(\theta) + O(\eta^2), \tag{19}$$

where $\mathcal{T}_0$ is the CFR cycle time and $\mathcal{T}_1(\theta)$ is the first-order IFR correction.

## C.5. Optimal Threshold via Implicit Function Theorem

**Setup.** Define the residual function (extending $\Phi$ from the CFR proof to depend on $\eta$):

$$\Phi(\theta, \eta) = \mathcal{T}(\theta, \eta) - \theta \cdot \frac{\partial \mathcal{T}}{\partial \theta}(\theta, \eta).$$

The optimal threshold $\theta^*(\eta)$ satisfies the first-order condition $\Phi(\theta^*, \eta) = 0$. At $\eta = 0$, we have $\theta^*(0) = \theta_0$, the CFR baseline from Proposition 3.1.

**Implicit differentiation.** By the implicit function theorem, if $\partial \Phi / \partial \theta \neq 0$ at $(\theta_0, 0)$, then:

$$\left. \frac{d\theta^*}{d\eta} \right|_{\eta=0} = - \left. \frac{\partial \Phi / \partial \eta}{\partial \Phi / \partial \theta} \right|_{(\theta_0, 0)}. \tag{20}$$

Thus the first-order correction is:

$$\Delta \theta = \theta^*(\eta) - \theta_0 = - \left. \frac{\partial \Phi / \partial \eta}{\partial \Phi / \partial \theta} \right|_{(\theta_0, 0)} \cdot \eta + O(\eta^2).$$

**Computing $\partial \Phi / \partial \theta$ at $\eta = 0$.**

From $\Phi = \mathcal{T} - \theta \mathcal{T}'$:

$$\frac{\partial \Phi}{\partial \theta} = \mathcal{T}' - \mathcal{T}' - \theta \mathcal{T}'' = -\theta \mathcal{T}''.$$

At $\eta = 0$:

$$\mathcal{T}_0''(\theta) = \frac{d}{d\theta} \left[ \frac{\alpha_d}{p_0(1-\theta)} + \frac{\beta_d N}{p_0} + \beta_p N \mu_L \right] = \frac{\alpha_d}{p_0(1-\theta)^2}.$$

Thus:

$$\left. \frac{\partial \Phi}{\partial \theta} \right|_{(\theta_0, 0)} = -\theta_0 \cdot \frac{\alpha_d}{p_0(1-\theta_0)^2} = -\frac{\theta_0 \alpha_d}{p_0(1-\theta_0)^2}.$$

**Computing $\partial \Phi / \partial \eta$ at $(\theta_0, 0)$.**

From $\mathcal{T} = \mathcal{T}_0 + \eta \mathcal{T}_1 + O(\eta^2)$:

$$\left. \frac{\partial \mathcal{T}}{\partial \eta} \right|_{\eta=0} = \mathcal{T}_1(\theta).$$

Similarly, $\mathcal{T}' = \mathcal{T}_0' + \eta\,\mathcal{T}_1' + O(\eta^2)$, so:

$$\frac{\partial}{\partial\eta}(\theta\mathcal{T}')\bigg|_{\eta=0} = \theta\,\mathcal{T}_1'(\theta).$$

Thus:

$$\frac{\partial\Phi}{\partial\eta}\bigg|_{(\theta_0,0)} = \mathcal{T}_1(\theta_0) - \theta_0\,\mathcal{T}_1'(\theta_0).$$

**Computing $\mathcal{T}_1$ and $\mathcal{T}_1'$.** From (19):

$$\mathcal{T}_1(\theta) = -\frac{\alpha_d\zeta^2}{2p_0^3} + \frac{\beta_d N}{p_0^3}\left[(1-\theta)\zeta - \theta\right].$$

Taking the derivative using $\zeta' = 1/(1-\theta)$:

$$\mathcal{T}_1'(\theta) = -\frac{\alpha_d\cdot 2\zeta\cdot\zeta'}{2p_0^3} + \frac{\beta_d N}{p_0^3}\left[-\zeta + (1-\theta)\zeta' - 1\right] = -\frac{\alpha_d\zeta}{p_0^3(1-\theta)} + \frac{\beta_d N}{p_0^3}\left[-\zeta + 1 - 1\right].$$

Simplifying:

$$\mathcal{T}_1'(\theta) = -\frac{\alpha_d\zeta}{p_0^3(1-\theta)} - \frac{\beta_d N\zeta}{p_0^3}.$$

Now compute $\mathcal{T}_1 - \theta\,\mathcal{T}_1'$:

$$
\begin{aligned}
\mathcal{T}_1(\theta_0) - \theta_0\,\mathcal{T}_1'(\theta_0) &= -\frac{\alpha_d\zeta^2}{2p_0^3} + \frac{\beta_d N}{p_0^3}[(1-\theta_0)\zeta - \theta_0] + \frac{\theta_0\alpha_d\zeta}{p_0^3(1-\theta_0)} + \frac{\theta_0\beta_d N\zeta}{p_0^3} \\
&= \frac{1}{p_0^3}\left[-\frac{\alpha_d\zeta^2}{2} + \frac{\theta_0\alpha_d\zeta}{1-\theta_0} + \beta_d N\left((1-\theta_0)\zeta - \theta_0 + \theta_0\zeta\right)\right] \\
&= \frac{1}{p_0^3}\left[-\frac{\alpha_d\zeta^2}{2} + \frac{\theta_0\alpha_d\zeta}{1-\theta_0} + \beta_d N(\zeta - \theta_0)\right].
\end{aligned}
$$

**Final expression.** Substituting into (20):

$$\Delta\theta = -\frac{\frac{1}{p_0^3}\left[-\frac{\alpha_d\zeta^2}{2} + \frac{\theta_0\alpha_d\zeta}{1-\theta_0} + \beta_d N(\zeta - \theta_0)\right]}{-\frac{\theta_0\alpha_d}{p_0(1-\theta_0)^2}}\cdot\eta.$$

Simplifying:

$$\Delta\theta = \frac{(1-\theta_0)^2}{p_0^2\theta_0\alpha_d}\left[-\frac{\alpha_d\zeta^2}{2} + \frac{\theta_0\alpha_d\zeta}{1-\theta_0} + \beta_d N(\zeta - \theta_0)\right]\cdot\eta.$$

Factoring out $\alpha_d$ from the first two terms:

$$\boxed{\Delta\theta = \frac{\eta(1-\theta_0)^2}{p_0^2\theta_0}\left[\zeta\left(\frac{\theta_0}{1-\theta_0} - \frac{\zeta}{2}\right) + \frac{\beta_d N}{\alpha_d}(\zeta - \theta_0)\right]} \tag{21}$$

This is Eq. (5) in the main text.

### C.6. Positivity of the Correction

**Proposition C.1.** $\Delta\theta > 0$ for all $\theta_0 \in (0, 1)$ and $\eta > 0$.

*Proof.* The prefactor $\frac{\eta(1-\theta_0)^2}{p_0^2\theta_0} > 0$ for all valid parameters.

For the bracketed term, we show each component is positive:

**Term 1:** $\zeta\left(\frac{\theta_0}{1-\theta_0} - \frac{\zeta}{2}\right)$.

Using the baseline condition (15), we have:

$$\frac{\theta_0}{1-\theta_0} = \gamma + \zeta, \quad \text{where } \gamma = \frac{p_0\alpha_p}{\alpha_d} > 0.$$

Thus:

$$\frac{\theta_0}{1-\theta_0} - \frac{\zeta}{2} = \gamma + \zeta - \frac{\zeta}{2} = \gamma + \frac{\zeta}{2} > 0.$$

Since $\zeta = -\ln(1-\theta_0) > 0$ for $\theta_0 \in (0,1)$:

$$\zeta\left(\frac{\theta_0}{1-\theta_0} - \frac{\zeta}{2}\right) = \zeta\left(\gamma + \frac{\zeta}{2}\right) > 0.$$

**Term 2:** $\frac{\beta_d N}{\alpha_d}(\zeta - \theta_0)$.

We need to show $\zeta > \theta_0$, i.e., $-\ln(1-\theta_0) > \theta_0$.

Define $q(x) = -\ln(1-x) - x$ for $x \in [0,1)$. Then:

- $q(0) = 0$.
- $q'(x) = \frac{1}{1-x} - 1 = \frac{x}{1-x} > 0$ for $x \in (0,1)$.

Since $q$ is strictly increasing with $q(0) = 0$, we have $q(\theta_0) > 0$ for all $\theta_0 \in (0,1)$, i.e., $\zeta > \theta_0$.

Since $\beta_d, N, \alpha_d > 0$, the second term is positive (or zero if $\beta_d = 0$).

**Conclusion.** Both terms are non-negative with the first strictly positive, hence $\Delta\theta > 0$. $\square$

### C.7. Physical Interpretation

The correction (21) decomposes into two distinct effects:

**Duration effect (first term).** Under IFR, the hazard rate increases over time, meaning later completions occur faster than earlier ones. This reduces the expected time to accumulate additional completions beyond the CFR prediction, making it beneficial to wait for more completions.

Mathematically, the term $\zeta\left(\frac{\theta_0}{1-\theta_0} - \frac{\zeta}{2}\right)$ captures the reduction in decode phase duration per additional completion achieved.

**Per-token cost effect (second term).** Under IFR, the batch size decays faster than exponential (the Gaussian factor $e^{-\eta t^2/2}$ accelerates the decay). This reduces the cumulative per-token cost $\beta_d \int n(s)ds$ for a given number of completions.

This benefit scales with $\rho = \beta_d N/\alpha_d$, the ratio of total per-token overhead to fixed overhead. For systems with large batches and significant per-token costs ($\rho \gg 1$), this effect dominates.

**Special case:** $\beta_d = 0$. When there is no per-token decode cost (pure fixed overhead), only the duration effect contributes:

$$\Delta\theta\big|_{\beta_d=0} = \frac{\eta(1-\theta_0)^2}{p_0^2\theta_0} \cdot \zeta\left(\gamma + \frac{\zeta}{2}\right).$$

### C.8. Second-Order Verification

**Lemma C.2.** *The corrected threshold $\theta^* = \theta_0 + \Delta\theta + O(\eta^2)$ remains a maximum of the throughput function for sufficiently small $\eta > 0$.*

*Proof.* The second-order condition requires $\partial^2 R/\partial\theta^2 < 0$ at the optimum.

At $\eta = 0$, we showed in Appendix B that $\theta_0$ is a strict local maximum with $\Phi'(\theta_0) = -\theta_0 \mathcal{T}_0''(\theta_0) < 0$.

By continuity, for sufficiently small $\eta$, the second derivative remains negative in a neighborhood of $\theta^*(\eta)$, confirming it remains a local maximum.

To verify it is the global maximum, observe that the boundary behavior ($R(0^+) = 0$ and $R(1^-)$ bounded) is preserved under small perturbations, ensuring the interior maximum is global. $\qquad\square$

This completes the proof of Theorem 3.2. $\qquad\square$

## D. Proof of Proposition 3.3: Memory-Safe Batch Size

The proof proceeds in four steps: (1) steady-state analysis of request ages, (2) expected initial memory, (3) variance of memory increments, and (4) supremum bound. We model the memory process as a random walk with i.i.d. increments and apply an infinite-horizon supremum bound—a standard heuristic-conservative approximation; a fully rigorous finite-horizon martingale concentration analysis is left to future work.

### D.1. Steady-State Request Age Distribution

Define the *age $A$* of a request as the number of decode phases it has participated in. In each cycle, $k$ out of $N$ requests complete and are replaced. At steady state, $A$ follows a geometric distribution:

$$\Pr(A = a) = \theta(1 - \theta)^{a-1}, \quad a = 1, 2, \ldots$$

with mean $\mathbb{E}[A] = 1/\theta$, where $\theta = k/N$.

At decode start, the batch contains $k$ new requests (age $A = 1$) and $(N - k)$ continuing requests (age $A \geq 2$). For continuing requests, the memoryless property gives:

$$\mathbb{E}[A - 1 \mid A \geq 2] = \mathbb{E}[A] = \frac{1}{\theta}.$$

### D.2. Expected Tokens Generated by Continuing Requests

Under the fluid approximation, starting with $N$ requests and waiting for $k$ completions, the expected decode phase duration (in iterations) is

$$t^* = \frac{1}{p_0} \ln \frac{1}{1 - \theta},$$

matching the CFR stopping time from Appendix B.

A continuing request has survived $(A - 1)$ previous phases, generating tokens in each. Thus:

$$\mathbb{E}[G \mid \text{continuing}] = \mathbb{E}[A - 1 \mid A \geq 2] \cdot t^* = \frac{1}{\theta} \cdot \frac{1}{p_0} \ln \frac{1}{1 - \theta} = \frac{1}{\theta p_0} \ln \frac{1}{1 - \theta}.$$

The expected initial memory is:

$$\mathbb{E}[X_0] = k \cdot \mu_L + (N - k) \cdot (\mu_L + \mathbb{E}[G]) = N\mu_L + \frac{N(1 - \theta)}{\theta p_0} \ln \frac{1}{1 - \theta}.$$

### D.3. Variance of Memory Increments

At each decode iteration with $n$ active requests, the memory change is:

$$\Delta X = n - \sum_{i=1}^{n} Z_i S_i,$$

where $Z_i \sim \text{Bernoulli}(p_0)$ indicates completion and $S_i = L_i + O_i$ is the total sequence length released upon completion. Let $J = \sum_{i=1}^{n} Z_i \sim \text{Binomial}(n, p_0)$.

**Expected change.** Since completed requests have $\mathbb{E}[S \mid \text{complete}] = \mu_L + 1/p_0$:

$$\mathbb{E}[\Delta X] = n - np_0(\mu_L + 1/p_0) = n - np_0\mu_L - n = -np_0\mu_L = d_n.$$

**Variance via law of total variance.** We compute $\text{Var}(\Delta X) = \mathbb{E}[\text{Var}(\Delta X \mid J)] + \text{Var}(\mathbb{E}[\Delta X \mid J])$.

For the first term: given $J = j$ completions, the released memory is $\sum_{i=1}^{j} S_i$ where $S_i$ are i.i.d. Thus $\text{Var}(\Delta X \mid J = j) = j \cdot \text{Var}(S)$, and:

$$\mathbb{E}[\text{Var}(\Delta X \mid J)] = \mathbb{E}[J] \cdot \text{Var}(S) = np_0 \cdot \text{Var}(S).$$

For the second term: $\mathbb{E}[\Delta X \mid J] = n - J \cdot \mathbb{E}[S]$, so:

$$\text{Var}(\mathbb{E}[\Delta X \mid J]) = \mathbb{E}[S]^2 \cdot \text{Var}(J) = (\mu_L + 1/p_0)^2 \cdot np_0(1 - p_0).$$

Combining and using $\text{Var}(S) = \sigma_L^2 + (1 - p_0)/p_0^2$:

$$v_n = np_0 \left[ \sigma_L^2 + \frac{1 - p_0}{p_0^2} \right] + np_0(1 - p_0) \left( \mu_L + \frac{1}{p_0} \right)^2.$$

For $p_0 \ll 1$ and outputs dominating inputs ($\mu_L \ll 1/p_0$), the dominant terms are:

$$v_n \approx np_0 \cdot \frac{1}{p_0^2} + np_0 \cdot \frac{1}{p_0^2} = \frac{2n}{p_0}.$$

### D.4. Supremum Bound

The memory process $Y_t = X_t - X_0$ resembles a random walk with negative drift $d_n < 0$ and per-step variance $v_n$. For such walks, the all-time supremum satisfies (Feller, 1971):

$$\mathbb{E}\left[\sup_{t \geq 0} Y_t\right] = \frac{v}{2|d|}, \quad \Pr\left(\sup_{t \geq 0} Y_t > x\right) \leq \exp\left(-\frac{2|d|x}{v}\right). \tag{22}$$

Since the batch size $n$ decreases during decode (from $N$ to $N - k$), increments are not strictly i.i.d. Using $n = N$ as an upper bound yields conservative estimates:

$$v_N \approx \frac{2N}{p_0}, \quad |d_N| = Np_0\mu_L.$$

The expected supremum is:

$$\mathbb{E}\left[\sup_{t \geq 0} Y_t\right] \approx \frac{v_N}{2|d_N|} = \frac{2N/p_0}{2Np_0\mu_L} = \frac{1}{p_0^2\mu_L} \triangleq \bar{v}.$$

We introduce $\bar{v} = (p_0^2\mu_L)^{-1}$ as shorthand for the remainder of this proof; it characterizes the memory volatility arising from stochastic completion timing, and is notably independent of $N$.

**Probabilistic bound.** For OOM probability at most $\epsilon$, we require $\Pr(\sup Y_t > C - \mathbb{E}[X_0]) \leq \epsilon$. From (22):

$$\Pr(\sup Y_t > x) \leq \exp\left(-\frac{2|d_N|x}{v_N}\right) = \exp\left(-\frac{x}{\bar{v}}\right).$$

Setting this to $\epsilon$ and solving: $x_\epsilon = \bar{v}\ln(1/\epsilon)$.

The constraint $\mathbb{E}[X_0] + x_\epsilon \leq C$ yields:

$$N\mu_L + \frac{N(1 - \theta)}{\theta p_0} \ln \frac{1}{1 - \theta} + \bar{v}\ln \frac{1}{\epsilon} \leq C.$$

Solving for $N$ gives (6). $\qquad \square$

## D.5. Comparison of Batch Size Bounds

We present three batch size bounds of increasing conservatism:

**Static bound** (ignores decode dynamics):

$$N_{\text{static}}^* = \left\lfloor \frac{C}{\mu_L + \frac{1-\theta}{\theta p_0} \ln \frac{1}{1-\theta}} \right\rfloor.$$

**Expected peak bound** (uses $\mathbb{E}[\sup Y_t] = \bar{v}$):

$$N_{\text{expected}}^* = \left\lfloor \frac{C - \bar{v}}{\mu_L + \frac{1-\theta}{\theta p_0} \ln \frac{1}{1-\theta}} \right\rfloor.$$

**Probabilistic bound** (OOM probability $\leq \epsilon$):

$$N^* = \left\lfloor \frac{C - \bar{v} \ln(1/\epsilon)}{\mu_L + \frac{1-\theta}{\theta p_0} \ln \frac{1}{1-\theta}} \right\rfloor.$$

These satisfy $N^* \leq N_{\text{expected}}^* \leq N_{\text{static}}^*$. The gap depends on $\bar{v} \ln(1/\epsilon)$, which is typically small relative to $C$ for practical workloads.

## D.6. Runtime Refinement: KV-aware Feasibility Gate

The design-time bound $N^*$ above guarantees *expected-value* memory safety: with $N \leq N^*$, the supremum of the memory process exceeds the KV capacity with probability at most $\epsilon$ under the model assumptions. Under high output-length variance ($\sigma_O$ large), the realized peak can transiently deviate from $\mathbb{E}[\sup Y_t]$ beyond the $\epsilon$-tolerance, especially for MoE models where weight activations leave less headroom (e.g., Qwen3-30B-A3B with $\sigma_O \in \{153, 245\}$ on ShareGPT/WildChat).

To handle this at runtime, our scheduler augments the ratio-based phase $1 \rightarrow 2$ transition condition with an instantaneous KV-occupancy check. The original integer-arithmetic condition fillable $\cdot (N - k^*) \geq n \cdot k^*$ is gated by

$$\text{free\_blocks} \geq f_{KV} \cdot \text{total\_blocks}, \quad f_{KV} = \min\left(0.6, \max\left(0.05, \frac{N \cdot \mu_O \cdot s_{KV}}{b \cdot \text{total\_blocks}} + f_0\right)\right), \tag{23}$$

where $b$ is the block size, $s_{KV}$ a safety multiplier on $\mu_O$ (default $s_{KV} = 0.5$), and $f_0$ a base reserve (default 0). The threshold $f_{KV}$ scales with the expected per-batch decode KV usage and is clipped into $[0.05, 0.6]$ for numerical stability.

When the gate fires, phase 2 is deferred and the scheduler stays in phase 1, letting existing decodes free KV via natural completion before attempting refill. This prevents the DECODE$\leftrightarrow$REFILL_PREFILL oscillation triggered when phase 2 ratio-eligibility coincides with insufficient KV for prefill block allocation (which would otherwise bounce back through the vLLM preemption path). The gate is orthogonal to $N^*$: $N^*$ bounds the static batch parameter, while the gate adds a runtime feasibility check at the moment of switching, providing defense-in-depth against transient pressure. Empirically (Table 2, H200 block), this reduces the EB($\hat{k}^*$) vs. v1 gap on Qwen3-30B-A3B ShareGPT from $-12.0\%$ to $-2.5\%$ and on NuminaMath from $-11.4\%$ to $-8.2\%$.

## D.7. Online Adaptive Algorithm: Pseudo-code

Algorithm 1 provides the pseudo-code for the online controller described in Section 3.4, which jointly updates $(\hat{k}^*, \hat{N}^*)$ from sliding-window estimates of the workload parameters.

---

**Algorithm 1** Online Joint Adaptation of $(\hat{k}^*, \hat{N}^*)$

---

**Require:** System parameters $(\alpha_p, \alpha_d, \beta_d)$; cache budget $C$; risk level $\epsilon$; window size $W$; minimum window $W_{\min}$; update interval $T_{\mathrm{upd}}$

**Ensure:** Continuously updated $(\hat{k}^*, \hat{N}^*)$

1: Initialize $\mathcal{W}_O, \mathcal{W}_L \leftarrow \emptyset$, $j \leftarrow 0$
2: Initialize $(\hat{k}^*, \hat{N}^*) \leftarrow (\lfloor \theta_{\mathrm{default}} N_{\mathrm{default}} \rfloor, N_{\mathrm{default}})$
3: **for** each completed request with input length $L$ and output length $O$ **do**
4:     $\mathcal{W}_O$.append($O$); $\mathcal{W}_L$.append($L$); **if** $|\mathcal{W}_O| > W$ **then** pop oldest (same for $\mathcal{W}_L$)
5:     $j \leftarrow j + 1$
6:     **if** $j \geq T_{\mathrm{upd}}$ **and** $|\mathcal{W}_O| \geq W_{\min}$ **then**
7:         $(\hat{p}_0, \hat{\eta}) \leftarrow \mathrm{FitLinearHazard}(\mathcal{W}_O)$
8:         $\hat{\mu}_L \leftarrow \mathrm{Mean}(\mathcal{W}_L)$
9:         $\hat{\theta}_0 \leftarrow \mathrm{SolveCFR}(\hat{p}_0)$                                           ▷ Eq. (3)
10:        $\hat{\theta}^* \leftarrow \mathrm{Clip}(\hat{\theta}_0 + \widehat{\Delta\theta}, \theta_{\min}, \theta_{\max})$                   ▷ Eq. (5)
11:        $\hat{N}^* \leftarrow \mathrm{MemSafeBatch}(\hat{\theta}^*, \hat{p}_0, \hat{\mu}_L)$                ▷ Eq. (6)
12:        $\hat{k}^* \leftarrow \lfloor \hat{\theta}^* \hat{N}^* \rfloor$; apply $(\hat{k}^*, \hat{N}^*)$; $j \leftarrow 0$
13:     **end if**
14: **end for**

---

# E. Workload Characterization and IFR Validation

## E.1. Workload Distribution Analysis

Table 9 reports basic input and output-length statistics and whether the workload exhibits increasing failure rate (IFR) in this reliable region. Note that WildChat comprises 3000 multi-turn conversations rather than independent requests; with an average of 9.3 turns per conversation (range $[6, 55]$), this yields approximately 27,900 total requests, providing comparable scale to the other workloads. Figure 11 summarizes output-length distributions and empirical hazard rates for the real workloads used in our evaluation. We estimate the hazard rate as $\hat{h}(t) = \#\{O_i = t\}/\#\{O_i \geq t\}$, and only interpret $\hat{h}(t)$ up to the 95th percentile due to tail sparsity.

*Table 9.* Input and output length statistics for experimental workloads.

| Workload | Samples | Input | | Output | | | |
| --- | --- | --- | --- | --- | --- | --- | --- |
| | | Range | $\mu_L$ | Range | $\mu_O$ | $\sigma_O$ | IFR |
| ShareGPT | 4000 | $[1, 20448]$ | 105 | $[2, 500]$ | 280 | 153 | ✓ |
| LongBench | 4000 | $[1000, 4000]$ | 2492 | $[1, 19]$ | 11 | 4 | ✓ |
| NuminaMath | 4000 | $[16, 1395]$ | 122 | $[800, 3149]$ | 1039 | 228 | ✓ |
| WildChat | 3000† | $[6, 55]$‡ | 9.3‡ | $[1, 1934]$ | 310 | 245 | ✓ |

†Conversations. ‡Turns per conversation.

## E.2. Baseline Threshold Validation

This appendix validates the structural properties of the CFR baseline normalized threshold $\theta_0$ implied by Proposition 3.1:

- **(P1) Scale invariance:** $\theta_0$ is invariant to batch size $N$.
- **(P2) Input-length independence:** $\theta_0$ is insensitive to mean input length $\mu_L$.
- **(P3) Output-length dependence:** $\theta_0$ decreases with mean output length $\mu_O$.

We empirically study how the optimal $\hat{\theta}^*$ varies with these parameters under geometric (CFR) output lengths. All experiments use Qwen3-8B on H200 with 4,000 requests per configuration (including a 100-request warmup).

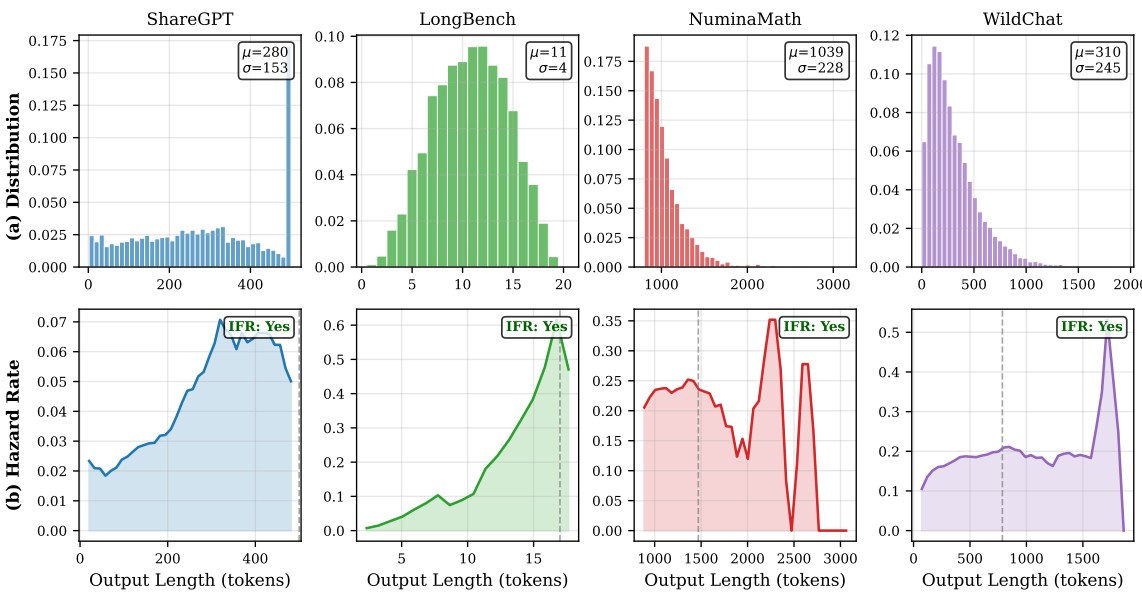

*Figure 11.* Output length distributions and hazard rates of real workloads (Gemma-3-1B-IT on NVIDIA RTX PRO 6000). (a) Output length histograms. (b) Empirical hazard rate $\hat{h}(t) = \#\{O_i = t\}/\#\{O_i \geq t\}$. Dashed lines mark the 95th percentile; beyond this point estimates become unreliable due to data sparsity.

### E.2.1. SCALE INVARIANCE AND OUTPUT-LENGTH DEPENDENCE (P1, P3)

**Setup.** We sweep $N \in \{64, 128, 256, 512\}$ and $\mu_O \in \{64, 128, 192, 256, 320, 384\}$ with $\mu_L = 1$. For each configuration, we run a threshold sweep (3 runs) and report the empirically optimal $\hat{\theta}^* = \hat{k}^*/N$ in Table 10.

**Results.** Consistent with Proposition 3.1:

- **(P1)** For fixed $\mu_O$, $\hat{\theta}^*$ shows no systematic dependence on $N$; the row summaries $\bar{\theta}^* \pm \mathrm{std}$ exhibit no trend with $N$.
- **(P3)** The row means decrease from 0.46 at $\mu_O = 64$ to 0.23 at $\mu_O = 384$, supporting the predicted monotone relationship.

The remaining variance is expected due to discrete $k$ choices and measurement noise.

*Table 10.* Empirical optimal threshold $\hat{\theta}^*$ across batch sizes and output lengths (geometric outputs, $\mu_L = 1$).

| | Batch Size $N$ | | | | |
|---|---|---|---|---|---|
| $\mu_O$ | 64 | 128 | 256 | 512 | $\bar{\theta}^* \pm \mathrm{std}$ |
| 64 | 0.53 | 0.46 | 0.40 | 0.46 | $0.46 \pm 0.05$ |
| 128 | 0.53 | 0.33 | 0.33 | 0.40 | $0.40 \pm 0.09$ |
| 192 | 0.39 | 0.20 | 0.13 | 0.60 | $0.33 \pm 0.21$ |
| 256 | 0.39 | 0.27 | 0.27 | 0.40 | $0.33 \pm 0.07$ |
| 320 | 0.33 | 0.27 | 0.33 | 0.13 | $0.26 \pm 0.09$ |
| 384 | 0.33 | 0.20 | 0.13 | 0.27 | $0.23 \pm 0.08$ |

### E.2.2. INPUT-LENGTH INDEPENDENCE (P2)

**Setup.** To probe dependence on $\mu_L$, we fix $N = 128$ and $\mu_O = 64$, vary $\mu_L \in \{1, 32, 64, 128\}$, and sweep $k$ subject to $k\mu_L \leq 16384$. Table 11 reports throughput for each $k$ and highlights the best $k^*$ per $\mu_L$.

**Results.** While the optimal discrete $k^*$ shifts with $\mu_L$ under the fixed token-budget constraint, the corresponding normalized optimum $\theta^*$ remains in a comparable range (0.438–0.625), indicating weak sensitivity to $\mu_L$ as predicted by Proposition 3.1.

*Table 11.* Throughput (tokens/s) for varying $k^*$ and $\mu_L$ with $N = 128$, $\mu_O = 64$. Bold indicates optimal $k^*$ for each $\mu_L$. All experiments satisfy $k \times \mu_L \leq 16384$.

| $k^*$ | $\theta$ | $\mu_L = 1$ | $\mu_L = 32$ | $\mu_L = 64$ | $\mu_L = 128$ |
|---|---|---|---|---|---|
| 8 | 0.062 | 3036 | 3366 | 3331 | 3047 |
| 16 | 0.125 | 4223 | 4276 | 4273 | 3867 |
| 24 | 0.188 | 5091 | 4949 | 4554 | 4278 |
| 32 | 0.250 | 5266 | 5413 | 4776 | 4421 |
| 40 | 0.312 | 6094 | 5652 | 5356 | 4427 |
| 48 | 0.375 | 5943 | 5801 | 5494 | 4578 |
| 56 | 0.438 | **6627** | 5735 | 5460 | 4701 |
| 64 | 0.500 | 6368 | 5990 | 5522 | 4580 |
| 72 | 0.562 | 5830 | 6059 | **5553** | **4825** |
| 80 | 0.625 | 5986 | **6103** | 5342 | 4701 |
| 88 | 0.688 | 6337 | 6001 | 5488 | 4637 |
| 96 | 0.750 | 6330 | 5892 | 5338 | 4563 |
| 104 | 0.812 | 6150 | 5751 | 5242 | 4460 |
| 112 | 0.875 | 5874 | 5461 | 4998 | 4241 |
| 120 | 0.938 | 5543 | 5184 | 4906 | 4300 |
| 128 | 1.000 | 5626 | 5313 | 4995 | 4315 |
| Optimal $k^*$ | | 56 | 80 | 72 | 72 |
| Optimal $\theta^*$ | | 0.438 | 0.625 | 0.562 | 0.562 |

### E.3. Empirical Evidence for $\Delta\theta > 0$ under IFR

Theorem 3.2 predicts that the IFR correction satisfies $\Delta\theta > 0$, i.e., IFR workloads favor later switching than the CFR baseline. We empirically validate this prediction through controlled experiments.

**Experimental Setup.** We generate synthetic output lengths from Gamma distributions with varying shape parameters to simulate different hazard rate behaviors. Specifically, we use $O \sim \text{Gamma}(a, \lambda)$ where:

- **CFR case** ($a = 1$): The Gamma distribution reduces to an exponential distribution, which exhibits a constant hazard rate $h(t) = \lambda$.
- **IFR case** ($a = 2$): The Gamma distribution with shape parameter $a > 1$ exhibits an increasing failure rate, where the hazard rate $h(t)$ increases monotonically with $t$.

To ensure a fair comparison, we adjust the rate parameter $\lambda$ such that the mean output length $\mu_O = a/\lambda$ remains constant across both settings. This isolates the effect of the hazard rate structure from differences in average workload characteristics. For each configuration, we sweep the normalized threshold $\theta = k^*/N$ from 0 to 1 and measure the resulting throughput. All experiments are repeated three times with different random seeds to assess variability.

**Results.** Figure 12 illustrates the throughput as a function of $\theta$ under both CFR and IFR output-length distributions. The shaded regions represent 95% confidence intervals across the three runs. Under the CFR setting (left panel, $a = 1$), the optimal threshold is $\theta^* = 0.16$. Under the IFR setting (right panel, $a = 2$), the optimal threshold shifts to $\theta^* = 0.36$, representing a substantial increase of $\Delta\theta = 0.20$. This empirical observation confirms the theoretical prediction that IFR workloads benefit from higher switching thresholds, as later switching allows the system to exploit the accelerating completion rate characteristic of IFR distributions.

### E.4. Statistical Tests for IFR in Real Workloads

Section 3.2 (Theorem 3.2) and Appendix E.1 report empirical hazard rates $\hat{h}(t)$ that visually appear monotonically increasing for our real-world workloads. Here we provide formal statistical tests of CFR vs. IFR. We apply two complementary tests on the empirical $\hat{h}(t)$ for each workload, restricted to the reliable region $t \in [1, t_{95}]$ where $t_{95}$ is the 95th percentile of output length: (i) the **Mann–Kendall** non-parametric trend test ($H_0$: no monotonic trend); (ii) a parametric **linear regression** $h(t) = p_0 + \eta t$ ($H_0$: $\eta = 0$).

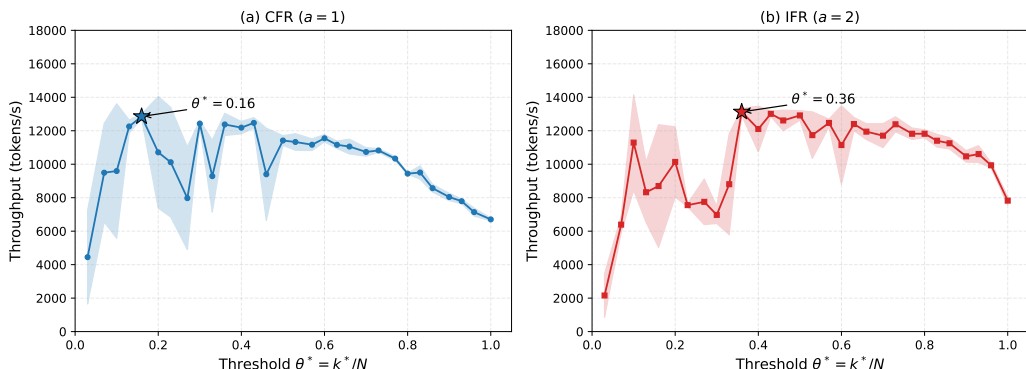

*Figure 12.* Throughput vs. switching threshold $\theta$ under CFR ($a = 1$, left) and IFR ($a = 2$, right) output-length distributions modeled by Gamma distributions with identical mean output lengths. The shaded regions indicate 95% confidence intervals over three runs. The optimal $\theta^*$ shifts from 0.16 (CFR) to 0.36 (IFR), consistent with $\Delta\theta > 0$ in Theorem 3.2.

*Table 12.* Statistical tests for IFR on real-world workloads. All three reject CFR ($H_0$: $\eta = 0$ or no monotonic trend) at $p < 10^{-5}$ under both tests, supporting Theorem 3.2's IFR assumption. LongBench is excluded as $\mu_O \approx 12$ is too short for hazard-rate analysis.

| Workload | $n$ | $p$ (Mann–Kendall) | $p$ (regression) |
|---|---|---|---|
| ShareGPT | 4,000 | $6.6 \times 10^{-10}$ | $2.4 \times 10^{-13}$ |
| NuminaMath | 4,000 | $1.0 \times 10^{-8}$ | $5.2 \times 10^{-10}$ |
| WildChat | 3,000 | $1.9 \times 10^{-5}$ | $3.7 \times 10^{-7}$ |

### E.5. Sensitivity to Hazard-Rate Estimation Accuracy

Theorem 3.2 and Algorithm 1 estimate $\hat{p}_0$ online from a sliding window of recent completions; if the estimate drifts during transient periods, the resulting $\hat{\theta}^*$ may move off the true optimum. We characterize this sensitivity by sweeping fixed $\theta^* \in [0.1, 0.9]$ and comparing against the adaptive controller on Qwen3-8B, RTX PRO 6000, with 3,000 requests per configuration.

Table 13 reports throughput under (i) a synthetic Gamma($a{=}2$) workload with $\mu_O = 256$, and (ii) ShareGPT ($\mu_O \approx 280$). Two observations support robustness to estimation error: **(1) Wide throughput plateau.** The plateau spans $\theta^* \in [0.6, 0.9]$ on Gamma ($\geq 87\%$ of peak) and $[0.5, 0.8]$ on ShareGPT ($\geq 95\%$). The optimal $\theta^*$ shifts across workloads (0.7 vs. 0.6), but any choice within the plateau yields near-peak throughput, so moderate estimation error has limited impact. **(2) Adaptive estimator stays in the plateau.** The online controller achieves **98% of peak** on both workloads. Online $\hat{p}_0$ shows $2.8 \times 10^{-4}$ absolute error (15% relative) on Gamma versus $7 \times 10^{-6}$ (0.5% relative) on ShareGPT—a $30\times$ gap in relative error—yet both attain 98% peak. Combined with the threshold clipping $\hat{\theta}^* \in [\theta_{\min}, \theta_{\max}]$ and vLLM's preemption mechanism, this provides defense-in-depth against transient mis-estimation.

*Table 13.* Sensitivity to hazard-rate estimation accuracy. "Adaptive" uses the online controller (Algorithm 1); other columns sweep fixed $\theta^*$. Throughput in tok/s; "% of peak" below each row.

| Workload | $\theta^*$ | 0.1 | 0.2 | 0.3 | 0.4 | 0.5 | 0.6 | 0.7 | 0.8 | 0.9 | Adaptive |
|---|---|---|---|---|---|---|---|---|---|---|---|
| Gamma($a{=}2$) | Throughput | 3,303 | 3,351 | 3,399 | 3,435 | 3,472 | 3,585 | **4,129** | 3,991 | 3,877 | **4,056** |
| ($\mu_O{=}256$) | % peak | 80% | 81% | 82% | 83% | 84% | 87% | **100%** | 97% | 94% | **98%** |
| ShareGPT | Throughput | 6,133 | 6,347 | 6,620 | 6,676 | 7,023 | **7,180** | 6,988 | 6,853 | 6,544 | **7,048** |
| ($\mu_O{\approx}280$) | % peak | 85% | 88% | 92% | 93% | 98% | **100%** | 97% | 95% | 91% | **98%** |

# F. Additional End-to-End Comparisons

## F.1. Sensitivity Analysis Across Workloads and Platforms

This appendix provides comprehensive sensitivity analyses for all workloads (ShareGPT, LongBench, WildChat, Numina-Math) across H200 and RTX PRO 6000 platforms. For each workload-platform combination, we perform a full grid search over token budget $B$ and batch size $N$ for v0, v1, and EB($\hat{k}^*$) schedulers. Tables 14 and 15 below report four robustness metrics: coefficient of variation (CV), range ratio, $B$-Sens, and $N$-Sens, with lower values indicating better stability across configurations.

### F.1.1. PARAMETER SENSITIVITY METRICS

We define the following metrics to quantify scheduling robustness across deployment configurations. Let $\text{TP}(B, N)$ denote the measured throughput (requests/second) for a scheduler under token budget $B$ and batch size $N$, and let $\mathcal{B}$ and $\mathcal{N}$ be the sets of evaluated token budgets and batch sizes, respectively.

**Coefficient of Variation (CV).** The CV measures overall throughput variability across all configurations:

$$\text{CV} = \frac{\text{std}(\text{TP}(B, N) : B \in \mathcal{B}, N \in \mathcal{N})}{\text{mean}(\text{TP}(B, N) : B \in \mathcal{B}, N \in \mathcal{N})}$$

where $\text{std}(\cdot)$ and $\text{mean}(\cdot)$ denote standard deviation and mean, respectively. Lower values indicate greater stability.

**Range Ratio.** The range ratio captures the extremes of performance variation:

$$\text{Range} = \frac{\max_{B,N} \text{TP}(B, N)}{\min_{B,N} \text{TP}(B, N)}$$

Values closer to 1.0 indicate more consistent performance across configurations.

**Token Budget Sensitivity ($B$-Sens).** This metric isolates the effect of varying $B$ while holding $N$ fixed:

$$B\text{-Sens} = \frac{1}{|\mathcal{N}|} \sum_{N \in \mathcal{N}} \frac{\text{std}(\text{TP}(B, N) : B \in \mathcal{B})}{\text{mean}(\text{TP}(B, N) : B \in \mathcal{B})}$$

**Batch Size Sensitivity ($N$-Sens).** This metric isolates the effect of varying $N$ while holding $B$ fixed:

$$N\text{-Sens} = \frac{1}{|\mathcal{B}|} \sum_{B \in \mathcal{B}} \frac{\text{std}(\text{TP}(B, N) : N \in \mathcal{N})}{\text{mean}(\text{TP}(B, N) : N \in \mathcal{N})}$$

### F.1.2. OPTIMAL CONFIGURATION DETAILS

Tables 14 and 15 report the optimal token budget ($B$) and batch size ($N$) for each scheduler across all workloads on RTX PRO 6000 and H200 respectively. These configurations were selected via grid search to maximize throughput.

**Configuration variability.** Optimal configurations vary substantially across workloads, models, and hardware. Token budgets range from 4,096 to 18,432, while batch sizes span 256 to 2,048. No single configuration universally dominates, highlighting the importance of workload-aware tuning.

**Sensitivity patterns.** Prefill-heavy workloads (LongBench) exhibit the lowest sensitivity (CV $< 5\%$, range ratio $< 1.2\times$), as throughput is dominated by prefill computation with limited scheduling flexibility. In contrast, balanced workloads (ShareGPT) show higher sensitivity, particularly for larger models (Qwen3-30B-A3B: CV up to 35%, range ratio up to $3.47\times$), where the prefill–decode balance is more delicate. Decode-heavy workloads (NuminaMath) demonstrate moderate sensitivity, with most configurations achieving CV $< 8\%$.

**Scheduler comparison.** All three schedulers exhibit comparable sensitivity profiles within each workload category. EB($\hat{k}^*$) does not introduce additional tuning burden: its average CV and range ratio are similar to v0 and v1 across most configurations. The primary differentiator remains throughput rather than robustness.

*Table 14.* Optimal configurations and sensitivity metrics for each scheduler across workloads on RTX PRO 6000. Lower sensitivity values indicate greater robustness.

| Scheduler | Metric | ShareGPT | | | LongBench | | | WildChat | | | NuminaMath | | |
| | | Qwen3-8B | 30B-A3B | Gemma | Qwen3-8B | 30B-A3B | Gemma | Qwen3-8B | 30B-A3B | Gemma | Qwen3-8B | 30B-A3B | Gemma |
| --- | --- | --- | --- | --- | --- | --- | --- | --- | --- | --- | --- | --- | --- |
| v0 | $B$ | 14336 | 4096 | 18432 | 16384 | 14336 | 16384 | 18432 | 10240 | 14336 | 8192 | 10240 | 16384 |
| | $N$ | 1536 | 1024 | 512 | 512 | 256 | 1024 | 1024 | 512 | 1536 | 256 | 512 | 512 |
| | CV | 4.7% | 12.5% | 6.3% | 1.2% | 3.9% | 4.7% | 7.8% | 9.4% | 4.6% | 2.3% | 5.7% | 7.2% |
| | Range | 1.17× | 1.56× | 1.23× | 1.05× | 1.14× | 1.18× | 1.29× | 1.38× | 1.17× | 1.08× | 1.21× | 1.26× |
| | $B$-Sens | 1.3% | 0.5% | 2.1% | 1.1% | 3.6% | 4.7% | 3.2% | 4.1% | 2.8% | 0.9% | 1.7% | 3.4% |
| | $N$-Sens | 4.9% | 18.7% | 6.8% | 0.7% | 1.8% | 1.3% | 8.3% | 10.2% | 4.3% | 2.4% | 6.1% | 7.9% |
| v1 | $B$ | 16384 | 8192 | 4096 | 10240 | 14336 | 14336 | 18432 | 14336 | 8192 | 14336 | 14336 | 14336 |
| | $N$ | 1536 | 1536 | 256 | 1024 | 2048 | 512 | 1024 | 1024 | 256 | 256 | 256 | 256 |
| | CV | 3.3% | 15.2% | 3.7% | 1.0% | 2.8% | 3.9% | 6.5% | 13.0% | 2.8% | 3.6% | 6.5% | 6.7% |
| | Range | 1.12× | 1.57× | 1.13× | 1.04× | 1.09× | 1.13× | 1.24× | 1.54× | 1.12× | 1.11× | 1.20× | 1.22× |
| | $B$-Sens | 0.8% | 0.4% | 0.5% | 1.1% | 2.8% | 4.2% | 4.9% | 2.0% | 0.8% | 0.5% | 0.2% | 0.6% |
| | $N$-Sens | 3.6% | 16.8% | 4.0% | 0.5% | 1.5% | 0.6% | 5.8% | 14.3% | 2.9% | 3.9% | 7.2% | 7.4% |
| EB($\hat{k}^*$) | $B$ | 14336 | 16384 | 16384 | 16384 | 16384 | 14336 | 10240 | 18432 | 14336 | 4096 | 16384 | 10240 |
| | $N$ | 1536 | 512 | 512 | 512 | 256 | 1024 | 1024 | 512 | 2048 | 256 | 256 | 256 |
| | CV | 12.0% | 22.0% | 5.1% | 0.8% | 3.6% | 4.5% | 8.5% | 10.8% | 2.2% | 2.7% | 8.2% | 2.3% |
| | Range | 1.40× | 2.12× | 1.19× | 1.03× | 1.14× | 1.16× | 1.28× | 1.53× | 1.09× | 1.09× | 1.28× | 1.10× |
| | $B$-Sens | 0.4% | 1.9% | 0.6% | 0.9% | 3.7% | 4.9% | 4.5% | 5.7% | 1.7% | 0.8% | 1.3% | 1.4% |
| | $N$-Sens | 13.2% | 24.1% | 5.6% | 0.4% | 1.4% | 0.5% | 9.1% | 10.7% | 2.1% | 2.9% | 8.9% | 2.3% |

*Table 15.* Optimal configurations and sensitivity metrics for each scheduler across workloads on H200. Lower sensitivity values indicate greater robustness.

| Scheduler | Metric | ShareGPT | | | LongBench | | | WildChat | | | NuminaMath | | |
| | | Qwen3-8B | 30B-A3B | Gemma | Qwen3-8B | 30B-A3B | Gemma | Qwen3-8B | 30B-A3B | Gemma | Qwen3-8B | 30B-A3B | Gemma |
| --- | --- | --- | --- | --- | --- | --- | --- | --- | --- | --- | --- | --- | --- |
| v0 | $B$ | 18432 | 14336 | 16384 | 18432 | 18432 | 18432 | 18432 | 16384 | 16384 | 10240 | 10240 | 14336 |
| | $N$ | 2048 | 1536 | 1024 | 256 | 256 | 512 | 1536 | 1024 | 1024 | 256 | 1024 | 512 |
| | CV | 2.9% | 35.2% | 5.4% | 0.9% | 4.5% | 3.8% | 2.9% | 5.5% | 4.9% | 3.5% | 2.0% | 6.3% |
| | Range | 1.12× | 3.47× | 1.19× | 1.03× | 1.15× | 1.13× | 1.20× | 1.21× | 1.18× | 1.11× | 1.08× | 1.22× |
| | $B$-Sens | 2.7% | 35.7% | 3.9% | 0.8% | 4.9% | 4.1% | 2.0% | 1.6% | 4.7% | 1.1% | 1.3% | 4.6% |
| | $N$-Sens | 2.4% | 29.5% | 4.8% | 0.5% | 0.4% | 0.6% | 1.9% | 5.9% | 4.2% | 3.8% | 1.9% | 5.7% |
| v1 | $B$ | 10240 | 8192 | 14336 | 14336 | 14336 | 16384 | 4096 | 4096 | 18432 | 14336 | 8192 | 16384 |
| | $N$ | 1024 | 2048 | 512 | 256 | 2048 | 512 | 2048 | 1536 | 256 | 256 | 512 | 1024 |
| | CV | 3.5% | 23.0% | 6.1% | 0.9% | 4.7% | 4.8% | 3.3% | 8.6% | 6.4% | 3.8% | 2.4% | 7.7% |
| | Range | 1.12× | 2.02× | 1.24× | 1.04× | 1.16× | 1.18× | 1.16× | 1.30× | 1.24× | 1.12× | 1.08× | 1.26× |
| | $B$-Sens | 0.9% | 1.4% | 4.5% | 0.9% | 5.0% | 5.1% | 2.9% | 1.3% | 6.1% | 0.7% | 0.6% | 5.4% |
| | $N$-Sens | 3.8% | 25.3% | 6.5% | 0.5% | 0.6% | 1.0% | 2.6% | 9.4% | 5.4% | 4.2% | 2.6% | 7.7% |
| EB($\hat{k}^*$) | $B$ | 16384 | 4096 | 10240 | 14336 | 16384 | 14336 | 16384 | 14336 | 18432 | 18432 | 10240 | 18432 |
| | $N$ | 1536 | 1536 | 256 | 2048 | 1024 | 1024 | 1024 | 1024 | 1536 | 256 | 512 | 512 |
| | CV | 9.3% | 29.1% | 9.4% | 1.1% | 4.4% | 4.5% | 3.5% | 12.3% | 3.7% | 2.5% | 2.3% | 6.4% |
| | Range | 1.33× | 2.50× | 1.32× | 1.04× | 1.16× | 1.16× | 1.12× | 1.60× | 1.14× | 1.08× | 1.07× | 1.23× |
| | $B$-Sens | 0.8% | 1.2% | 3.2% | 1.1% | 4.7% | 4.9% | 2.0% | 3.4% | 3.6% | 0.6% | 0.9% | 3.6% |
| | $N$-Sens | 10.2% | 32.0% | 10.2% | 0.6% | 0.7% | 0.9% | 3.7% | 13.0% | 3.6% | 2.7% | 2.5% | 6.7% |

### F.1.3. Optimal Configurations for Synthetic Workloads (Figure 4)

Tables 14 and 15 cover the four real-world workloads. The three synthetic workloads used in Figure 4 (decode-heavy, balanced, prefill-heavy) have separate optima, obtained from the same grid search over $B \in \{4096, \ldots, 18432\}$ and $N \in \{256, \ldots, 2048\}$ and reported in Table 16.

*Table 16.* Best-throughput $(B, N)$ per scheduler for each synthetic workload (Qwen3-8B). These configurations underlie Figure 4; selected from grid search.

|  |  | RTX PRO 6000 | | H200 | |
| --- | --- | --- | --- | --- | --- |
| Workload | Scheduler | $B$ | $N$ | $B$ | $N$ |
| Decode-heavy | v1 | 4096 | 256 | 18432 | 512 |
|  | EB($\hat{k}^*$) | 4096 | 512 | 14336 | 1536 |
| Balanced | v1 | 8192 | 256 | 10240 | 512 |
|  | EB($\hat{k}^*$) | 14336 | 512 | 18432 | 1024 |
| Prefill-heavy | v1 | 10240 | 256 | 18432 | 1024 |
|  | EB($\hat{k}^*$) | 18432 | 512 | 10240 | 512 |

### F.2. End-to-End Results for Gemma-3-1B-IT

Table 17 reports end-to-end throughput for Gemma-3-1B-IT on both GPUs. The 1B-parameter model sits at a different operating point than the Qwen models in the main text: its lightweight Attention avoids bandwidth-induced interference (Appendix A.2), so the marginal-cost gap $\beta_{\mathrm{MB}}^{e} - \beta_{\mathrm{EB}}^{w}$ on the LHS of (7) is small. The RHS is also small at this scale because per-iteration fixed overhead $\alpha$ scales with model size. Both sides shrink together, leaving a narrow but consistent margin in EB's favor. On the RTX PRO 6000, EB($\hat{k}^*$) achieves consistent improvements over v1 across all workloads (average +3.9%), primarily from better scheduling of decode-heavy phases. On the H200, the advantage is larger on ShareGPT (+10.6%) and LongBench (+4.4%), while WildChat shows a slight regression (−1.5%).

*Table 17.* Throughput (RPS) for Gemma-3-1B-IT on real-world workloads. % Diff. as defined in Table 2.

|  | RTX PRO 6000 | | | | H200 | | | |
| --- | --- | --- | --- | --- | --- | --- | --- | --- |
| Workload | v0 | v1 | EB($\hat{k}^*$) | % Diff. | v0 | v1 | EB($\hat{k}^*$) | % Diff. |
| ShareGPT | 49.31 | 48.94 | **50.84** | +3.9% | 51.24 | 52.73 | **58.33** | +10.6% |
| LongBench | 51.09 | 55.08 | **55.31** | +0.4% | 92.74 | 92.84 | **96.93** | +4.4% |
| WildChat | 55.97 | 53.36 | **56.26** | +5.4% | 71.82 | **75.17** | 74.05 | −1.5% |
| NuminaMath | 7.01 | 8.36 | **8.87** | +6.1% | 11.25 | 12.49 | **12.80** | +2.5% |
| *Average* |  |  |  | *+3.9%* |  |  |  | *+4.0%* |

### F.3. SLO-Constrained Goodput for EB⁺

Section 4.4 summarizes the SLO-attainment behavior of EB⁺ at moderate load. Here we report per-request SLO attainment (fraction of requests meeting both TTFT and TPOT targets simultaneously) on RTX PRO 6000 with Qwen3-8B at $c = 512$. Under a strict TPOT target ($< 50\,\mathrm{ms}$), all schedulers fail because mean TPOT exceeds the target on this bandwidth-constrained GPU. Under a relaxed target ($< 100\,\mathrm{ms}$), v1 collapses to $\sim 5\%$ attainment because its TPOT distribution is wide; EB⁺ achieves the best balance, attaining $80.3\%$ at TTFT$< 10\,\mathrm{s}$ and $48.4\%$ at TTFT$< 5\,\mathrm{s}$. On H200 at $c = 512$, mean TPOT is universally low ($< 53\,\mathrm{ms}$) so TTFT dominates the SLO; EB⁺ selects v1 there and matches its attainment at all targets.

### F.4. Comparison with Prefill–Decode Disaggregation

Disaggregated serving (DistServe (Zhong et al., 2024), Splitwise (Patel et al., 2024)) is an alternative to MB that physically separates prefill and decode onto dedicated GPU pools. We compare EB⁺ (which separates phases temporally on the same GPUs under data parallelism) against vLLM's built-in disaggregation scheduler.

*Table 18.* SLO attainment (% of requests meeting both TTFT and TPOT targets) on RTX PRO 6000 (Qwen3-8B, $c = 512$). Under strict TPOT all schedulers fail; under relaxed TPOT, EB$^+$ dominates at moderate-to-loose TTFT targets.

| | TPOT $< 50$ ms | | | TPOT $< 100$ ms | | |
|---|---|---|---|---|---|---|
| TTFT target | v1 | EB($\hat{k}^*$) | EB$^+$ | v1 | EB($\hat{k}^*$) | EB$^+$ |
| $< 2$ s | **0.6** | 0.0 | 0.0 | **4.9** | 0.0 | 1.2 |
| $< 5$ s | 0.6 | 0.0 | **1.1** | 5.0 | 6.2 | **48.4** |
| $< 10$ s | 0.6 | 0.5 | **1.3** | 5.8 | 77.3 | **80.3** |

**Two-GPU setting (DP=2 vs. 1P+1D).** Table 19 compares v1, EB$^+$, and vLLM's 1P+1D disaggregation scheduler on $2\times$ RTX PRO 6000 and $2\times$ H200 (Qwen3-8B, $\mu_L = 512$, $\mu_O = 256$). EB$^+$ matches or exceeds v1 throughput at every concurrency on both GPUs. On RTX PRO 6000, EB$^+$ outperforms disaggregation by $+31.8\%$ throughput with $3.3\times$ lower TTFT at $c = 512$ and by $+22.8\%$ over v1 at $c = 2048$. On H200, EB$^+$ leads disaggregation by $+17.7\%$ at $c = 64$ and $+14.8\%$ at $c = 512$, with $+4.3\%$ over v1 at $c = 2048$. Disaggregation OOMs at $c = 2048$ on both GPUs ("–") because KV blocks remain pinned during prefill-to-decode transfer and the scheduler lacks admission backpressure.

*Table 19.* Two-GPU comparison: v1, EB$^+$ (DP=2), and vLLM 1P+1D disaggregation. Throughput in tok/s. "–" denotes OOM.

| $c$ | Metric | $2\times$ H200 | | | $2\times$ RTX PRO 6000 | | |
|---|---|---|---|---|---|---|---|
| | | v1 | EB$^+$ | Disagg | v1 | EB$^+$ | Disagg |
| | Throughput | 26,545 | **26,857** | 22,825 | 8,529 | **8,691** | 8,606 |
| 64 | TTFT (s) | 0.042 | **0.040** | 0.091 | 0.087 | **0.084** | 0.164 |
| | TPOT (ms) | 7.11 | **7.03** | 8.08 | 22.30 | 22.10 | **21.10** |
| | Throughput | **45,731** | 45,539 | 39,649 | 21,526 | **21,710** | 16,467 |
| 512 | TTFT (s) | 0.788 | **0.779** | 3.608 | 1.900 | **1.900** | 6.219 |
| | TPOT (ms) | 29.18 | 29.44 | **22.81** | 58.90 | **58.40** | 58.64 |
| | Throughput | 46,042 | **48,015** | – | 16,228 | **19,926** | – |
| 2048 | TTFT (s) | 7.601 | **6.330** | – | 45.900 | **45.300** | – |
| | TPOT (ms) | **97.44** | 99.71 | – | 150.00 | **63.80** | – |

**Four-GPU setting across P:D ratios.** We further compare DP=4 against disaggregation with 1P+3D, 2P+2D, and 3P+1D allocations on $4\times$ RTX PRO 6000 and $4\times$ H200 (Qwen3-8B, vLLM P2P NCCL KV transfer; "–" denotes OOM/timeout on the decode GPU). Three workload mixes span the prefill–decode spectrum: *prefill-heavy* ($\mu_L = 1024$, $\mu_O = 128$), *balanced* ($\mu_L = 512$, $\mu_O = 512$), and *decode-heavy* ($\mu_L = 128$, $\mu_O = 1024$); 2k requests per setting at $c \in \{128, 256, 512\}$.

Three observations emerge from Table 20:

**(1) The optimal P:D ratio is workload-dependent and choosing wrong is costly.** At $c = 128$ on $4\times$ RTX PRO 6000: prefill-heavy favors 2P+2D ($+63\%$ over EB$^+$), balanced favors 1P+3D ($+14\%$), and decode-heavy favors EB$^+$ itself. No single ratio dominates: 2P+2D is best for prefill-heavy but worst for decode-heavy among disagg options; 1P+3D is best for balanced but worst for prefill-heavy.

**(2) Disaggregation exhibits structural memory fragility.** Under disaggregation, a request's entire prefill KV cache ($L$ tokens) is transferred to the decode GPU in bulk upon prefill completion. Multiple prefill GPUs produce completed requests faster than a single decode GPU can retire them, causing concurrent KV occupancy on the decode GPU to scale as $O(L \cdot n_{\text{pending}})$ with no backpressure mechanism. This explains the OOM pattern: at $c = 256$, 3P+1D OOMs on prefill-heavy ($L = 1024$) but not on decode-heavy ($L = 128$)—the same number of pending requests consumes $8\times$ more KV memory. At $c = 512$, 3P+1D OOMs on balanced ($L = 512$) as well; at $c = 2048$ (Appendix F.4) all disagg configurations OOM.

**(3) EB$^+$ is consistently competitive without manual tuning.** EB$^+$ achieves the best or near-best throughput in 7 of 9 (workload $\times$ $c$) settings on RTX PRO 6000, with 2P+2D excelling on prefill-heavy at $c = 128$ and $c = 256$. Crucially, EB$^+$ maintains 3–18$\times$ lower TTFT than every disagg configuration across every setting and never OOMs. In a deployment where workload mix is unknown or shifts over time, EB$^+$ provides robust performance without operators predicting the optimal

P:D ratio.

*Table 20.* 4-GPU disaggregation comparison across concurrencies $c \in \{128, 256, 512\}$ (Qwen3-8B, 2k prompts). Throughput in tok/s. "–" denotes OOM/timeout. v1 and EB$^+$ both use DP=4.

| $c$ | Workload | Metric | 4× RTX PRO 6000 | | | | | 4× H200 | | | | |
|---|---|---|---|---|---|---|---|---|---|---|---|---|
| | | | v1 | EB$^+$ | 1P+3D | 2P+2D | 3P+1D | v1 | EB$^+$ | 1P+3D | 2P+2D | 3P+1D |
| 128 | Prefill-heavy | Throughput | 25,994 | 26,207 | 23,245 | **42,577** | 31,490 | 67,278 | 65,297 | 40,535 | **82,400** | 70,255 |
| | | TTFT (ms) | 185 | **171** | 4,385 | 476 | 482 | 133 | **124** | 2,763 | 662 | 412 |
| | | TPOT (ms) | 43.0 | 42.8 | **14.6** | 23.1 | 32.6 | 16.1 | 16.7 | **6.4** | 8.5 | 13.1 |
| | Balanced | Throughput | 11,013 | 11,063 | **12,629** | 11,640 | 9,023 | 28,646 | 28,686 | **31,341** | 29,388 | 21,963 |
| | | TTFT (ms) | 121 | **116** | 316 | 229 | 272 | 75 | **74** | 249 | 178 | 233 |
| | | TPOT (ms) | 22.6 | 22.5 | **19.2** | 21.1 | 27.3 | 8.6 | 8.6 | **7.5** | 8.2 | 11.0 |
| | Decode-heavy | Throughput | 7,723 | **7,690** | 7,469 | 6,848 | 5,450 | 18,245 | **19,336** | 18,861 | 17,227 | 13,536 |
| | | TTFT (ms) | 65 | **62** | 223 | 166 | 220 | 46 | **44** | 233 | 179 | 220 |
| | | TPOT (ms) | 18.1 | 18.2 | **18.6** | 20.4 | 25.6 | 7.6 | **7.2** | **7.2** | 7.9 | 10.2 |
| 256 | Prefill-heavy | Throughput | 37,623 | 37,208 | 23,102 | **44,892** | – | **88,488** | 87,268 | 40,638 | 77,117 | – |
| | | TTFT (ms) | 380 | **381** | 10,283 | 3,369 | – | **304** | 307 | 6,103 | 2,614 | – |
| | | TPOT (ms) | 57.8 | 58.5 | **14.5** | 23.3 | – | 23.4 | 23.8 | **6.4** | 8.3 | – |
| | Balanced | Throughput | 16,913 | 16,870 | **19,831** | 17,551 | 11,402 | 43,648 | 43,729 | **47,433** | 42,016 | 26,455 |
| | | TTFT (ms) | 235 | **229** | 844 | 505 | 664 | 158 | **152** | 681 | 395 | 596 |
| | | TPOT (ms) | 28.6 | 28.7 | **22.9** | 26.9 | 42.1 | 10.9 | 10.9 | **8.9** | 10.9 | 17.6 |
| | Decode-heavy | Throughput | 12,678 | **12,699** | 12,176 | 10,595 | 7,021 | **30,696** | 29,763 | 29,495 | 24,820 | 16,682 |
| | | TTFT (ms) | 125 | **119** | 719 | 404 | 592 | 108 | **110** | 715 | 438 | 599 |
| | | TPOT (ms) | 21.4 | **21.3** | 21.7 | 25.4 | 38.9 | 8.7 | 8.9 | **8.5** | 10.6 | 16.0 |
| 512 | Prefill-heavy | Throughput | **47,947** | 47,322 | 22,531 | 43,326 | – | **106,324** | 105,432 | 39,925 | 78,486 | – |
| | | TTFT (ms) | **1,141** | 1,205 | 21,412 | 9,245 | – | **805** | 843 | 12,357 | 5,741 | – |
| | | TPOT (ms) | 85.3 | 86.1 | **14.6** | 23.9 | – | 36.2 | 36.2 | **6.4** | 8.2 | – |
| | Balanced | Throughput | 23,914 | 24,037 | **25,080** | 20,397 | – | **61,106** | 60,988 | 46,261 | 48,895 | – |
| | | TTFT (ms) | **697** | 697 | 4,013 | 1,676 | – | 464 | **413** | 5,753 | 1,128 | – |
| | | TPOT (ms) | 38.7 | 38.6 | **29.9** | 43.4 | – | **14.7** | 14.9 | 9.2 | 17.3 | – |
| | Decode-heavy | Throughput | 18,414 | **18,396** | 17,224 | 13,169 | 7,579 | 46,622 | **46,782** | 38,287 | 30,567 | 17,544 |
| | | TTFT (ms) | 319 | **284** | 1,842 | 1,074 | 2,295 | **251** | 314 | 1,908 | 1,216 | 1,720 |
| | | TPOT (ms) | **28.2** | 28.2 | 28.8 | 39.1 | 69.4 | **10.9** | 10.8 | 11.9 | 16.2 | 29.4 |

