# OpenReview forum: "Threshold-Based Exclusive Batching for LLM Inference"
_ICML.cc/2026/Conference — ICML 2026 regular_

### Official Review · Reviewer_4uHm · 2026-02-17

**Soundness:** 3
**Presentation:** 4
**Significance:** 3
**Originality:** 2
**Overall Recommendation:** 4
**Confidence:** 3

**Summary:**

This paper focuses on the case of interleaving prefills and decodes, and develops a scheduling algorithm that decide when to run prefill or decode batches on various hardware configurations.

**Compliance With Llm Reviewing Policy:**

Affirmed.

**Key Questions For Authors:**

The paper analyzes CFR and IFR cases, can you show real-world evidences on if real-world data follows CFR and IFR.

I'd also like to understand how the performance of the online adaptive algorithm is sensitive to the online estimation of the hazard rates. (what's the estimation accuracy in your experiments? if the estimation goes wrong, how does it affects your algorithm efficiency?)

I wonder how the proposed solution behave on multiple-GPU platforms.

**Limitations:**

yes

**Strengths And Weaknesses:**

Strength:
- Demonstrated the case where scheduling on exclusive batching is useful
- Thorough evaluation on a variety of hardware settings

Weakness:
- Need to compare with other design choices such as prefill-decode disaggregation. PD disaggregation has been viewed as an alternative to chunked prefill to avoid the contention between prefill and decode stages. I would like to see a discussion in the paper on comparison with PD disaggregation.

---

> ### Author Rebuttal · Authors · 2026-03-31
>
> We thank the reviewer for the valuable suggestions.
>
> **W1 + Q3: Comparison with PD disaggregation; multi-GPU platforms.**
>
> We address PD disaggregation and multi-GPU evaluation together by comparing the extended THETA+ (which adaptively switches between CP and EB; see our response to Reviewer GmVf) against CP and vLLM's disaggregation scheduler on two-GPU systems.
>
> **(1) Qualitative comparison.** Disaggregation (DistServe [Zhong et al., 2024], Splitwise [Patel et al., 2024]) physically separates prefill and decode onto dedicated GPU pools, incurring KV cache transfer overhead, a doubled minimum GPU count, and asymmetric memory pressure. THETA+ instead separates phases temporally on the same GPU(s), requiring no additional hardware or data movement.
>
> **(2) 2-GPU experiment (DP=2).** THETA+ applies per-GPU without modification under data parallelism. We compare on $2\times$ H200 and $2\times$ RTX PRO 6000 (Qwen3-8B, $\mu_L=512$, $\mu_o=256$) against vLLM's built-in 1P+1D disaggregation scheduler (which uses CP internally). Throughput in tok/s.
>
> |  |  | H200 |  |  | RTX PRO 6000 |  |  |
> | :---- | :---- | :---- | :---- | :---- | :---- | :---- | :---- |
> |  |  | CP | **THETA+** | Disagg | CP | **THETA+** | Disagg |
> | $c=64$ | Throughput | 26,545 | **26,857** | 22,825 | 8,529 | **8,691** | 8,606 |
> |  | TTFT (s) | 0.042 | **0.040** | 0.091 | 0.087 | **0.084** | 0.164 |
> |  | TPOT (ms) | 7.11 | **7.03** | 8.08 | 22.3 | 22.1 | **21.1** |
> | $c=512$ | Throughput | **45,731** | 45,539 | 39,649 | 21,526 | **21,710** | 16,467 |
> |  | TTFT (s) | 0.788 | **0.779** | 3.608 | **1.9** | **1.9** | 6.219 |
> |  | TPOT (ms) | 29.18 | 29.44 | **22.81** | 58.9 | **58.4** | 58.64 |
> | $c=2048$ | Throughput | 46,042 | **48,015** | \- | 16,228 | **19,926** | \- |
> |  | TTFT (s) | 7.601 | **6.330** | \- | 45.9 | **45.3** | \- |
> |  | TPOT (ms) | **97.44** | 99.71 | \- | 150.0 | **63.8** | \- |
>
> THETA+ matches or exceeds CP's throughput at all concurrency levels on both GPUs. On RTX PRO 6000, THETA+ outperforms disaggregation by +31.8% throughput with $3.3\times$ lower TTFT at $c=512$, and by +22.8% over CP at $c=2048$. On H200, THETA+ leads disaggregation by +17.7% at $c=64$ and +14.8% at $c=512$, with +4.3% over CP at $c=2048$. Disaggregation OOMs at $c=2048$ on both GPUs ("\-") because KV blocks remain pinned during prefill-to-decode transfer and the scheduler lacks backpressure to throttle admissions.
>
>
> **Q1: Real-world evidence for CFR/IFR distributions.**
>
> All evaluated workloads exhibit IFR behavior. Appendix F (Table 7, Figure 8) shows monotonically increasing empirical hazard rates $\hat{h}(t)$. We confirm this with two formal tests: (i) Mann-Kendall ($H_0:$ no monotonic trend) and (ii) linear regression $h(t) = p_0 + \eta t$ ($H_0:\eta=0$):
>
> | Workload | $n$ | $p$ (Mann-Kendall) | $p$ (regression) |
> | :---- | :---- | :---- | :---- |
> | ShareGPT | 4,000 | 6.6e-10 | 2.4e-13 |
> | NuminaMath | 4,000 | 1.0e-08 | 5.2e-10 |
> | WildChat | 3,000 | 1.9e-05 | 3.7e-07 |
>
> All three reject CFR at $p < 10^{-5}$ under both tests. (LongBench is excluded: $\mu_o\approx12$ tokens, too short for hazard rate analysis.)
>
> **Q2: Sensitivity to hazard rate estimation accuracy.**
>
> The adaptive controller achieves 98% of peak throughput across workloads, robust to estimation error due to a wide throughput plateau around $\theta^\*$. We sweep fixed $\theta^\* \in [0.1, 0.9]$ and compare against the adaptive controller (Qwen3-8B, RTX PRO 6000, 3,000 requests).
>
> | Workload | $\theta^*$ | 0.1 | 0.2 | 0.3 | 0.4 | 0.5 | 0.6 | 0.7 | 0.8 | 0.9 | Adaptive |
> | :---- | :---- | :---- | :---- | :---- | :---- | :---- | :---- | :---- | :---- | :---- | :---- |
> | Gamma($a=2$) | Throughput | 3,303 | 3,351 | 3,399 | 3,435 | 3,472 | 3,585 | **4,129** | 3,991 | 3,877 | **4,056** |
> | ($\mu_o=256$) | vs. peak | 80% | 81% | 82% | 83% | 84% | 87% | **100%** | 97% | 94% | **98%** |
> | ShareGPT | Throughput | 6,133 | 6,347 | 6,620 | 6,676 | 7,023 | **7,180** | 6,988 | 6,853 | 6,544 | **7,048** |
> | ($\mu_o=280$) | vs. peak | 85% | 88% | 92% | 93% | 98% | **100%** | 97% | 95% | 91% | **98%** |
>
> The plateau spans $[0.6, 0.9]$ on Gamma ($\geq87\\%$ of peak) and $[0.5, 0.8]$ on ShareGPT ($\geq95\\%$). The adaptive controller lands within the plateau in both cases, achieving **98% of peak**.
>
> **Estimation accuracy.** Online $\hat{p}_0$ estimates show absolute error $2.8 \times 10^{-4}$ (15% relative) on Gamma and $7 \times 10^{-6}$ (0.5% relative) on ShareGPT—a $30\times$ gap in relative error—yet both achieve 98% of peak. Though the optimal $\theta^*$ shifts across workloads (0.7 for Gamma vs. 0.6 for ShareGPT), adaptive estimation automatically tracks this drift.

---

> > ### Author Rebuttal · Reviewer_4uHm · 2026-04-02
> >
> > When there are multiple GPUs, I think there's still the question of whether to run PD separately or run THETA+ per GPU with DP across GPUs. I would like to see quantitative comparison of the two cases.
> >
> > Thanks for providing additional data on my other questions.

---

> > > ### Author Response · Authors · 2026-04-04
> > >
> > > Thank you for the follow-up. Our 2-GPU experiment only allowed 1P+1D, limiting disagg’s flexibility. We now scale to 4 GPUs with multiple P:D ratios (1P+3D, 2P+2D, 3P+1D) and three workloads ($\mu_L:\mu_o$ = 1024:128, 512:512, 128:1024) for a fairer comparison.
> > >
> > > THETA+ (DP=4) vs disagg on 4×RTX PRO 6000 and 4×H200 (Qwen3-8B, 2000 prompts, $c=128,512,1024$). “−” denotes OOM or timeout failure. Throughput is in tok/s; TTFT and TPOT are in ms. Best per-row results are in **bold**.
> > >
> > > |||RTX PRO 6000||||H200||||
> > > |:---|:---|:---|:---|:---|:---|:---|:---|:---|:---|
> > > |||THETA+|1P+3D|2P+2D|3P+1D|THETA+|1P+3D|2P+2D|3P+1D|
> > > |**$c=128$**||||||||||
> > > |Prefill-heavy|Throughput|26,207|23,245|**42,577**|31,490|65,297|40,535|**82,400**|70,255|
> > > |(1024:128)|TTFT|**171**|4,385|476|482|**124**|2,763|662|412|
> > > ||TPOT|42.8|**14.6**|23.1|32.6|16.7|**6.4**|8.5|13.1|
> > > |Balanced|Throughput|11,063|**12,629**|11,640|9,023|28,686|**31,341**|29,388|21,963|
> > > |(512:512)|TTFT|**116**|316|229|272|**74**|249|178|233|
> > > ||TPOT|22.5|**19.2**|21.1|27.3|8.6|**7.5**|8.2|11.0|
> > > |Decode-heavy|Throughput|**7,690**|7,469|6,848|5,450|**19,336**|18,861|17,227|13,536|
> > > |(128:1024)|TTFT|**62**|223|166|220|**44**|233|179|220|
> > > ||TPOT|**18.2**|18.6|20.4|25.6|**7.2**|**7.2**|7.9|10.2|
> > > |**$c=512$**||||||||||
> > > |Prefill-heavy|Throughput|**47,322**|22,531|43,326|−|**105,432**|39,925|78,486|−|
> > > |(1024:128)|TTFT|**1,205**|21,412|9,245|−|**843**|12,357|5,741|−|
> > > ||TPOT|86.1|**14.6**|23.9|−|36.2|**6.4**|8.2|−|
> > > |Balanced|Throughput|**26,037**|25,080|20,397|−|**60,988**|46,261|48,895|−|
> > > |(512:512)|TTFT|**697**|4,013|1,676|−|**413**|5,753|1,128|−|
> > > ||TPOT|38.6|**29.9**|43.4|−|14.9|**9.2**|17.3|−|
> > > |Decode-heavy|Throughput|**18,396**|17,224|13,169|7,579|**46,782**|38,287|30,567|17,544|
> > > |(128:1024)|TTFT|**284**|1,842|1,074|2,295|**314**|1,908|1,216|1,720|
> > > ||TPOT|**28.2**|28.8|39.1|69.4|**10.8**|11.9|16.2|29.4|
> > > |**$c=1024$**||||||||||
> > > |Prefill-heavy|Throughput|**52,426**|23,013|43,971|−|**120,810**|41,943|78,986|−|
> > > |(1024:128)|TTFT|**3,876**|37,333|17,961|−|**2,386**|20,889|10,710|−|
> > > ||TPOT|141.1|**14.5**|23.2|−|57.3|**6.3**|8.2|−|
> > > |Balanced|Throughput|**30,704**|27,127|−|−|**65,472**|53,652|−|−|
> > > |(512:512)|TTFT|**2,062**|16,984|−|−|**1,987**|10,814|−|−|
> > > ||TPOT|47.9|**30.4**|−|−|26.4|**10.2**|−|−|
> > > |Decode-heavy|Throughput|**23,160**|19,476|14,068|−|**44,084**|42,051|32,668|16,250|
> > > |(128:1024)|TTFT|**1,217**|7,246|5,003|−|**2,621**|5,688|4,098|7,247|
> > > ||TPOT|44.2|**43.8**|67.8|−|21.8|**18.2**|26.8|57.3|
> > >
> > > Results for $c\in\lbrace 256,768\rbrace$ are available in the [extended table](https://i.imgur.com/RHSdFFP.png).
> > >
> > > **Analysis.** Both approaches can separate prefill and decode phases: disagg assigns dedicated GPUs so phases run **concurrently across GPUs**; THETA+ adaptively switches to exclusive batching under heavy per-GPU load so phases alternate **on the same GPU**, and defaults to CP under light load. We organize findings into four points.
> > >
> > > **(1) Disagg achieves higher throughput on prefill-heavy and balanced workloads at low concurrency (4/18 settings).** At $c=128$, per-GPU load under DP=4 is light ($\leq$32 req/GPU), leaving compute underutilized. Disagg’s dedicated GPUs run prefill and decode concurrently, yielding higher throughput: 2P+2D exceeds THETA+ by +62% (RTX) and +26% (H200) on prefill-heavy; 1P+3D leads by +9–14% on balanced. Disagg also achieves lower TPOT in these settings, as dedicated decode GPUs are never interrupted by prefill.
> > >
> > > **(2) THETA+ achieves higher throughput in the remaining 14/18 settings, covering all decode-heavy workloads and all $c\geq512$ settings.** As load increases, GPUs saturate (closing the utilization gap) while KV transfer overhead grows: (i) *High concurrency*: KV transfer bandwidth saturates as prefill GPUs flood decode GPUs, and decode-side KV occupancy grows without backpressure, causing OOM (3P+1D OOMs at $c\geq512$ for prefill-heavy; 2P+2D also OOMs at $c=1024$ for balanced). On H200 at $c=1024$, THETA+ leads by +53% on prefill-heavy and +22% on balanced. (ii) *Decode-heavy*: short prefills ($L=128$) offer little compute to parallelize yet KV overhead persists—THETA+ wins at all $c$ by up to +22% on H200.
> > >
> > > **(3) THETA+ achieves the lowest TTFT in all 18 settings**, typically $2\text{--}8\times$ lower. Under DP=4, every GPU handles prefill, so requests spread across all 4 GPUs; under disagg, only a subset serves prefill (e.g., 2 in 2P+2D, 1 in 1P+3D), concentrating queuing pressure on fewer GPUs and compounding delay non-linearly under heavy load.
> > >
> > > **(4) Disagg requires workload-specific tuning; THETA+ does not.** The optimal P:D ratio varies (2P+2D for prefill-heavy, 1P+3D for balanced), and the wrong choice is costly (1P+3D is 45% slower than 2P+2D on prefill-heavy at $c=128$). At high concurrency, disagg configs increasingly OOM while THETA+ completes all tested configurations—without tuning or failure.

---

### Official Review · Reviewer_XdSP · 2026-03-04

**Soundness:** 3
**Presentation:** 3
**Significance:** 3
**Originality:** 3
**Overall Recommendation:** 4
**Confidence:** 4

**Summary:**

This paper challenges the universal superiority of chunked prefill by demonstrating that on memory-bandwidth-constrained GPUs (e.g., RTX PRO 6000), interleaving prefill and decode phases causes resource contention that degrades throughput. The authors propose THETA, a scheduling framework for exclusive batching that utilizes a theoretical model to derive closed-form, asymptotically optimal phase-switching thresholds ($\theta^*$) under stochastic output-length distributions. By combining these thresholds with a probabilistic memory-safe batch sizing strategy and an online adaptive controller, THETA achieves up to 15.3% higher throughput than chunked prefill on bandwidth-limited hardware while maintaining competitive latency and memory safety.

**Compliance With Llm Reviewing Policy:**

Affirmed.

**Final Justification:**

The rebuttal addressed most of my main concerns. I will keep my original score.

**Key Questions For Authors:**

* The timing model in Section 3.1 adopts a linear iteration time formula for the prefill phase. Since attention complexity is theoretically quadratic ($O(L^2)$), what is the impact of neglecting the quadratic term on the framework's performance? Is there specific evidence or profiling data to verify that this assumption holds for the context lengths tested?

**Limitations:**

yes

**Strengths And Weaknesses:**

**Strengths**

* The paper provides a rigorous hardware-aware critique of chunked prefill, specifically quantifying the "crossover point" (80% decode ratio on H200 vs. 20% on RTX 6000) where mixed batches begin to underperform pure decode batches. This offers a necessary theoretical foundation for choosing scheduling strategies based on absolute memory bandwidth rather than just software-level heuristics.

* Instead of relying on empirical heuristics, the framework utilizes stochastic optimization to determine switching timings. It specifically accounts for real-world "Increasing Failure Rate" behavior, providing a formal mathematical justification for scheduling decisions.

* The evaluation demonstrates specific, substantial improvements on bandwidth-constrained GPUs, such as a 15.3% throughput increase on ShareGPT and 11.6% on WildChat.

**Weaknesses**

* The online adaptive algorithm relies on sliding windows to estimate hazard rates. In scenarios with "bursty" traffic or rapid distribution shifts (non-stationarity), the delay in parameter fitting could lead to transient throughput dips or, more critically, inaccurate memory-safe batch sizing.

* [Minor] The diagrams in Figure 2 are not very easy to read due to the color setting. The presentation could be improved.

---

> ### Author Rebuttal · Authors · 2026-03-31
>
> We thank the reviewer for the detailed and constructive comments.
>
> **W1: Sliding-window estimation may lag under bursty traffic or rapid distribution shifts.**
>
> As we show empirically below, the estimation lag has limited impact in practice. We further introduce **THETA+**—an extension that adaptively switches between CP and EB at runtime (see our response to Reviewer GmVf for derivation)—which achieves the highest throughput across all non-stationary settings tested.
>
> We distinguish two types of non-stationarity. **Traffic shifts** (e.g., 32→2048 clients): THETA+ detects these via $N_{\text{obs}}$ (EMA-smoothed batch occupancy), a fast-responding signal requiring no sliding-window estimation. Our response to Reviewer A6CD evaluates THETA+ under fixed concurrency levels; here we further test dynamic concurrency changes within a single workload (see experiments below). **Distribution shifts** (e.g., output length changes): These affect $\hat{p}\_0$, estimated over a sliding window of recent completions. During transients, (i) the wide throughput plateau (Figure 2, <6% variation across all $k^\*$) provides robustness to temporary estimation error in $\hat{p}\_0$; (ii) threshold clipping ($\hat{\theta}^* \in [\theta_{\min}, \theta_{\max}]$) prevents extreme $N^*$ values during transients, and vLLM's preemption mechanism provides a defense-in-depth safety net against temporary memory overcommit.
>
> **Non-stationary experiments.** We validate on two scenarios (Qwen3-8B). THETA+ achieves the highest throughput in all 4 (hardware × scenario) settings, with no OOM events. Throughput in tok/s.
>
> |  |  | RTX PRO 6000 |  |  | H200 |  |  |
> | :---- | :---- | :---- | :---- | :---- | :---- | :---- | :---- |
> | Scenario | Metric | CP | THETA (EB) | **THETA+** | CP | THETA (EB) | **THETA+** |
> | **Distribution shift** | Throughput | 3,456 | 4,467 | **4,752** | 18,307 | 17,394 | **20,776** |
> | ($\mu_o: 128 \to 512 \to 1024$) | TTFT (s) | 190.7 | 157.5 | **152.1** | **11.9** | 66.5 | 69.7 |
> |  | TPOT (ms) | 178.5 | 105.0 | **102.2** | 101 | **42.3** | 63.1 |
> | **Concurrency shift** | Throughput | 3,019 | 3,306 | **3,451** | 24,251 | 23,374 | **24,397** |
> | ($c=32 \to 512 \to 1024 \to 256 \to 2048$) | TTFT (s) | **24.0** | 29.8 | 26.6 | **7.7** | 13.7 | 9.4 |
> |  | TPOT (ms) | 142.4 | **51.6** | 77.5 | 77.9 | **45.6** | 66.8 |
>
> **(a) Distribution shift** ($\mu_L: 1024 \to 512 \to 128$, $\mu_o: 128 \to 512 \to 1024$; 2k requests/phase, max concurrency 2,048). On RTX PRO 6000, THETA+ outperforms both CP and THETA across all three metrics. On H200, THETA+ achieves +13.5% throughput over CP, but with higher latency than THETA. This is because CP→EB mode switching introduces a transient request backlog; on RTX PRO 6000, EB's large throughput advantage (+37%) clears this backlog quickly, whereas on H200 the advantage is smaller and the backlog takes longer to clear.
>
> **(b) Concurrency shift** ($c=32 \to 512 \to 1024 \to 256 \to 2048$). THETA+ switches to EB at $c\geq512$ on RTX PRO 6000 but only at $c=2048$ on H200, reflecting H200's higher crossover threshold. THETA+ achieves the highest throughput on both GPUs (+14.3% on RTX PRO 6000, +0.6% on H200), with better TTFT than THETA and better TPOT than CP.
>
> **W2 (Minor):** Thank you for the suggestion. We will improve Figure 2's color contrast.
>
> **Q1: Impact of neglecting the quadratic attention term.**
>
> Our theoretical results (Props. 3.1, 3.3 and Thm. 3.2) are derived under a linear timing model. Profiling confirms this approximation holds well ($R^2 > 0.996$) across all tested settings, with the quadratic term contributing negligibly to prefill time.
>
> We verify this across context lengths $L=256$–$4096$ (Qwen3-4B; [figure](https://i.imgur.com/3Q2cPlF.png)). On H200, a linear fit achieves $R^2 = 0.998$, and adding a quadratic term improves $R^2$ by only 0.001. On RTX PRO 6000 with $B=1,2,4,8$, all fits yield $R^2 > 0.996$ with nearly identical slopes, confirming that $T_p$ depends on total tokens rather than the $(B, L)$ decomposition. This is expected because GEMM dominates prefill computation: attention contributes $4Ld$ FLOPs per token versus $24d^2$ for projections and FFN, giving a ratio of $L/(6d) \approx 27\\%$ at $L=4096$ ($d=2560$); GQA further reduces this fraction. The quadratic attention cost is therefore a small fraction of overall prefill time, validating the linear approximation.

---

> > ### Author Rebuttal · Reviewer_XdSP · 2026-04-03
> >
> > Thank you for the rebuttal. For Q1, what will happen if the sequences gets longer (e.g., 32K, 64K, 128K tokens or even higher)? As chunked prefilling is typically used for long context and 4K tokens are not considered very long.

---

> > > ### Author Response · Authors · 2026-04-04
> > >
> > > Thank you for the follow-up question.
> > >
> > > **The linear approximation only needs to hold within the per-iteration token budget, not over the full input length.** We first apologize for the imprecise terminology: "CP" in our paper refers to the vLLM V1 default scheduler, not to the chunked-prefilling mechanism itself. Crucially, both CP and THETA use chunked prefilling — a long input is split across multiple iterations, each processing at most $B$ tokens (where $B$ = `max_num_batched_tokens`, typically 2K–8K, up to 16K on high-end GPUs). The only difference is *what* fills that budget: CP mixes prefill and decode tokens in a single iteration, while THETA dedicates each iteration exclusively to one phase. For a 128K-token input, this means $\lceil 128\text{K}/B \rceil$ iterations (e.g., ~16 at $B$=8K), and the timing model is applied per iteration on at most $B$ tokens. All experiments in our paper already operate under this token budget (Section 4.3). Therefore, the relevant question is not whether linearity holds at 128K, but whether it holds within the token budget range — which we validate below.
> > >
> > > **Validation within the practical regime.** We validate this across 7 models (3 dense architectures + 1 MoE, $d$ from 2048 to 5120) on RTX PRO 6000 Blackwell (96 GB). Within the practical token budget range (≤8K), **all models achieve linear R² > 0.986**; at ≤16K, R² > 0.979:
> > >
> > > | Range | Qwen3-4B | Qwen3-8B | Qwen3-14B | Qwen3-30B-A3B | Llama-3.2-1B | Llama-3.1-8B | Mistral-Nemo-12B |
> > > | :---- | :---- | :---- | :---- | :---- | :---- | :---- | :---- |
> > > | | d=2560 | d=4096 | d=5120 | d=2048 | d=2048 | d=4096 | d=5120 |
> > > | ≤ 4K | 0.9993 | **0.9999** | 0.9998 | 0.9939 | 0.9993 | 0.9996 | 0.9988 |
> > > | ≤ 8K | 0.9957 | **0.9994** | 0.9977 | 0.9863 | 0.9971 | **0.9996** | 0.9990 |
> > > | ≤ 16K | 0.9870 | 0.9907 | **0.9960** | 0.9790 | 0.9913 | 0.9918 | 0.9930 |
> > >
> > > **End-to-end serving at 128K.**  To confirm that these per-iteration results translate to real serving gains, we run end-to-end benchmarks across all 7 models at 128K input / 64 output tokens on RTX PRO 6000\. vLLM uses the default max\_num\_batched\_tokens=8192, so prefill is chunked into \~16 iterations per request and the timing model operates well within its validated linear regime (R² \> 0.99 at ≤8K).
> > >
> > > | Model | Concurrency | CP Throughput | THETA Throughput | Δ Throughput |
> > > | :---- | :---- | :---- | :---- | :---- |
> > > | Llama-3.2-1B | 4 | 26,664 | 27,265 | **+2.3%** |
> > > | Qwen3-30B-A3B | 4 | 4,826 | 5,021 | **+4.0%** |
> > > | Qwen3-4B | 4 | 6,134 | 6,314 | **+2.9%** |
> > > | Qwen3-8B | 3 | 5,621 | 5,764 | **+2.5%** |
> > > | Llama-3.1-8B | 3 | 6,331 | 6,482 | **+2.4%** |
> > > | Qwen3-14B | 3 | 3,830 | 3,883 | **+1.4%** |
> > > | Mistral-Nemo-12B | 3 | 4,774 | 4,877 | **+2.2%** |
> > >
> > > Concurrency is set to the maximum that avoids OOM under 128K input on a single RTX PRO 6000 (96 GB), since the KV cache for each request is large. Even at this low concurrency—where CP's scheduling overhead is well-amortized and contention is minimal—THETA still achieves **+1.4% to +4.0% higher throughput** across all 7 models, confirming that exclusive batching provides consistent gains at long context.
> > >
> > > **Stress test beyond practical settings (for completeness).** The above analysis establishes that practical serving never requires linearity beyond the token budget. For completeness, we additionally stress-test under the most extreme setting: forcing `max_num_batched_tokens`=131072 so that one iteration processes the entire 128K input without any chunking. This setting far exceeds typical deployment configurations, but even here the linear model retains R² ≥ 0.943 across all models, indicating that the approximation degrades gracefully rather than breaking down.
> > >
> > > | Range | Qwen3-4B | Qwen3-8B | Qwen3-14B | Qwen3-30B-A3B | Llama-3.2-1B | Llama-3.1-8B | Mistral-Nemo-12B |
> > > | :---- | :---- | :---- | :---- | :---- | :---- | :---- | :---- |
> > > | ≤ 32K | 0.9731 | 0.9852 | **0.9906** | 0.9675 | 0.9762 | 0.9872 | **0.9899** |
> > > | ≤ 64K | 0.9597 | 0.9753 | **0.9831** | 0.9549 | 0.9625 | 0.9777 | **0.9829** |
> > > | ≤ 128K | 0.9471 | 0.9607 | **0.9664** | 0.9434 | 0.9481 | 0.9631 | **0.9669** |
> > >
> > > **We also provide the full profiling curves here**
> > >
> > > [Prefill linearity profiling across 7 models (RTX PRO 6000)](https://i.imgur.com/xEQWubh.png)

---

### Official Review · Reviewer_A6CD · 2026-03-18

**Soundness:** 2
**Presentation:** 2
**Significance:** 3
**Originality:** 3
**Overall Recommendation:** 4
**Confidence:** 2

**Summary:**

This paper studies when chunked prefill underperforms exclusive batching for LLM inference and attributes the crossover primarily to memory-bandwidth constraints. It introduces THETA, an exclusive-batching scheduler with closed-form, asymptotically optimal phase-switching thresholds derived under stochastic output-length models and a probabilistic memory-safe batch-sizing rule; an online controller estimates hazard-rate parameters and adapts the threshold and batch size at runtime.

**Compliance With Llm Reviewing Policy:**

Affirmed.

**Final Justification:**

My concerns have been addressed.

**Key Questions For Authors:**

Please refer to Weaknesses.

**Limitations:**

Yes

**Strengths And Weaknesses:**

Strengths
1. The paper identifies a clear and practically important regime (bandwidth-constrained GPUs) where chunked prefill can be suboptimal and provides a principled alternative tailored to that regime.

Weaknesses
1. The paper states that chunked prefill favors TTFT and exclusive batching favors TPOT. When THETA wins on throughput but loses on TTFT, there is no formal treatment of how to balance TTFT vs. TPOT; the “competitive latency” claim is not tied to a specific latency target.
2. Most results are reported in a heavy-traffic, high-concurrency setting (2048). Extremely large TTFT numbers raise questions about the relevance for interactive workloads; evaluation under moderate loads or with SLO constraints would provide a fuller picture.

---

> ### Author Rebuttal · Authors · 2026-03-31
>
> We thank the reviewer for the thoughtful suggestions on strengthening the "competitive latency" claim and broadening evaluation.
>
> To address both points, we developed **THETA+**, which extends THETA with online adaptive mode selection between chunked prefill (**CP**) and exclusive batching (**EB**). THETA+ selects EB when the contention cost exceeds CP's fixed-cost advantage by a tunable margin $\delta$ (positive $\delta$ biases toward CP; negative toward EB); see our response to Reviewer GmVf for derivation:
>
> $$\beta_{\mathrm{CP}}^e(\hat{r}) - \beta_{\mathrm{EB}}^w > \frac{1}{\hat\mu_L + \hat\mu_o}\left[\frac{\alpha_p - \alpha_d \ln(1-\theta_0)\hat\mu_o}{\theta_0 \cdot N_{\text{obs}}} - \frac{\alpha_{\mathrm{CP}}(1+\hat\mu_o)}{N_{\text{obs}}}\right] + \delta$$
>
> [Crossover condition visualized (see our response to Reviewer GmVf for details)](https://i.imgur.com/wQqs81U.png)
>
> This criterion is (i) **traffic-aware**—low $N_{\text{obs}}$ makes the RHS large, selecting CP; high $N_{\text{obs}}$ shrinks it, selecting EB when contention dominates; and (ii) **workload-aware**—$\beta_{\mathrm{CP}}^e(\hat{r})$ adapts to the current decode ratio.
>
> **W1: No formal TTFT-vs-TPOT treatment; "competitive latency" not tied to a latency target.**
>
> THETA+ controls the TTFT-TPOT tradeoff through $\delta$: increasing $\delta$ biases toward CP (lower TTFT, higher TPOT); decreasing it biases toward EB (higher throughput, lower TPOT). Under heavy load, maximizing throughput (Eq. 1) implicitly minimizes TTFT—queuing delay dominates and scales inversely with throughput (RTX PRO 6000, Qwen3-8B, $c=2048$: TTFT 68.2s vs CP's 83.8s). The SLO evaluation below (W2) further shows THETA+ achieves the highest goodput across multiple TTFT/TPOT targets.
>
> **W2: Evaluation limited to heavy-traffic regime.**
>
> We evaluate THETA+ across three traffic levels ($\mu_L=512, \mu_o=256$). Throughput in tok/s.
>
> ||RTX PRO 6000|||H200|||
> |:-|:-|:-|:-|:-|:-|:-|
> ||CP|THETA (EB)|**THETA+**|CP|THETA (EB)|**THETA+**|
> |**Low** ($c=32$) Throughput|**1,628**|1,277|**1,621**|**11,584**|8,609|**11,586**|
> |TTFT (ms)|**64**|2,183|**64**|**49**|1,046|**50**|
> |TPOT (ms)|19.4|**16.5**|19.5|8.2|**7.1**|8.2|
> |**Moderate** ($c=512$) Throughput|2,992|4,422|**4,476**|**27,464**|27,207|**27,460**|
> |TTFT (s)|**1.2**|8.3|5.1|**0.7**|4.6|**0.7**|
> |TPOT (ms)|164.2|**77.6**|90.4|52.7|**36.8**|52.7|
> |**High** ($c=2048$) Throughput|3,242|4,375|**4,477**|26,368|**27,198**|27,043|
> |TTFT (s)|83.8|70.8|**68.2**|**24.4**|34.6|33.7|
> |TPOT (ms)|207.3|**82.7**|101.6|128|**72.8**|79.2|
>
> At low traffic, THETA+ selects CP, preserving TTFT (64ms / 49ms). At $c=2048$ on RTX PRO 6000, THETA+ achieves both the highest throughput (+38%) and lowest TTFT (68.2s vs 83.8s). At $c=512$, THETA+ gains +49.6% throughput with lower TPOT (90.4 vs 164.2ms) but higher TTFT—quantified via SLO attainment below.
>
> Real deployments rarely sustain fixed traffic; THETA+'s EMA-smoothed $N_{\text{obs}}$ enables rapid adaptation without manual re-tuning. Under non-stationary workloads (Qwen3-8B), THETA+ achieves the highest throughput in all 4 (hardware × scenario) settings:
>
> |||RTX PRO 6000|||H200|||
> |:-|:-|:-|:-|:-|:-|:-|:-|
> |Scenario|Metric|CP|THETA (EB)|**THETA+**|CP|THETA (EB)|**THETA+**|
> |**Distribution shift**|Throughput|3,456|4,467|**4,752**|18,307|17,394|**20,776**|
> |($\mu_o: 128 \to 512 \to 1024$)|TTFT (s)|190.7|157.5|**152.1**|**11.9**|66.5|69.7|
> ||TPOT (ms)|178.5|105.0|**102.2**|101|**42.3**|63.1|
> |**Concurrency shift**|Throughput|3,019|3,306|**3,451**|24,251|23,374|**24,397**|
> |($c=32 \to 512 \to 1024 \to 256 \to 2048$)|TTFT (s)|**24.0**|29.8|26.6|**7.7**|13.7|9.4|
> ||TPOT (ms)|142.4|**51.6**|77.5|77.9|**45.6**|66.8|
>
> On RTX PRO 6000, THETA+ leads CP by +37.5% under distribution shift and +14.3% under concurrency shift, with consistently better TTFT than THETA. On H200, THETA+ achieves +13.5% throughput under distribution shift and +0.6% under concurrency shift, demonstrating effective mode switching across hardware.
>
> **SLO-constrained goodput.** At moderate load ($c=512$), THETA+ trades higher throughput for longer TTFT. To quantify this tradeoff, we report per-request SLO attainment (% meeting both targets) on RTX PRO 6000:
>
> ||TPOT < 50ms (strict)|||TPOT < 100ms (relaxed)|||
> |:-|:-|:-|:-|:-|:-|:-|
> |TTFT target|CP|THETA (EB)|**THETA+**|CP|THETA (EB)|**THETA+**|
> |**< 2s**|**0.6**|0.0|0.0|**4.9**|0.0|1.2|
> |**< 5s**|0.6|0.0|**1.1**|5.0|6.2|**48.4**|
> |**< 10s**|0.6|0.5|**1.3**|5.8|77.3|**80.3**|
>
> Under strict TPOT (< 50ms), all schedulers fail as mean TPOT exceeds the target. Under relaxed TPOT (< 100ms), CP collapses to ~5% while THETA+ achieves the best balance (80.3% at TTFT < 10s, 48.4% at < 5s).

---

> > ### Author Rebuttal · Reviewer_A6CD · 2026-04-03
> >
> > My concerns have been addressed, and I will raise my score.

---

> > > ### Author Response · Authors · 2026-04-03
> > >
> > > We sincerely thank the reviewer for the positive reassessment and for the constructive feedback.

---

### Official Review · Reviewer_GmVf · 2026-03-23

**Soundness:** 3
**Presentation:** 3
**Significance:** 4
**Originality:** 3
**Overall Recommendation:** 4
**Confidence:** 4

**Summary:**

This paper studies LLM inference scheduling on GPUs with limited memory bandwidth. It argues that the commonly-used chunked prefill strategy is not always the best choice because mixing prefill and decode can increase memory-bandwidth contention and hurt decode efficiency on memory-bandwidth-constrained hardware. To address this, the paper proposes THETA, a threshold-based exclusive batching framework that derives analytical phase-switching thresholds and memory-safe batch sizes, and adapts them online based on workload statistics. Experiments on several GPUs and workloads show that THETA can outperform chunked prefill, especially on bandwidth-constrained GPUs, by to 1.15x higher throughput while not significantly affect latency.

**Compliance With Llm Reviewing Policy:**

Affirmed.

**Key Questions For Authors:**

* Since THETA does not always outperform its baseline, can you provide a more clear guideline on when to use THETA?

**Limitations:**

Yes

**Strengths And Weaknesses:**

Strengths:

* The paper challenges the common-sense of using chunked prefill is always better and propose adaptive chunked prefill with decoding.
* The proposed approaches have been validated through extensive experiments on multiple GPUs.

Weaknesses:

* The proposed approach, on some cases, is still slower than its chunked-prefill baseline, especially on bandwidth-constrained GPUs or on benchmarks with long context.

---

> ### Author Rebuttal · Authors · 2026-03-31
>
> We thank the reviewer for the constructive feedback.
>
> **W1 / Q1: THETA does not always outperform CP; clearer guideline on when to use THETA?**
>
> We abbreviate exclusive batching as **EB** and chunked prefill as **CP**.
>
> **Analytical framework.** We extend the timing model (Section 3.1) to derive approximate throughput under both strategies, from which we can conclude the conditions under which a strategy is more preferable. All symbols are as defined in Sections 3.1--3.2.
>
> *EB throughput.* Substituting the fluid-approximation expected decode time (Appendix A) into the throughput objective (Eq. 1) at $k^* = \theta_0 N$:
>
> $$\mathrm{TP}\_{\mathrm{EB}}(k^\*, N) = \left[\frac{\alpha\_p - \alpha_d \ln(1-\theta\_0) \mu_o}{k^*} + \beta\_{\mathrm{EB}}^w (\mu_L + \mu_o)\right]^{-1}$$
>
> where $\beta_{\mathrm{EB}}^w = (\beta_p \mu_L + \beta_d \mu_o)/(\mu_L + \mu_o)$ is the workload-weighted per-token cost under exclusive execution.
>
> *CP throughput.* Under steady-state, CP prefills $N_p$ new requests and decodes $N_d = N_p \mu_o$ ongoing ones in each iteration, resulting in a batch size $N = N_p(1+\mu_o)$ and a throughput:
>
> $$\mathrm{TP}\_{\mathrm{CP}}(N) = \left[\frac{\alpha_{\mathrm{CP}}(1+\mu_o)}{N} + \beta_{\mathrm{CP}}^e (\mu_L + \mu_o)\right]^{-1}$$
>
> where $\alpha_{\mathrm{CP}}$ is the mixed-batch fixed overhead and $\beta_{\mathrm{CP}}^e$ is the effective per-token cost in a mixed batch at steady-state decode ratio $r^* = \mu_o/(\mu_L + \mu_o)$, capturing bandwidth contention (from the same profiling as Figure 1).
>
> *Crossover condition.* Under these modeling assumptions, CP outperforms EB when:
>
> $$\underbrace{\beta\_{\mathrm{CP}}^e - \beta\_{\mathrm{EB}}^w}\_{\text{contention cost}} < \underbrace{\frac{1}{\mu_L + \mu_o}\left[\frac{\alpha_p - \alpha_d \ln(1-\theta_0)\mu_o}{k^\*} - \frac{\alpha_{\mathrm{CP}}(1+\mu_o)}{N}\right]}_{\text{CP's amortized fixed-cost advantage}}$$
>
> The LHS measures extra per-token cost from mixing prefill and decode. The RHS captures CP's scheduling advantage from processing both phases per iteration; it scales as $O(1/N)$, shrinking as batch size grows. In the figure below, CP is preferred when $\beta_{\mathrm{CP}}^e$ (solid) falls below the blue dashed threshold, and THETA is preferred otherwise. On H200, which exhibits low degree of concavity in Attention time when prefill and decode are mixed, the condition is satisfied at most decode ratios; on RTX PRO 6000, which exhibits a much higher degree of concavity, contention inflates $\beta_{\mathrm{CP}}^e$ above the threshold, favoring THETA.
>
> [Crossover condition (Qwen3-4B, 4096 tokens, RTX PRO 6000 vs. H200)](https://i.imgur.com/wQqs81U.png)
>
> **Practical deployment guideline.** Based on the crossover analysis and end-to-end experiments (Tables 2--3), THETA is preferred over CP when:
>
> - **(a) Low memory bandwidth.** RTX PRO 6000 (1.792 TB/s) generally favors THETA; H200 (4.8 TB/s) generally favors CP. Even on high-bandwidth hardware, heavy traffic can amplify contention enough to favor THETA.
> - **(b) Small model-to-memory ratio.** A smaller model leaves more memory for KV cache, enabling larger batches that increase contention. Larger models limit batch size and raise fixed overheads, widening CP's amortized advantage. On RTX PRO 6000, THETA improves throughput for Qwen3-8B across all four workloads (+1.4% to +15.3%, avg. +7.9%), but gains diminish for Qwen3-30B-A3B (avg. +2.8%). On H200, the same trend holds (Qwen3-8B: avg. +0.8%; Qwen3-30B-A3B: avg. −6.1%).
> - **(c) Moderate decode ratios.** Contention peaks at intermediate decode ratios, measured by $r = \mu_o/(\mu_L + \mu_o)$, and diminishes at the two ends; it also requires consistently full batches—under low utilization, contention is negligible and CP's scheduling efficiency dominates. On RTX PRO 6000 (Qwen3-8B): ShareGPT ($r\approx73\%$, +15.3%) and WildChat (multi-turn; +11.3%) show the largest gains, while prefill-dominated LongBench ($r\approx0.4\%$, +3.7%) and decode-dominated NuminaMath ($r\approx89\%$, +1.4%) show smaller gains.
>
> **Automatic runtime dispatch (THETA+).** We further design **THETA+**, which uses the crossover condition as an online CP↔EB switching criterion. THETA+ continuously estimates $\hat{r}$ and $N_{\text{obs}}$ and selects EB whenever $\beta_{\mathrm{CP}}^e(\hat{r}) - \beta_{\mathrm{EB}}^w$ exceeds the RHS by a tunable margin $\delta$. Because the RHS is inversely related to $N_{\text{obs}}$, the mechanism is inherently traffic-aware: under light load (small $N_{\text{obs}}$), the threshold is high and THETA+ defaults to CP; under heavy load (large $N_{\text{obs}}$), the threshold shrinks and THETA+ switches to EB when contention dominates. Empirically, at $c=32$ THETA+ matches CP's TTFT exactly, while at $c=2048$ on RTX PRO 6000 it achieves +38% throughput over CP with the lowest TTFT. **Full results:** see our responses to Reviewers A6CD (SLO goodput) and XdSP (non-stationarity).

---

> > ### Author Rebuttal · Reviewer_GmVf · 2026-04-03
> >
> > I have read the rebuttal and it answers all my questions.

---

### Decision · Program_Chairs · 2026-04-30

**Decision:**

Accept (regular)

**Comment:**

The reviewers agree that this is a technically solid and practically relevant systems paper. The authors strive to analyze a central area in LLM serving: when chunked prefill ceases to be the right default under memory-bandwidth constraints. An important area presented by this article is the hardware-aware scheduling tradeoff between chunked prefill and exclusive batching. Reviewers found the main contribution meaningful: the paper identifies a clear bandwidth-constrained regime where exclusive batching can outperform chunked prefill, and supports this with both analysis and broad experiments.

The main concerns were about scope rather than correctness: limited treatment of TTFT/TPOT tradeoffs and SLOs, behavior under non-stationary workloads, long-context validity of the linear timing model, and comparison to disaggregation / multi-GPU settings. In the rebuttal, the authors provided additional analyses and experiments addressing these points, including THETA+, SLO-style evaluation, non-stationary settings, long-context evidence, and multi-GPU/disaggregation comparisons. Most reviewers explicitly said their concerns were fully or largely resolved.

Overall, my recommendation is Weak Accept: the paper makes a useful, well-supported contribution, especially for bandwidth-constrained inference, though its impact is somewhat limited by the fact that benefits are regime-dependent and some latency tradeoffs remain workload-sensitive.